# Meta-analysis of randomized controlled trials examining social comparison as a behaviour change technique across the behavioural sciences

Thole H. Hoppen [1,3] ✉, Rieke M. Cuno [1,3], Janna Nelson[1], Frederike Lemmel[1], Pascal Schlechter[1] & Nexhmedin Morina [1,2]

Research on social comparison as a behaviour change technique (SC-BCT) has increased substantially. We conducted a random-effects meta-analysis of randomized controlled trials investigating SC-BCTs across the behavioural sciences (PROSPERO: CRD42022343154). We searched MEDLINE, PsycINFO and Web of Science from inception to January 2024. Seventy-nine randomized controlled trials ($N = 1,356,521$) investigating effects on behaviours related to climate change mitigation, health, performance and service were included. In the short term (mean 3.7 months post-intervention), SC-BCTs produced small effects relative to both passive (Hedges' $g = 0.17$; 95% confidence interval, 0.11–0.23; $k = 37$; $P < 0.001$) and active control conditions ($g = 0.23$; 95% confidence interval, 0.15–0.31; $k = 42$; $P < 0.001$). A greater number of SC-BCT sessions and emphasis on desired (versus undesired) behaviours were associated with larger effects. Moderation effects were observed in only a few analyses, highlighting the need for further testing. SC-BCTs also produced significant small effects in the long term (mean 6.2 months post-intervention). Small effects should be interpreted in the context of low cost and scalability (for example, sending one or two emails). Certainty of evidence, using GRADE criteria, ranged from low to moderate depending on the analysis. More high-quality research is needed.

Humans have an innate tendency to compare themselves to others[1,2]. Social comparison—thinking about social information in relation to the self[3]—is ubiquitous in human cognition[4]. Comparing oneself to others serves self-motives and is involved in various social behaviours and coping processes[5–9]. Accordingly, social comparison exerts a fundamental influence on people's private, public and collective behaviour[7].

Social comparison is a process that involves selecting a social standard (for example, another individual), evaluating (dis-)similarities between the self and the standard on a particular dimension (for

example, energy usage), and reacting to the comparison outcome (for example, behavioural adaptation or maintenance)[5]. Social comparison standards can be perceived as superior (upward), similar (lateral) or inferior (downward) to the self. Various intra-personal factors such as self-motives[10], cognitive resources, perceived malleability of the dimension or self-efficacy beliefs[11] can affect the cognitive, affective and behavioural reactions to social comparison. Similarly, contextual factors can influence social comparison processes. For instance, social comparison increases in the wake of uncertainty or threat[6,12,13]. This

[1]Institute of Psychology, University of Münster, Münster, Germany. [2]Department of Psychology, New School for Social Research, New York, NY, USA. [3]These authors contributed equally: Thole H. Hoppen, Rieke M. Cuno. ✉e-mail: thoppen@uni-muenster.de

relates to both collective crises (for example, climate change[14]) and private challenges (for example, major health threats[13]). Crucially, social comparison has been associated with both adaptive and maladaptive coping behaviour. For example, during the COVID-19 pandemic, social comparison was related to increased risk-reduction behaviour[15] and increased well-being[16]. Yet, frequent upward social comparison has been associated with poorer mental and physical health outcomes[13,17,18].

The potential of social comparison as a behaviour change technique (SC-BCT) to increase desired behaviours (for example, recycling or sunscreen use)[19–21] or to decrease undesired behaviours (for example, the use of finite resources or alcohol consumption)[19,22,23] has been investigated in numerous randomized controlled trials (RCTs) and some domain-specific meta-analytic syntheses thereof. Attention to SC-BCTs has increased substantialy during the past few decades[19,24–31]. Yet, a systematic review and meta-analysis on the use and efficacy of SC-BCTs across the behavioural sciences is lacking. The present work attempts to fill this gap by means of a comprehensive systematic review and meta-analysis summarizing RCTs across the behavioural sciences. We were interested in how social comparison was used as a BCT in RCTs and to what effect. Accordingly, our work attempted to answer the following two main research questions: (1) How effective are SC-BCTs when applied as the primary intervention? (2) How effective are SC-BCTs when added to a BCT bundle? To this end, we aimed to include only literature explicitly referring to social comparison or related terms.

## Results

### Included trials
The PRISMA flowchart provides an overview of the study synthesis (Fig. 1). Of the identified 18,058 unique hits after duplicate deletion, 17,314 were excluded on the basis of screened title and abstracts for not meeting inclusion criteria. One potentially relevant full text was inaccessible. Of the 744 records entering the full-text screening, 669 were excluded for not meeting inclusion criteria. Twelve of these[32–43] were principally eligible but did not report data in a usable format to calculate Hedges' $g$, and the primary authors did not respond to at least two data request emails. Two other principally eligible publications[44,45] targeted preventing undesired behaviour from occurring (that is, alcohol consumption on a specific future occasion) and included various participants who did not engage in the undesired behaviour at baseline (that is, abstainers) and were thus excluded. Lastly, four other publications[46–49] reported on RCTs that assessed only long-term cognitive or affective change following SC-BCTs but not behaviour change, and were thus excluded. A short summary of each of these 18 aforementioned publications is provided in Supplementary Appendix C. The exclusion of all other full texts was straightforward. In total, 74 publications reporting on 79 independent RCTs were included in the present meta-analysis[20,21,23,50–120]. Five publications[50–52,86,113] each reported on two independent eligible RCTs. We received data for 26 trials[21,23,50–54,58,59,61,68,80,81,93,99,102,105,110,111,113,115–119] via email communication.

### Meta-analytic synthesis
**Basic trial characteristics.** Supplementary Appendix D provides an overview of the characteristics of the trials included in the meta-analytic synthesis. The 79 RCTs involved a total of $N = 1,356,521$ participants. A total of 71 RCTs investigated the efficacy of SC-CBTs as the primary intervention (that is, the first research question), whereas 8 RCTs[72,75,82,98,100,103,104,109] investigated the efficacy of SC-BCTs as an add-on intervention (that is, the second research question). One dismantling RCT investigated both research questions[109]. Of those assessing the efficacy of SC-BCTs as the primary intervention, 14 independent RCTs reported in 13 publications[20,63–65,73,74,78,85,86,90,91,96,99] investigated SC-BCTs as a stand-alone BCT. The remaining 57 independent RCTs reported in 54 publications[21,23,50–62,66–71,76,77,79–81,83,84,87–89,92–95,97,101,102,105–120] investigated SC-BCTs in conjunction with other BCTs. Provision of intra-individual

feedback on the behavioural dimension, alongside social feedback on the same dimension, delivered to participants on two or more occasions (that is, enabling both temporal and social comparison) was the most common complementary BCT accompanying the SC-BCT among the 56 RCTs investigating SC-BCT as the primary (but not stand-alone) intervention. Most data were from the USA and other high-income countries. On average (that is, unweighted mean across the 79 RCTs), short-term assessments took place (or covered for continuous assessments) about 3.7 months post-intervention (mean, 110 days; s.d., 193 days). Long-term assessments took place (or covered for continuous assessments) about 6.2 months post-intervention (mean, 187 days; s.d., 179 days) on average (that is, unweighted mean) across the 23 RCTs assessing behaviour change more than once. While passive control conditions were homogenous (that is, assessments only), active control conditions varied considerably, with intra-individual feedback on the target dimension being the most commonly applied active control condition. Mean age ranged from 9.3 to 65.4 years with a weighted mean across trials of 38.92 years (s.d., 14.50 years). While some studies included a large age range (for example, 17 to 74 years[23]), others included a selective age range such as adolescents and young adults only (for example, 16 to 20 years[69]) or older adults (60+ years old)[76]. About half of the trials (48% or $k = 38$) involved student/pupil samples. While usually referred to as convenience samples, many of these trials involved very large samples (for example, up to 105 schools in one study[103]). Most other trials involved general population samples, with most of these investigating effects on climate change mitigation behaviour such as water or electricity usage on a population level[54,62,64,67] and some of these investigating behaviour change on a population level such as behaviour change regarding traffic violations[23]. Lastly, a few studies targeted other selective populations (that is, non-student) such as primary care physicians[97]. Two studies involved female participants only[88,96], and one study involved male participants only[69]; all other studies involved mixed-gender samples. Thirty-eight trials (48%) used online methods (for example, email or apps such as leaderboards) to apply SC-BCTs. In 12 trials (15%), the SC-BCT was applied personally in a lab setting. Another 16 trials (20%) sent SC-BCT letters home. Other studies sent SC-BCT letters to schools ($k = 7$; 9%) or work environments ($k = 5$; 6%). One trial ($k = 1$; 1%) applied the SC-BCT in a hospital (that is, to inpatients before discharge)[79].

**Risk of bias.** Figure 2 provides all risk-of-bias assessments. In most trials ($k = 63$; 80%), some concern of bias emerged from the randomization process. In eight trials (10%), the risk of bias was judged to be low. Conversely, eight trials (10%) were judged to be at high risk of bias due to concern with randomization. Half of the trials ($k = 41$; 52%) had low risk of bias due to deviations from the intended interventions, 29% of trials ($k = 23$) had some concern and 19% of trials ($k = 15$) had high risk of bias. Most studies ($k = 65$; 82%) had low risk of bias due to missing outcome data, and the remaining trials had high risk of bias. Most studies ($k = 74$; 94%) had low risk of bias arising from the outcome assessment, and the remaining trials had high risk of bias. Lastly, most trials ($k = 66$; 84%) had some concern of bias regarding selection of the reported result(s) given that preregistrations and prespecified analysis protocols were rare. The remaining 13 trials (16%) had low risk of bias regarding selection of the reported result(s).

### Results for research question 1
**Efficacy of SC-BCTs compared to passive control conditions.** Table 1 presents the results of all overarching analyses. In the short term, SC-BCTs produced a significant ($P < 0.001$) small effect in terms of behaviour change in the intended direction relative to passive control conditions ($g = 0.17$; 95% confidence interval (CI), 0.11–0.23; $k = 37$; $I^2 = 94\%$; Fig. 3). This analysis involved 578,792 independent participants examined across 37 RCTs. High and significant heterogeneity in outcomes was found. The results remained similar after two outliers

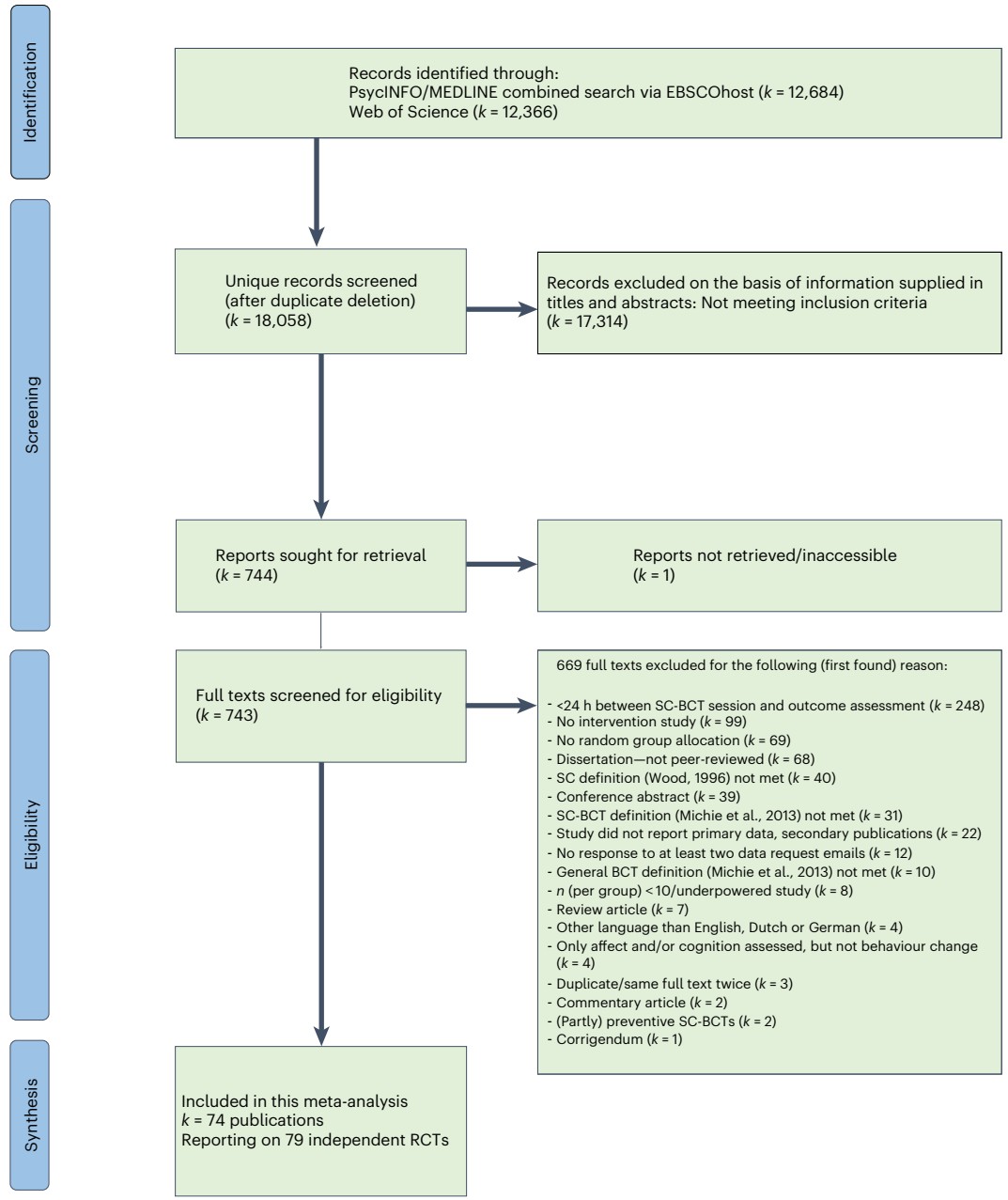

**Fig. 1 | Study process.** PRISMA flow chart depicting the study synthesis process.

were removed ($g = 0.15$; 95% CI, 0.09–0.21; $k = 35$; $I^2 = 94\%$). Certainty of evidence was rated as low due to concern about risk of bias and considerable significant unexplained heterogeneity between outcomes. Risk of bias emerged from insufficient reporting of the randomization process, deviations from the intended intervention (for example, study participants or interventionists were likely to be aware of the assigned intervention), missing outcome data (for example, data were not available for all randomized participants) or lack of predefined analysis protocols in several studies. Significant unexplained heterogeneity was addressed by multiple sub-analyses. Non-important heterogeneity was found for four sub-analyses: intended upward SC-BCTs (across outcomes), desired outcomes only, health outcomes only and performance outcomes only. No concern regarding indirectness emerged from the selection of the population (that is, the majority of trials presented directness regarding population, whereas ten trials presented probable indirectness regarding population because they investigated a relatively specific sample without providing a rationale

for its selection—for example, examining the efficacy of a SC-BCT for climate change mitigation behaviour in a student sample). No concern of indirectness emerged regarding intervention (that is, social comparison and SC-BCT definitions were always met; see inclusion criteria), comparator (that is, assessment only in passive control conditions) and outcome (that is, all behavioural outcomes matched the given social comparison dimension in SC-BCTs). All directness (versus indirectness) ratings of evidence for each included RCT using GRADE criteria are detailed in Supplementary Appendix E (that is, for this analysis as well as all other overarching analyses). The CI for the pooled effect excluded the null, signalling confidence in a significant behaviour change in the desired direction. Egger's test did not indicate significant small study effects (that is, publication bias was deemed unlikely). Most studies targeted a decrease in undesired behaviour ($k = 32$), whereas only a few studies targeted an increase in desired behaviour ($k = 5$). Most studies investigated climate change mitigation behaviour ($k = 20$), followed by health behaviour ($k = 13$) and performance behaviour ($k = 4$).

In the long term, data availability was considerably more limited ($k = 13$). The results remained similar to those in the short term. In the overarching analysis across all data at follow-up, SC-BCTs produced a highly significant ($P < 0.001$) small effect relative to passive control conditions ($g = 0.10$; 95% CI, 0.06–0.13; $k = 13$; $I^2 = 0\%$). Heterogeneity in outcomes was low and non-significant. The results remained very similar after one outlier was excluded ($g = 0.10$; 95% CI, 0.06–0.13; $k = 12$; $I^2 = 0\%$). Certainty of evidence was rated to be moderate due to concern about risk of bias, such as insufficient reporting of the randomization process, missing outcome data (for example, data were not available for all randomized participants) or missing predefined analysis protocols in several studies. Statistical analyses revealed non-important and non-significant heterogeneity between the outcomes. No concern of indirectness emerged regarding population (that is, the majority of trials presented directness regarding population, whereas three trials presented probable indirectness regarding population due to investigating a rather specific sample without presenting a rationale for its selection). Similarly, no concern of indirectness emerged in relation to intervention (that is, social comparison and SC-BCT definitions were always met), comparator (that is, assessment only in passive control conditions) and outcome (that is, all behavioural outcomes matched the given social comparison dimension in SC-BCTs). The CI for the pooled effect excluded the null, signalling confidence in a significant change in behaviour in the desired direction. Egger's test did not indicate significant small study effects (that is, publication bias was deemed unlikely). Most trials reporting follow-up data targeted health ($k = 5$) or climate change mitigation behaviour ($k = 6$).

**Efficacy of SC-BCTs compared to active control conditions.** Table 1 displays the results from the overarching analyses. The most commonly used control condition involved the provision of intra-individual feedback on the behavioural dimension. Other examples of active control conditions were shaping knowledge (for example, instructions on how to lower electricity usage) and goal setting (for example, setting a goal concerning lowing electricity usage). See Supplementary Appendix D (column 9) for all active control conditions. In the overarching analysis in the short term, SC-BCTs were also significantly ($P < 0.001$) more efficacious in changing behaviour in the intended direction relative to active control conditions in the short term, with a small pooled effect ($g = 0.23$; 95% CI, 0.15–0.31; $k = 42$; $I^2 = 96\%$; Fig. 4). This analysis involved 148,233 independent participants examined across 42 RCTs. Heterogeneity in outcomes was high and significant. The results remained similar when two statistical outliers were removed. Egger's test indicated significant small study effects (that is, potentially due to publication bias), and the trim-and-fill method added ten studies to the left to establish symmetry. The results remained similar ($g = 0.15$; $P < 0.01$; 95% CI, 0.05–0.25; $k = 52$; $I^2 = 98\%$). Certainty of evidence was rated very low due to concern about risk of bias, considerable unexplained heterogeneity between outcomes and potential publication bias. Risk of bias emerged from insufficient reporting of the randomization process, deviations from the intended intervention (for example, participants or people delivering the intervention were likely to be aware of the assigned intervention), missing outcome data (for example, data were not available for all randomized participants) or lack of predefined analysis protocols in several studies. Non-important heterogeneity was found for two sub-analyses: clearly upward SC-BCTs (across outcomes) and service outcomes only. No concern of indirectness emerged regarding population (that is, the majority of trials presented directness regarding population, whereas 16 trials presented probable indirectness regarding population due to investigating a rather specific sample without presenting a rationale for its selection). Similarly, no concern of indirectness emerged regarding intervention (that is, social comparison and SC-BCT definitions were always met), comparator (that is, active control conditions as a comparator regarding behaviour change) and outcome (that is, all

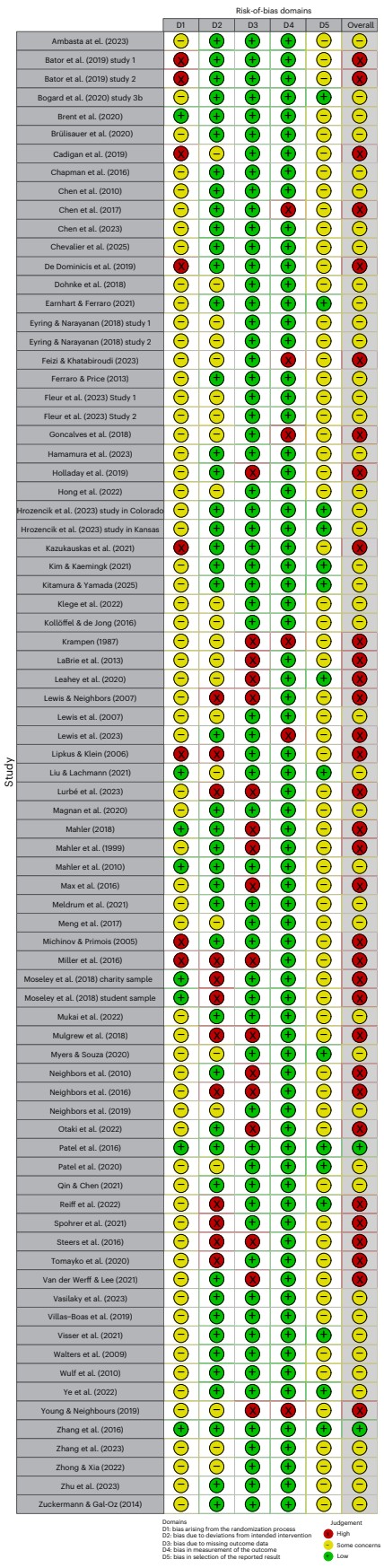

**Fig. 2 | Risk-of-bias assessments.** Risk of bias of the included studies.

## Table 1 | Efficacy of SC-BCT

| Comparison | k (N) | g | P | 95% CI (95% PI) | I² (%) | CoE |
|---|---|---|---|---|---|---|
| **Post-intervention results: short-term efficacy** | | | | | | |
| SC-BCTs versus passive control conditions (Fig. 3) | 37 (578,792) | 0.17*** | <0.001 | 0.11 to 0.23 (−0.12 to 0.46) | 94.35*** | ++ |
| SC-BCTs versus passive control conditions (outlier-adjusted) | 35 (578,658) | 0.15*** | <0.001 | 0.09 to 0.21 (−0.11 to 0.42) | 93.83*** | NA |
| SC-BCTs versus active control conditions (Fig. 4) | 42 (148,233) | 0.23*** | <0.001 | 0.15 to 0.31 (−0.17 to 0.63) | 96.06*** | + |
| SC-BCTs versus active control conditions (trim-and-fill-adjusted) | 52 (NA) | 0.15** | 0.003 | 0.05 to 0.25 | 97.82*** | NA |
| SC-BCTs versus active control conditions (outlier-adjusted) | 40 (148,135) | 0.20*** | <0.001 | 0.13 to 0.28 (−0.14 to 0.55) | 95.13*** | NA |
| **Follow-up results: long-term efficacy** | | | | | | |
| SC-BCTs versus passive control conditions | 13 (48,057) | **0.10***** | <0.001 | **0.06 to 0.13 (0.06 to 0.13)** | 0.03 | +++ |
| SC-BCTs versus passive control conditions (outlier-adjusted) | 12 (47,993) | **0.10***** | <0.001 | **0.06 to 0.13 (0.06 to 0.13)** | 0.00 | NA |
| SC-BCTs versus active control conditions | 6 (19,350) | 0.24* | 0.027 | 0.03 to 0.45 (−0.20 to 0.76) | 76.54** | ++ |

CoE, certainty of evidence as assessed with GRADE criteria (+++, moderate certainty; ++, low certainty; +, very low certainty); NA, not applicable; PI, prediction interval. $I^2$ is the heterogeneity in outcomes, with asterisks indicating the statistical significance level of the corresponding Q statistic; k is the number of independent data points included in the analysis; N is the total number of (independent) participants included in the respective meta-analysis; and P is the two-sided P value indicating the statistical significance level of the pooled effect (Hedges' g). Bold font indicates that both the 95% CI and the 95% PI excluded the null, highlighting particular statistical certainty in the significance of the difference in behaviour change observed between the respective comparison groups. The values are standardized mean differences (that is, Hedges' g) derived from the random-effects meta-analysis. Positive values of g indicate that the social comparison conditions (SC-BCTs) showed a higher pooled behaviour change (that is, a higher increase for desired behaviour and a higher decrease for undesired behaviour) than the passive or active control conditions and vice versa. Please note that the trim-and-fill method (adjustment for funnel plot asymmetry) does not supply PIs. *P < 0.05; **P < 0.01; ***P < 0.001.

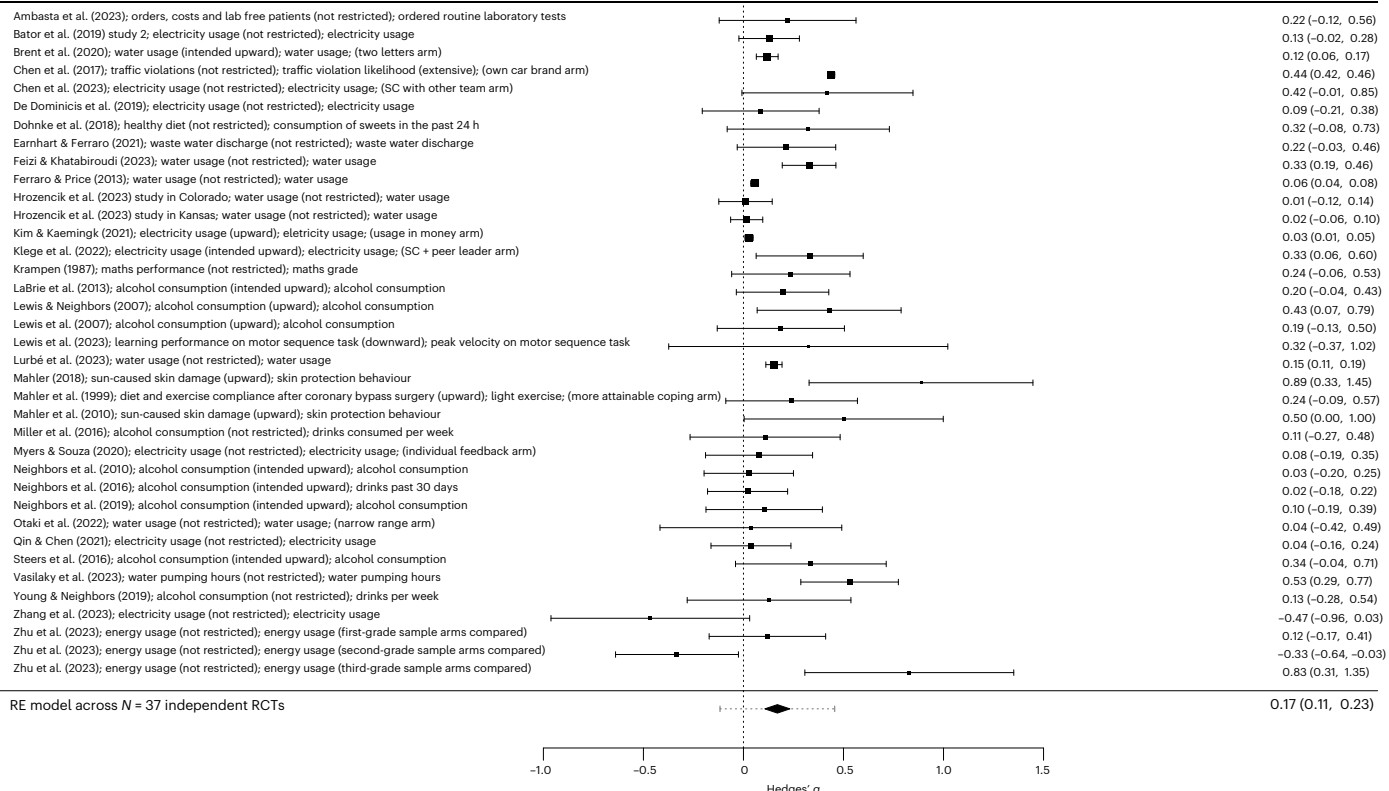

**Fig. 3 | Forest plot depicting the efficacy of SC-BCT relative to passive control conditions.** BC, behavioural change; RE model, random-effects model; SC, social comparison. For each study, the black square represents the effect size (standardized mean difference, Hedges' g), and the horizontal bars represent the 95% CI. The size of the black squares is proportional to the sample size of the given RCT. The diamond denotes the 95% CI of the pooled effect across the N = 37 independent RCTs, and the error bars of the diamond denote the corresponding 95% PI. Positive (negative) Hedges' g values indicate a higher (lower) efficacy regarding behaviour change (that is, a larger increase in desired behaviour and a larger decrease in undesired behaviour) in the SC-BCT arms than in the passive control condition arms.

behavioural outcomes matched the given social comparison dimension in SC-BCTs). The CI for the pooled effect excluded the null, signalling confidence in a significant behaviour change in the desired direction. As mentioned above, concern regarding a potential publication bias emerged from significant small study effects detected by Egger's test. Most studies targeted an increase in desired behaviour (k = 27), whereas 14 RCTs targeted a decrease in undesired behaviour. The highest number of studies investigated health behaviour (k = 14), followed

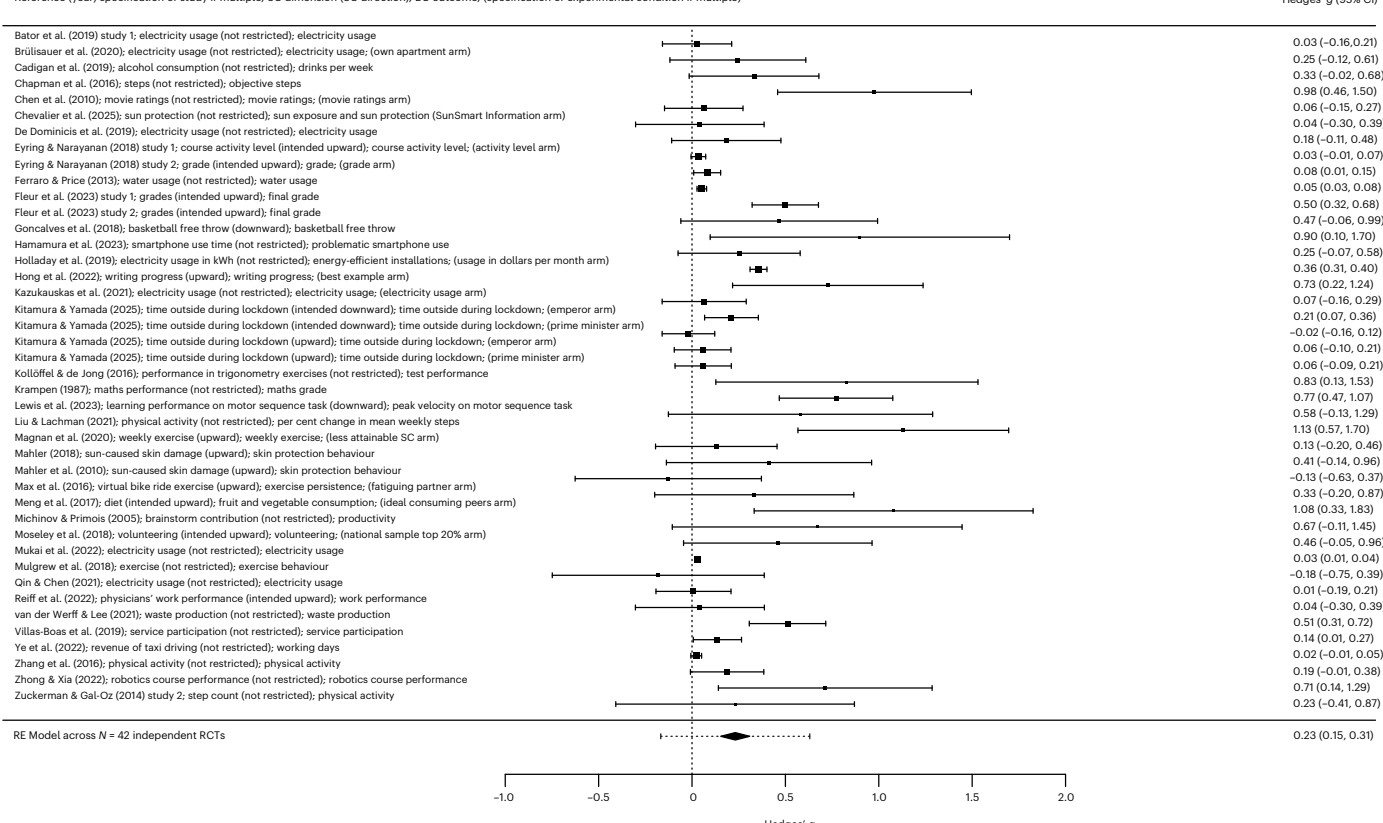

Reference (year) specification of study if multiple; SC dimension (SC direction); BC outcome; (specification of experimental condition if multiple)

Hedges' g (95% CI)

| Study | Hedges' g (95% CI) |
|---|---|
| Bator et al. (2019) study 1; electricity usage (not restricted); electricity usage | 0.03 (−0.16, 0.21) |
| Brülisauer et al. (2020); electricity usage (not restricted); electricity usage; (own apartment arm) | 0.25 (−0.12, 0.61) |
| Cadigan et al. (2019); alcohol consumption (not restricted); drinks per week | 0.33 (−0.02, 0.68) |
| Chapman et al. (2016); steps (not restricted); objective steps | 0.98 (0.46, 1.50) |
| Chen et al. (2010); movie ratings (not restricted); movie ratings; (movie ratings arm) | 0.06 (−0.15, 0.27) |
| Chevalier et al. (2025); sun protection (not restricted); sun exposure and sun protection (SunSmart Information arm) | 0.04 (−0.30, 0.39) |
| De Dominicis et al. (2019); electricity usage (not restricted); electricity usage | 0.18 (−0.11, 0.48) |
| Eyring & Narayanan (2018) study 1; course activity level (intended upward); course activity level; (activity level arm) | 0.03 (−0.01, 0.07) |
| Eyring & Narayanan (2018) study 2; grade (intended upward); grade; (grade arm) | 0.08 (0.01, 0.15) |
| Ferraro & Price (2013); water usage (not restricted); water usage | 0.05 (0.03, 0.08) |
| Fleur et al. (2023) study 1; grades (intended upward); final grade | 0.50 (0.32, 0.68) |
| Fleur et al. (2023) study 2; grades (intended upward); final grade | 0.47 (−0.06, 0.99) |
| Goncalves et al. (2018); basketball free throw (downward); basketball free throw | 0.90 (0.10, 1.70) |
| Hamamura et al. (2023); smartphone use time (not restricted); problematic smartphone use | 0.25 (−0.07, 0.58) |
| Holladay et al. (2019); electricity usage in kWh (not restricted); energy-efficient installations; (usage in dollars per month arm) | 0.36 (0.31, 0.40) |
| Hong et al. (2022); writing progress (upward); writing progress; (best example arm) | 0.73 (0.22, 1.24) |
| Kazukauskas et al. (2021); electricity usage (not restricted); electricity usage; (electricity usage arm) | 0.07 (−0.16, 0.29) |
| Kitamura & Yamada (2025); time outside during lockdown (intended downward); time outside during lockdown; (emperor arm) | 0.21 (0.07, 0.36) |
| Kitamura & Yamada (2025); time outside during lockdown (intended downward); time outside during lockdown; (prime minister arm) | −0.02 (−0.16, 0.12) |
| Kitamura & Yamada (2025); time outside during lockdown (upward); time outside during lockdown; (emperor arm) | 0.06 (−0.10, 0.21) |
| Kitamura & Yamada (2025); time outside during lockdown (upward); time outside during lockdown; (prime minister arm) | 0.06 (−0.09, 0.21) |
| Kollöffel & de Jong (2016); performance in trigonometry exercises (not restricted); test performance | 0.83 (0.13, 1.53) |
| Krampen (1987); maths performance (not restricted); maths grade | 0.77 (0.47, 1.07) |
| Lewis et al. (2023); learning performance on motor sequence task (downward); peak velocity on motor sequence task | 0.58 (−0.13, 1.29) |
| Liu & Lachman (2021); physical activity (not restricted); per cent change in mean weekly steps | 1.13 (0.57, 1.70) |
| Magnan et al. (2020); weekly exercise (upward); weekly exercise; (less attainable SC arm) | 0.13 (−0.20, 0.46) |
| Mahler (2018); sun-caused skin damage (upward); skin protection behaviour | 0.41 (−0.14, 0.96) |
| Mahler et al. (2010); sun-caused skin damage (upward); skin protection behaviour | −0.13 (−0.63, 0.37) |
| Max et al. (2016); virtual bike ride exercise (upward); exercise persistence; (fatiguing partner arm) | 0.33 (−0.20, 0.87) |
| Meng et al. (2017); diet (intended upward); fruit and vegetable consumption; (ideal consuming peers arm) | 1.08 (0.33, 1.83) |
| Michinov & Primois (2005); brainstorm contribution (not restricted); productivity | 0.67 (−0.11, 1.45) |
| Moseley et al. (2018); volunteering (intended upward); volunteering; (national sample top 20% arm) | 0.46 (−0.05, 0.96) |
| Mukai et al. (2022); electricity usage (not restricted); electricity usage | 0.03 (0.01, 0.04) |
| Mulgrew et al. (2018); exercise (not restricted); exercise behaviour | −0.18 (−0.75, 0.39) |
| Qin & Chen (2021); electricity usage (not restricted); electricity usage | 0.01 (−0.19, 0.21) |
| Reiff et al. (2022); physicians' work performance (intended upward); work performance | 0.04 (−0.30, 0.39) |
| van der Werff & Lee (2021); waste production (not restricted); waste production | 0.51 (0.31, 0.72) |
| Villas-Boas et al. (2019); service participation (not restricted); service participation | 0.14 (0.01, 0.27) |
| Ye et al. (2022); revenue of taxi driving (not restricted); working days | 0.02 (−0.01, 0.05) |
| Zhang et al. (2016); physical activity (not restricted); physical activity | 0.19 (−0.01, 0.38) |
| Zhong & Xia (2022); robotics course performance (not restricted); robotics course performance | 0.71 (0.14, 1.29) |
| Zuckerman & Gal-Oz (2014) study 2; step count (not restricted); physical activity | 0.23 (−0.41, 0.87) |
| RE Model across N = 42 independent RCTs | 0.23 (0.15, 0.31) |

−1.0   −0.5   0   0.5   1.0   1.5   2.0

Hedges' g

**Fig. 4 | Forest plot depicting the efficacy of SC-BCT relative to active control conditions.** For each study, the black square represents the effect size (standardized mean difference, Hedges' g), and the horizontal bars represent the 95% CI. The size of the black squares is proportional to the sample size of the given RCT. The diamond denotes the 95% CI of the pooled effect across the N = 42 independent RCTs, and the error bars of the diamond denote the corresponding 95% PI. Positive (negative) Hedges' g values indicate a higher (lower) efficacy regarding behaviour change (that is, a larger increase in desired behaviour and a larger decrease in undesired behaviour) in the SC-BCT arms than in the active control condition arms.

by performance behaviour (k = 12), climate change mitigation behaviour (k = 9) and service participation behaviour (k = 7).

In the long term, data availability was considerably thinner (k = 6). The results remained similar to the short-term results. SC-BCTs produced a significant (P < 0.05) small effect relative to active control conditions (g = 0.24; 95% CI, 0.03–0.45; k = 6; I² = 77%). Heterogeneity in outcomes was high and significant. No outliers were identified. Certainty of evidence was rated low due to concern about risk of bias and considerable significant unexplained heterogeneity between outcomes. Risk of bias emerged from insufficient reporting of the randomization process, deviations from the intended intervention (for example, participants or people delivering the intervention were likely to be aware of the assigned intervention), missing outcome data (for example, data were not available for all randomized participants) or lack of predefined analysis protocols in several studies. Data availability allowed for two sub-analyses (desired outcomes only and non-restricted SC-BCTs only), which revealed similar significant unexplained heterogeneity. No concern of indirectness emerged regarding population (that is, the majority of trials presented directness regarding population, whereas two trials presented probable indirectness regarding population due to investigating a rather specific sample without presenting a rationale for its selection). Similarly, no concern of indirectness emerged regarding intervention (that is, social comparison and SC-BCT definitions were always met), comparator (that is, active control conditions as a comparator regarding behaviour change) and outcome (that is, all behavioural outcomes matched the given social comparison dimension in SC-BCTs). The CI for the effect excluded the null, signalling confidence in a significant change in

behaviour in the desired direction. Egger's test was infeasible (k < 10). Most trials reporting follow-up data targeted desired behaviour (k = 4) and applied SC-BCTs not restricted in direction (k = 5).

**Moderator analyses.** Table 2 displays all moderator analysis results. In the overarching analysis comparing SC-BCTs to passive controls in the short term, no significant moderations were observed. In the analysis comparing SC-BCTs to active controls, a higher number of SC-BCT sessions (k = 33, β = 0.02, P = 0.007) and targeting a desired (versus undesired) behavioural outcome (k = 41, β = 0.19, P = 0.017) were both related to higher short-term efficacy of SC-BCTs. Unexplained heterogeneity in outcomes remained significant and high for all analyses, irrespective of the significance of moderation. No evidence was found for a significant difference in the efficacy of upward versus non-restricted SC-BCTs, nor for SC-BCTs targeting health versus climate change mitigation behaviours, nor for SC-BCTs targeting health versus performance behaviours.

**Sub-analyses.** Table 3 displays the results from the sub-analyses. In the short term and relative to passive control conditions, both SC-BCTs targeting undesired behaviour (k = 32) and SC-BCTs targeting desired behaviour (k = 5) yielded highly significant (all P < 0.001) small effects when analysed in isolation. When studies investigating climate change mitigation behaviour (k = 20), health behaviour (k = 13) and performance behaviour (k = 4) were analysed in isolation, all three produced highly significant (all P < 0.01) small effects. Most studies applied non-restricted SC-BCTs (k = 24) or upward SC-BCTs (k = 14), and when analysed in isolation both yielded highly significant (all P < 0.001)

**Table 2 | Moderators of the efficacy of SC-BCT**

| Post-intervention results: short-term efficacy | | | |
|---|---|---|---|
| **Overarching analysis: SC-BCTs versus passive control conditions** | | | |
| **Analysed continuous potential moderator** | **$k^a$** | **β** | **P** | **Rem. $I^2$ (%)** |
| Number of social comparison sessions | 36 | <0.00 | 0.584 | 94.44*** |
| **Analysed categorical potential moderator** | **$k^a$** | **β** | **P** | **Rem. $I^2$ (%)** |
| Desired behaviour (k=5) versus undesired behaviour (k=32) | NA (k<10 for desired behaviour) | | | |
| Upward (intended + clear) social comparison (k=13) versus non-restricted social comparison (k=23) | 37 | 0.03 | 0.645 | 92.92*** |
| Health behaviour (k=13) versus climate change mitigation behaviour (k=20) | 33 | −0.08 | 0.207 | 88.52*** |
| Health behaviour (k=13) versus performance behaviour (k=4) | NA (k<10 for performance behaviour) | | | |
| **Overarching analysis: SC-BCTs versus active control conditions** | | | |
| **Analysed continuous potential moderator** | **$k^a$** | **β** | **P** | **Rem. $I^2$ (%)** |
| Number of social comparison sessions | 33 | **0.02**** | 0.007 | 93.31*** |
| **Analysed categorical potential moderator** | **$k^a$** | **β** | **P** | **Rem. $I^2$ (%)** |
| Desired behaviour (k=27) versus undesired behaviour (k=14) | 41[b] | **0.19*** | 0.017 | 94.73*** |
| Upward (intended + clear) social comparison (k=14) versus non-restricted social comparison (k=27) | 41 | <−0.01 | 0.960 | 95.93*** |
| Health behaviour (k=13) versus climate change mitigation behaviour (k=9) | NA (k<10 for climate change mitigation behaviour) | | | |
| Health behaviour (k=13) versus performance behaviour (k=13) | 26 | 0.07 | 0.646 | 94.44*** |

β is the regression coefficient of the given moderator analysis; P is the two-sided P value indicating the level of statistical significance of the given moderator analysis; and rem. $I^2$ is the remaining variance in outcomes (that is, when the analysed moderator was accounted for), with asterisks indicating the statistical significance level of the corresponding Q statistic. Bold font indicates statistical significance of the given moderation. [a]Trials not reporting (sufficiently) on the given information (for example, the number of SC-BCT sessions) had to be excluded from the given moderator analysis. [b]Hamamura et al.[114] investigated a mix of desired behaviour (sun protection behaviour) and undesired behaviour (sun exposure); this study was therefore excluded from this analysis. *P<0.05; **P<0.01; ***P<0.001.

small effects. Too few trials investigated other directions (for example, downward SC-BCTs) for isolated review. About an even number applied clearly upward SC-BCTs (k = 7) versus intended upward SC-BCTs (k = 7; that is, upward for most but not all participants), with both yielding significant small effects when analysed in isolation. Studies investigating service participation behaviour were too few for isolated synthesis (k < 4). In the long term, data availability was scarce. When SC-BCTs targeting health (k = 5) or climate change mitigation behaviour (k = 6) were analysed in isolation, only SC-BCTs targeting climate change mitigation behaviour (but not health) produced a significant (P < 0.001) small-sized long-term effect. No other sub-analyses were feasible (k < 4).

In the short term and relative to active control conditions, both SC-BCTs targeting an increase in desired behaviour (k = 27) and SC-BCTs targeting a decrease in undesired behaviour (k = 14) yielded highly significant (all P < 0.01) small effects when analysed in isolation. One trial reported a primary outcome that mixed desired and undesired behaviour (that is, aggregated sun protection behaviours such as sunscreen use as well as sun exposure) and was consequently not included in the isolated syntheses[114]. When SC-BCTs targeting health behaviour (k = 14), performance behaviour (k = 12), climate change mitigation behaviour (k = 9) or service participation behaviour (k = 7) were analysed in isolation, all produced significant small effects. Most studies applied non-restricted SC-BCTs (k = 27), followed by upward SC-BCTs (k = 15) and then downward SC-BCTs (k = 6). Only the former two (but not downward SC-BCTs) produced significant (small) effects relative to active control conditions. Clearly upward (k = 8) and intended upward SC-BCTs (k = 7) both produced significant small effects. At follow-up, data availability was scarce (k = 6). There was insufficient evidence for isolated review of behavioural domains (k < 4). Only two isolated reviews were feasible, with a low number of included trials and limited power to detect effects. When SC-BCTs targeting desired behaviour were analysed in isolation, no evidence was found for a significant long-term effect relative to active control conditions (g = 0.34; 95% CI, −0.02 to 0.70; k = 4; $I^2$ = 82%). Similarly, non-restricted SC-BCTs analysed in isolation also yielded no significant long-term

effect when compared to active controls (g = 0.21; 95% CI, −0.01 to 0.43; k = 5; $I^2$ = 79%).

**Relative efficacy of different types of SC-BCTs.** Table 4 shows the results on the relative efficacy of different types of SC-BCTs. The number of trials was low (k = 9), limiting statistical power to detect significant differences. Across this thin evidence base, no evidence was found for significant differences in the short-term efficacy of upward SC-BCTs versus downward SC-BCTs, upward SC-BCTs versus non-restricted SC-BCTs, or SC-BCTs presenting more versus less attainable social comparison standards. In the long term, however, SC-BCTs presenting more (versus less) attainable social comparison standards were superior (P < 0.01), with a small pooled effect (g = 0.18; 95% CI, 0.05–0.31; k = 4; $I^2$ = 0%). See Supplementary Appendix E for a detailed description of certainty-of-evidence assessments for each of the abovementioned analyses. Certainty of evidence ranged from very low to moderate, depending on the analysis.

### Results for research question 2
**Efficacy of social comparison as an add-on BCT.** Table 5 shows the results for research question 2. The number of trials was low (k = 8), limiting statistical power to detect significant differences. In both the overarching analysis and the sub-analysis (that is, upward SC-BCTs only), differences in efficacy between BCT bundles with and without the SC-BCT were non-significant. Certainty of evidence of the overarching analysis was rated very low due to concern about risk of bias, considerable significant unexplained heterogeneity between outcomes and imprecision. See Supplementary Appendix E for a detailed description of ratings of indirectness using GRADE criteria.

### Discussion
We conducted a comprehensive meta-analysis covering data from RCTs on the efficacy of SC-BCTs across the behavioural sciences. In 79 RCTs, we found evidence supporting the efficacy of SC-BCTs in shaping behaviour in the desired direction, albeit with small magnitudes of pooled effects. We found evidence across types of control condition (that is,

**Table 3 | Efficacy of SC-BCT: sub-analyses**

| Comparison | k | g | P | 95% CI (95% PI) | I² (%) |
|---|---|---|---|---|---|
| **Post-intervention results: short-term efficacy** | | | | | |
| **Efficacy relative to passive control conditions** | | | | | |
| SC-BCTs versus passive control conditions—desired outcomes only | 5 | **0.36*** | <0.001 | **0.16 to 0.55 (0.12 to 0.60)** | 9.16 |
| SC-BCTs versus passive control conditions—undesired outcomes only | 32 | 0.15*** | <0.001 | 0.09 to 0.22 (−0.13 to 0.44) | 94.97*** |
| SC-BCTs versus passive control conditions—undesired outcomes only (outlier-adjusted) | 30 | 0.15*** | <0.001 | 0.09 to 0.22 (−0.11 to 0.42) | 94.62*** |
| Upward SC-BCTs (intended + clear) versus passive control conditions | 13 | 0.16*** | <0.001 | 0.08 to 0.25 (−0.05 to 0.38) | 72.35*** |
| Upward SC-BCTs (intended + clear) versus passive control conditions (trim-and-fill-adjusted) | 18 | 0.09 | 0.092 | −0.02 to 0.20 | 86.11*** |
| Upward SC-BCTs (intended + clear) versus passive control conditions (outlier-adjusted) | 12 | 0.13*** | <0.001 | 0.06 to 0.21 (−0.04 to 0.30) | 62.81** |
| Clearly upward SC-BCTs versus passive control conditions | 6 | 0.30** | 0.009 | 0.08 to 0.52 (−0.18 to 0.76) | 70.82** |
| Intended upward SC-BCTs versus passive control conditions | 7 | **0.12*** | <0.001 | **0.07 to 0.17 (0.07 to 0.17)** | 0.00 |
| Intended upward SC-BCTs versus passive control conditions (outlier-adjusted) | 6 | **0.12*** | <0.001 | **0.07 to 0.17 (0.07 to 0.17)** | 0.05 |
| Non-restricted SC-BCTs versus passive control conditions | 24 | 0.15*** | <0.001 | 0.07 to 0.23 (−0.17 to 0.48) | 95.56*** |
| Non-restricted SC-BCTs versus passive control conditions (outlier-adjusted) | 23 | 0.14*** | <0.001 | 0.06 to 0.22 (−0.17 to 0.45) | 95.30*** |
| **Efficacy relative to active control conditions** | | | | | |
| SC-BCTs versus active control conditions—desired outcomes only | 27[a] | 0.35*** | <0.001 | 0.22 to 0.48 (−0.19 to 0.89) | 95.43*** |
| SC-BCTs versus active control conditions—desired outcomes only (trim-and-fill-adjusted) | 36 | 0.19* | 0.018 | 0.03 to 0.34 | 97.27*** |
| SC-BCTs versus active control conditions—undesired outcomes only | 14[a] | 0.11** | 0.003 | 0.04 to 0.18 (−0.10 to 0.33) | 90.00*** |
| SC-BCTs versus active control conditions—undesired outcomes only (trim-and-fill-adjusted) | 19 | 0.04 | 0.371 | −0.05 to 0.14 | 95.08*** |
| Upward SC-BCTs (intended + clear) versus active control conditions (all of which happened to investigate desired outcomes) | 14 | 0.21** | 0.001 | 0.08 to 0.34 (−0.17 to 0.59) | 84.16*** |
| Upward SC-BCTs (intended + clear) versus active control conditions (trim-and-fill-adjusted) | 20 | 0.06 | 0.506 | −0.11 to 0.23 | 93.24*** |
| Upward SC-BCTs (intended + clear) versus active control conditions (outlier-adjusted) | 13 | 0.19** | 0.003 | 0.06 to 0.31 (−0.15 to 0.52) | 81.85*** |
| Upward SC-BCTs (intended + clear) versus active control conditions—desired outcomes only | 12 | 0.27** | 0.001 | 0.11 to 0.43 (−0.17 to 0.71) | 86.31*** |
| Upward SC-BCTs (intended + clear) versus active control conditions—desired outcomes only (trim-and-fill-adjusted) | 17 | 0.08 | 0.451 | −0.13 to 0.29 | 94.38*** |
| Clearly upward SC-BCTs versus active control conditions | 7 | **0.10*** | 0.039 | **0.01 to 0.19 (0.01 to 0.19)** | 0.02 |
| Clearly upward SC-BCTs versus active control conditions (outlier-adjusted) | 6 | 0.08 | 0.115 | −0.02 to 0.17 (−0.02 to 0.17) | 0.00 |
| Clearly upward SC-BCTs versus active control conditions—desired outcomes only | 5 | 0.27* | 0.048 | 0.00 to 0.54 (−0.19 to 0.73) | 38.42 |
| Intended upward SC-BCTs versus active control conditions | 7 | 0.28* | 0.012 | 0.06 to 0.49 (−0.24 to 0.79) | 93.14*** |
| Downward SC-BCTs (intended + clear) versus active control conditions | 7 | 0.37 | 0.230 | −0.24 to 0.98 (−1.26 to 2.00) | 96.28*** |
| Downward SC-BCTs (intended + clear) versus active control conditions—desired outcomes only | 5 | 0.51 | 0.261 | −0.38 to 1.39 (−1.56 to 2.57) | 89.95*** |
| Non-restricted SC-BCTs versus active control conditions | 27 | 0.22*** | <0.001 | 0.13 to 0.31 (−0.19 to 0.64) | 97.32*** |
| Non-restricted SC-BCTs versus active control conditions (outlier-adjusted) | 26 | 0.20*** | <0.001 | 0.11 to 0.28 (−0.15 to 0.54) | 96.56*** |
| Non-restricted SC-BCTs versus active control conditions—desired outcomes only | 16 | 0.32*** | <0.001 | 0.15 to 0.49 (−0.27 to 0.91) | 97.57*** |
| Non-restricted SC-BCTs versus active control conditions—desired outcomes only (trim-and-fill-adjusted) | 19 | 0.23* | 0.027 | 0.03 to 0.43 | 98.39*** |

**Table 3 (continued) | Efficacy of SC-BCT: sub-analyses**

| Comparison | k | g | P | 95% CI (95% PI) | I² (%) |
|---|---|---|---|---|---|
| Non-restricted SC-BCTs versus active control conditions—undesired outcomes only | 10 | 0.14* | 0.010 | 0.03 to 0.25 (−0.14 to 0.42) | 95.15*** |
| **Health-related outcomes only** | | | | | |
| **Efficacy relative to passive control conditions** | | | | | |
| SC-BCTs versus passive control conditions | 13 | **0.19** | <0.001 | **0.09 to 0.28 (0.03 to 0.35)** | 14.45 |
| SC-BCTs versus passive control conditions (trim-and-fill-adjusted) | 17 | 0.13* | 0.018 | 0.02 to 0.23 | 37.68* |
| SC-BCTs versus passive control conditions (outlier-adjusted) | 12 | **0.16*** | <0.001 | **0.07 to 0.25 (0.07 to 0.25)** | 0.00 |
| SC-BCTs versus passive control conditions—undesired outcomes only | 10 | **0.14** | 0.002 | **0.05 to 0.23 (0.05 to 0.23)** | 0.00 |
| Upward SC-BCTs (intended + clear) versus passive control conditions | 10 | 0.20*** | <0.001 | 0.09 to 0.32 (−0.03 to 0.43) | 30.91 |
| Upward SC-BCTs (intended + clear) versus passive control conditions (trim-and-fill-adjusted) | 14 | 0.11 | 0.121 | −0.03 to 0.26 | 61.40** |
| Upward SC-BCTs (intended + clear) versus passive control conditions (outlier-adjusted) | 9 | **0.16*** | 0.001 | **0.06 to 0.26 (0.03 to 0.29)** | 7.17 |
| Upward SC-BCTs (intended + clear) versus passive control conditions—undesired outcomes only | 7 | **0.13** | 0.008 | **0.03 to 0.23 (0.03 to 0.23)** | 0.09 |
| Clearly upward SC-BCTs versus passive control conditions | 5 | **0.37*** | <0.001 | **0.18 to 0.56 (0.12 to 0.61)** | 13.88 |
| Intended upward SC-BCTs versus passive control conditions | 5 | 0.10 | 0.073 | −0.01 to 0.21 (−0.01 to 0.21) | 0.00 |
| **Efficacy relative to active control conditions** | | | | | |
| SC-BCTs versus active control conditions | 13 | 0.32** | 0.001 | 0.13 to 0.52 (−0.26 to 0.91) | 64.79** |
| SC-BCTs versus active control conditions—desired outcomes only | 10 | 0.38** | 0.007 | 0.10 to 0.66 (−0.39 to 1.16) | 72.87** |
| Upward SC-BCTs (intended + clear) versus active control conditions | 5 | 0.29 | 0.071 | −0.02 to 0.59 (−0.27 to 0.84) | 46.64 |
| Clearly upward SC-BCTs versus active control conditions | 4 | 0.16 | 0.157 | −0.06 to 0.38 (−0.06 to 0.38) | 0.00 |
| Non-restricted SC-BCTs versus active control conditions | 8 | 0.35* | 0.013 | 0.07 to 0.62 (−0.35 to 1.05) | 74.80** |
| Non-restricted SC-BCTs versus active control conditions—desired outcomes only | 5 | 0.46 | 0.059 | −0.02 to 0.94 (−0.60 to 1.52) | 81.47*** |
| **Climate change mitigation behaviour outcomes only** | | | | | |
| **Efficacy relative to passive control conditions** | | | | | |
| SC-BCTs versus passive control conditions | 20 | 0.12** | 0.001 | 0.05 to 0.20 (−0.15 to 0.40) | 93.98*** |
| SC-BCTs versus passive control conditions (outlier-adjusted) | 18 | 0.12*** | <0.001 | 0.05 to 0.19 (−0.11 to 0.35) | 92.45*** |
| Non-restricted SC-BCTs versus passive control conditions | 18 | 0.12** | 0.007 | 0.03 to 0.20 (−0.18 to 0.41) | 90.94*** |
| Non-restricted SC-BCTs versus passive control conditions (outlier-adjusted) | 17 | 0.10** | 0.009 | 0.03 to 0.18 (−0.16 to 0.36) | 89.10*** |
| **Efficacy relative to active control conditions** | | | | | |
| SC-BCTs versus active control conditions (all of which happened to investigate non-restricted social comparison) | 9 | 0.16* | 0.011 | 0.04 to 0.28 (−0.18 to 0.49) | 97.58*** |
| SC-BCTs versus active control conditions—undesired outcomes only | 8 | 0.12* | 0.045 | 0.00 to 0.24 (−0.17 to 0.41) | 96.33*** |
| **Performance outcomes only** | | | | | |
| **Efficacy relative to passive control conditions** | | | | | |
| SC-BCTs versus passive control conditions | 4 | **0.37*** | <0.001 | **0.22 to 0.52 (0.13 to 0.61)** | 33.94 |
| **Efficacy relative to active control conditions** | | | | | |
| SC-BCTs versus active control conditions | 13 | 0.39*** | <0.001 | 0.20 to 0.59 (−0.21 to 1.00) | 97.06*** |
| SC-BCTs versus active control conditions (trim-and-fill-adjusted) | 19 | 0.19 | 0.073 | −0.02 to 0.40 | 97.59*** |

**Table 3 (continued) | Efficacy of SC-BCT: sub-analyses**

| Comparison | *k* | *g* | *P* | 95% CI (95% PI) | *I²* (%) |
|---|---|---|---|---|---|
| Upward SC-BCTs (intended + clear) versus active control conditions | 6 | 0.25* | 0.024 | 0.03 to 0.47 (−0.25 to 0.76) | 93.92*** |
| Intended upward SC-BCTs versus active control conditions | 5 | 0.19 | 0.065 | −0.01 to 0.40 (−0.25 to 0.64) | 93.52*** |
| Non-restricted SC-BCTs versus active control conditions | 7 | 0.36* | 0.013 | 0.08 to 0.64 (−0.35 to 1.06) | 98.76*** |
| **Service outcomes only** | | | | | |
| **Efficacy relative to active control conditions** | | | | | |
| SC-BCTs versus active control conditions | 7 | 0.09** | 0.007 | 0.03 to 0.16 (−0.01 to 0.20) | 20.09 |
| **Follow-up results: long-term efficacy** | | | | | |
| **Efficacy relative to passive control conditions** | | | | | |
| SC-BCTs versus passive control conditions—undesired outcomes only | 11 | **0.10***** | <0.001 | **0.06 to 0.13 (0.06 to 0.13)** | 0.05 |
| SC-BCTs versus passive control conditions—undesired outcomes only (outlier-adjusted) | 10 | **0.08***** | <0.001 | **0.04 to 0.13 (0.03 to 0.14)** | 4.36 |
| Upward SC-BCTs (intended + clear) versus passive control conditions | 5 | **0.10***** | <0.001 | **0.06 to 0.14 (0.06 to 0.14)** | 0.00 |
| Upward SC-BCTs (intended + clear) versus passive control conditions (outlier-adjusted) | 4 | **0.10***** | <0.001 | **0.07 to 0.14 (0.07 to 0.14)** | 0.00 |
| Upward SC-BCTs (intended + clear) versus passive control conditions—undesired outcomes only | 4 | **0.09***** | <0.001 | **0.04 to 0.15 (0.02 to 0.16)** | 5.79 |
| Non-restricted SC-BCTs versus passive control conditions | 8 | **0.07***** | <0.001 | **0.03 to 0.10 (0.03 to 0.10)** | 0.07 |
| Non-restricted SC-BCTs versus passive control conditions (outlier-adjusted) | 7 | **0.06***** | <0.001 | **0.02 to 0.10 (0.02 to 0.10)** | 0.00 |
| Non-restricted SC-BCTs versus passive control conditions—undesired outcomes only | 7 | **0.06***** | <0.001 | **0.03 to 0.10 (0.03 to 0.10)** | 0.01 |
| Non-restricted SC-BCTs versus passive control conditions—undesired outcomes only (outlier-adjusted) | 6 | **0.06**** | 0.001 | **0.02 to 0.10 (0.02 to 0.10)** | 0.00 |
| **Efficacy relative to active control conditions** | | | | | |
| SC-BCTs versus active control conditions | 6 | 0.24* | 0.027 | 0.03 to 0.45 (−0.20 to 0.76) | 76.54** |
| SC-BCTs versus active control conditions—desired outcomes only | 4 | 0.34 | 0.063 | −0.02 to 0.70 (−0.37 to 1.05) | 81.94** |
| Non-restricted SC-BCTs versus active control conditions | 5 | 0.21 | 0.060 | −0.01 to 0.43 (−0.25 to 0.67) | 79.34** |
| **Health outcomes only** | | | | | |
| **Efficacy relative to passive control conditions** | | | | | |
| SC-BCTs versus passive control conditions | 5 | 0.02 | 0.679 | −0.09 to 0.13 (−0.09 to 0.13) | 0.00 |
| SC-BCTs versus passive control conditions—undesired outcomes only | 4 | <−0.00 | 0.994 | −0.12 to 0.11 (−0.12 to 0.11) | 0.00 |
| **Climate change mitigation behaviour outcomes only** | | | | | |
| **Efficacy relative to passive control conditions** | | | | | |
| SC-BCTs versus passive control conditions (all of which happened to investigate non-restricted social comparison) | 6 | **0.10***** | <0.001 | **0.07 to 0.14 (0.07 to 0.14)** | 0.01 |
| SC-BCTs versus passive control conditions (outlier-adjusted) | 5 | **0.10***** | <0.001 | **0.06 to 0.14 (0.06 to 0.14)** | 0.00 |
| Non-restricted SC-BCTs versus passive control conditions | 6 | **0.06***** | <0.001 | **0.03 to 0.10 (0.03 to 0.10)** | 0.03 |
| Non-restricted SC-BCTs versus passive control conditions (outlier adjusted) | 5 | **0.06**** | 0.001 | **0.02 to 0.10 (0.02 to 0.10)** | 0.00 |

*P* is the two-sided *P* value indicating the statistical significance level of the pooled effect (Hedges' *g*). Bold font indicates that both the 95% CI and the 95% PI excluded the null, highlighting particular statistical certainty in the significance of the difference in behaviour change observed between the respective comparison groups. The values are standardized mean differences (that is, Hedges' *g*) derived from the random-effects meta-analysis. Positive values of *g* indicate that the social comparison conditions (SC-BCTs) showed a higher pooled behaviour change (that is, a higher increase for desired behaviour and a higher decrease for undesired behaviour) than passive or active control conditions and vice versa. Please note that the trim-and-fill method (adjustment for funnel plot asymmetry) does not supply PIs. [a]Hamamura et al.[114] investigated of mix of desired behaviour (sun protection behaviour) and undesired behaviour (sun exposure); this study was thus excluded from the sub-analyses concerning desired and undesired behaviour only. *P<0.05; **P<0.01; ***P<0.001.

## Table 4 | Efficacy of varying SC-BCTs directly compared in trials

| Comparison | k (N) | g | P | 95% CI (95% PI) | I² | CoE |
|---|---|---|---|---|---|---|
| **Post-intervention results: short-term efficacy** | | | | | | |
| **Upward social comparison versus downward social comparison** | | | | | | |
| Upward SC-BCTs versus downward SC-BCTs | 5 (321) | −0.19 | 0.554 | −0.80 to 0.43 (−1.59 to 1.21) | 85.56*** | + |
| **Upward social comparison versus non-restricted social comparison** | | | | | | |
| Upward SC-BCTs versus non-restricted SC-BCTs | 7 (19,363) | **−0.03*** | 0.038 | **−0.06 to −0.00 (−0.06 to −0.00)** | 0.04 | +++ |
| Upward SC-BCTs versus non-restricted SC-BCTs (outlier-adjusted) | 4 (18,992) | −0.03 | 0.065 | −0.06 to 0.00 (−0.06 to 0.00) | 0.00 | NA |
| Upward SC-BCTs versus non-restricted SC-BCTs—desired outcomes only | 6 (13,784) | −0.03 | 0.059 | −0.07 to 0.00 (−0.07 to 0.00) | 0.01 | NA |
| **More attainable versus less attainable upward social comparison** | | | | | | |
| More attainable upward SC-BCTs versus less attainable upward SC-BCTs | 9 (1,100) | 0.03 | 0.703 | −0.12 to 0.18 (−0.27 to 0.33) | 33.12 | ++ |
| More attainable upward SC-BCTs versus less attainable upward SC-BCTs—desired outcomes only | 6 (529) | 0.04 | 0.760 | −0.20 to 0.28 (−0.42 to 0.49) | 45.36 | NA |
| **Follow-up results: long-term efficacy** | | | | | | |
| **More attainable versus less attainable upward social comparison** | | | | | | |
| More attainable upward SC-BCTs versus less attainable upward SC-BCTs | 4 (22,022) | **0.18**** | 0.008 | **0.05 to 0.31 (0.05 to 0.31)** | 0.02 | +++ |

CoE as assessed with GRADE criteria: +++, moderate certainty; ++, low certainty; +, very low certainty. P is the two-sided P value indicating the statistical significance level of the pooled effect (Hedges' g). Bold font indicates that both the 95% CI and the 95% PI excluded the null, highlighting particular statistical certainty in the significance of the difference in behaviour change observed between the respective comparison groups. The values are standardized mean differences (that is, Hedges' g) derived from the random-effects meta-analysis. Positive values of g indicate that the first-mentioned social comparison conditions showed a higher pooled behaviour change (that is, a higher increase for desired behaviour and a higher decrease for undesired behaviour) than the second-mentioned social comparison conditions and vice versa. *P < 0.05; **P < 0.01; ***P < 0.001.

## Table 5 | Efficacy of social comparison as an add-on BCT

| Comparison | k (N) | g | P | 95% CI (95% PI) | I² | CoE |
|---|---|---|---|---|---|---|
| **Post-intervention results: short-term efficacy** | | | | | | |
| BCT bundle with SC-BCTs versus BCT bundle without SC-BCTs | 8 (327,783) | 0.22 | 0.076 | −0.02 to 0.47 (−0.43 to 0.87) | 96.01*** | + |
| BCT bundle with upward SC-BCTs (intended + clear) versus BCT bundle without SC-BCTs | 5 (324,585) | 0.28 | 0.130 | −0.08 to 0.64 (−0.53 to 1.09) | 88.46*** | NA |

CoE as assessed with GRADE criteria: +, very low certainty. P is the two-sided P value indicating the statistical significance level of the pooled effect (Hedges' g). The values are standardized mean differences (that is, Hedges' g) derived from the random-effects meta-analysis. Positive values of g indicate that the BCT bundle including the social comparison conditions showed a higher pooled behaviour change (that is, a higher increase for desired behaviour and a higher decrease for undesired behaviour) than the BCT bundle excluding the social comparison conditions and vice versa. ***P < 0.001.

passive controls and active controls), behavioural domains (that is, health behaviour, climate change mitigation behaviour, performance behaviour and service participation behaviour), SC-BCTs focusing on increasing desired behaviours, SC-BCTs focusing on decreasing undesired behaviours and assessment timeline (that is, short- and long-term assessments). However, certainty of evidence was often limited, mainly due to concern about risk of bias and considerable unexplained heterogeneity. Further high-quality research is needed to thoroughly examine the robustness of the current findings. Notably, the vast majority of studies did not investigate SC-BCTs as a stand-alone intervention, but rather as the primary BCT accompanied by another BCT, such as the provision of repeated intra-individual feedback enabling temporal comparison in addition to social comparison.

Our results align with prior findings in related fields. Other meta-analyses on the efficacy of SC-BCTs also revealed significant results on behaviour change, including in climate change mitigation behaviour[14,24,27,30] and work performance[31]. In one review, for instance, SC-BCTs were identified as one of the two most effective BCTs for changing climate change mitigation behaviour, alongside financial approaches[24].

We extended prior work by performing various moderator analyses. In the overarching analysis comparing SC-BCTs to active controls, the number of SC-BCT sessions was positively associated with efficacy. This suggests a dose–response relationship, which might be attributable to cumulative reinforcement, habit formation, cognitive shifts or emotional shifts, fostering sustained engagement and internalization

of change. However, in the overarching analysis comparing SC-BCTs to passive controls, the number of SC-BCT sessions did not moderate efficacy. More research is needed to determine when and under what circumstances a dose–response relationship can be assumed. Similarly, the significant negative moderation of study quality in the overarching analysis comparing SC-BCTs to active (but not passive) control conditions suggests that effects are overestimated. Pooled effects are biased by lower-quality evidence finding larger effects. Future research needs to improve methodological quality to ensure accurate estimation of effects. As more trials accumulate over time, future meta-analytic research should re-evaluate the present finding, especially as the present work found this association in only one analysis (but not the other). In studies comparing SC-BCTs to active control conditions (but not in those comparing them to passive control conditions), SC-BCTs targeting desired behaviours were associated with higher efficacy in behaviour change than SC-BCTs targeting undesired behaviours. This suggests that SC-BCTs might be more effective when they target a desired behaviour (for example, increasing healthy food intake) rather than an undesired behaviour (for example, reducing unhealthy food intake). However, given mixed findings across the moderator analyses relative to passive and active controls, this association remains preliminary and requires further testing. Lastly, trials directly comparing upward SC-BCTs portraying a more (versus less) attainable social comparison standard were associated with higher long-term behaviour change, whereas behaviour change in the short term did not differ significantly. This suggests that more (versus less) attainable

upward SC-BCTs might yield longer-lasting behaviour change. Given that the number of trials was rather low, this association should also be interpreted with caution and re-examined in future research.

Overall, the small effect sizes found for SC-BCTs need to be interpreted in the context of low cost in developing and disseminating SC-BCTs, making them realistic options for large-scale implementation (for example, for preventive health interventions). For instance, in various studies only one or two letters or emails with social comparison information (for example, personal energy usage versus that of a social standard) were sent to thousands of participants, or scalable low-cost digital health interventions using peer comparison or leaderboards were applied. Small effects according to statistical benchmarks[121] might therefore have large real-life impacts when costs are low and scalability is large.

There are limitations to our meta-analysis. First, we aimed at only including research that explicitly referred to social comparison or social-comparison-related terms (for example, upward/downward/ lateral comparison). This choice was made in an effort to maximize internal validity. Yet, this may have resulted in missing related literature. For example, research on audit and feedback to influence health professional behaviour often involves interpersonal comparison (for example, comparison with median performance) as one part of a BCT bundle[122]. Likewise, any group-based intervention is likely to be influenced by social comparison processes. Yet, studies not explicitly referencing social comparison (recall that we performed all-fields searches) were deemed unlikely to feature a study design capable of isolating the individual impact of SC-BCTs on behaviour change. Despite the restriction on research explicitly referring to social comparison terms, the present work covered a broad range of literature (79 RCTs) and a very large number of included participants ($N = 1,356,521$). Second, data were scarce for some analyses (for example, RCTs directly comparing different SC-BCTs), limiting statistical power. As more research accumulates, more (fine-grained) meta-analytic analyses will become feasible. Third, the generalizability of our results is limited, as most of the included trials were conducted in the USA and other high-income countries, necessitating more research from other contexts. Fourth, while we adjusted for the potential impact of publication bias on meta-analytic synthesis (that is, Egger's test and the trim-and-fill method), we excluded non-peer-reviewed data. While maximizing internal validity through synthesizing only quality-controlled data, this decision may have diminished the external validity of the results, and the results might be biased by publication bias (that is, underreporting of null results). Some included RCTs did report on null results. Yet, it remains unknown to what extent the present results are affected by publication bias. Fifth, the results concern only collective behaviour (that is, group mean differences). Future qualitative and quantitative research is necessary to investigate individual processes evoked by SC-BCT (for example, investigating how many participants do versus do not change behaviour and for which reasons), which may help in tailoring and optimizing SC-BCTs.

On the basis of the present literature, SC-BCTs appear to have the potential to shape adaptive behaviour change concerning climate change mitigation, health, performance, and service participation. Small effect sizes need to be interpreted in terms of low cost and large scalability. Generalization of the results is limited given that most of the available data are from the USA or other high-income countries. More research in diverse contexts is needed to further investigate the generalizability of the results. Certainty of evidence was constrained by various sources of bias in the current literature, highlighting the need for more high-quality research to more robustly examine the efficacy of SC-BCTs in future research and meta-analytic syntheses.

## Methods

### Preregistration and guidelines
This work was preregistered with the PROSPERO database (CRD42022343154) and followed PRISMA 2020 guidelines[123]. One

deviation from our preregistration should be noted. In line with a peer reviewer's recommendation, we included only behavioural outcomes (for example, electricity usage) in the meta-analysis and excluded studies exclusively investigating cognitive (for example, intentions) and/or affective outcomes (for example, feelings). The systematic literature search, data extractions and statistical analyses were conducted by at least two authors independently (T.H.H., R.M.C., J.N. and F.L.). In cases of disagreement, consensus was reached among T.H.H., R.M.C. and N.M. in personal discussions.

### Definitions of social comparison and SC-BCTs
We primarily adhered to Wood's[3] definition of social comparison as thinking about social information in relation to the self. Furthermore, we followed the general comparative-processing model[5] to conceptualize social comparison as a process encompassing the selection of the social comparison standard, the basic comparison process itself (that is, evaluation of similarity or discrepancy between the target and the social standard) and the resulting reactions. Lastly, we followed the definition and taxonomy of BCTs provided by Michie et al.[124]. Accordingly, BCTs are defined as observable, replicable and irreducible interventions or components of a more complex intervention designed to have a causal influence on behaviour change[124]. Michie et al. defined SC-BCTs as interventions in which attention is drawn to the performance of others to enable social comparison on a particular dimension. Nonetheless, our decision on whether a study used social comparison as an intervention was primarily based on Wood's definition of social comparison. For this comparison to occur, sufficient information needs to be provided, particularly for covert behaviour. For instance, an intervention on reducing energy usage needs to provide quantitative feedback about one's own energy usage and the usage of a given social standard. Accordingly, descriptive norm interventions that provide such quantitative feedback belong to the category of SC-BCTs, whereas injunctive norm interventions (that is, interventions providing information on the degree to which a particular behaviour is socially approved or disapproved[31,125]) do not belong to SC-BCTs. The efficacy of SC-BCTs may be accompanied by other potentially contributing factors, even when trialists aim to explicitly and solely focus on the efficacy of SC-BCTs. For instance, if an intervention provides quantitative feedback on an individual's performance (alongside the social standard) multiple times, it enables not only social comparison but also temporal comparison (that is, intra-individual comparison with past performance). Similarly, an intervention may include an SC-BCT accompanied by a frowning face or a downward-pointing thumb, introducing additional potential influences on behaviour change processes. As a result, other factors could potentially interfere with the efficacy of SC-BCTs, preventing them from functioning as stand-alone interventions. Regarding our first research question on the efficacy of SC-BCTs, we focused on trials examining SC-BCTs either as the stand-alone BCT or as the primary BCT. Regarding our second research question ('How effective are SC-BCTs when added to a BCT bundle?'), we focused on studies investigating SC as an add-on component. Such trials compare the efficacy of two intervention arms, both using BCT bundles, where the only difference between the bundles is the inclusion of the SC-BCT component (that is, the BCT bundle including SC-BCTs versus the BCT bundle excluding only the SC-BCT component). Add-on studies thus examine whether adding social comparison to a bundle of BCTs adds incremental value in behaviour change (that is, relative to the BCT bundle without SC-BCT). Categorizations of BCTs were based on Michie et al.'s taxonomy and conducted independently by two of the authors (T.H.H. and R.M.C.). Discrepancies were discussed among three authors (T.H.H., R.M.C. and N.M.) until consensus was reached.

### Sub-categorization of SC-BCTs by comparison direction
In all three sub-categories, SC-BCTs were further sub-categorized according to the applied social comparison direction of the SC-BCTs.

In some SC-BCTs, a group average (for example, the average energy usage across a given residential area) or multiple social standards with varying degrees of the desired outcome are provided. These examples represent a non-restricted comparison direction, as the perceived comparison direction may vary between participants[5]. In other SC-BCTs, a group average for a selected sub-population is provided. For instance, providing feedback on the energy usage of the top 20% energy-efficient neighbours represents an upward social standard for most, but not all, study participants. The present work refers to the direction of such SC-BCTs as intended upward (and vice versa for intended downward).

### Categorization of control conditions

Control groups were categorized into either passive control conditions (defined as assessment-only conditions without any form of manipulation) or active control conditions (defined as conditions that received any kind of non-social-comparison-related BCT or unspecific control task).

### Systematic literature search

We systematically searched MEDLINE, PsycINFO and Web of Science with the preregistered search strategy. In line with the systematic search by Gerber et al.[12], who chose to conduct a broad search by using only the term 'social comparison', our equally broad search used the category 'all-fields' and included only terms related to social comparison (TX 'social compar*' OR TX upward comparison* OR TX downward comparison* OR TX 'lateral comparison*'). The search was conducted on 2 January 2024 and was not time-restricted (that is, involving all electronic hits from inception to the search date). No restrictions were made concerning scientific disciplines. Yet, all data turned out to be from the behavioural sciences. Also, no restrictions were made concerning sample characteristics. Languages of publications were limited to English, Dutch and German, and searches were carried out with English search terms only. The full search strategy is shown in Supplementary Appendix A.

### Inclusion criteria

Studies had to meet all of the following inclusion criteria to be included in the present work: (1) the study was an RCT, (2) at least one arm investigated the efficacy of SC (as defined by Wood[3]) as a BCT (as defined by Michie et al.[124]) and used either a stand-alone or primary SC-BCT (research question 1) or an add-on SC-BCT (research question 2), (3) data for at least one behavioural outcome (for example, electricity usage) were reported, (4) the outcome was assessed at least 24 hours after the induction of the (first) SC-BCT session to exclude experimental studies assessing only immediate reactions to social comparison, (5) the available outcome data included at least ten participants per arm[126] to exclude potential chance findings, and (6) the data were peer-reviewed. Withdrawn (that is, retracted) publications were not included.

### Assessment intervals: short-term and long-term

We divided data assessment points or intervals into two categories: short-term and long-term outcome assessment. As we included data from diverse behavioural sciences covering a broad range of studied behaviours, we had to rely on primary study author-defined assessment time points or intervals. While some behaviours were assessed over long and continuous intervals (for example, energy usage), other behaviours were assessed at a given time point (for example, basketball free throw ability). Predefined cut-offs for categorizing short-term versus long-term data were therefore not meaningful. For the analyses on short-term efficacy, we extracted the first (that is, shortest) outcome assessment that met inclusion criterion number four (≥24 h after the (first) SC-BCT session). If a trial reported data for only one assessment time point or interval, this assessment was automatically extracted for the short-term category. For trials reporting data for two or more assessments, the first assessment (≥24 h after the (first) SC-BCT session) was extracted for the short-term category, and the last assessment was extracted for the long-term category.

### Data prioritization: primary outcome and comparison

For complex trials investigating multiple outcomes and/or multiple arms, we applied a data prioritization algorithm to avoid data dependencies in a given meta-analysis. When applicable, data were prioritized as follows: (1) when multiple outcomes were reported, but only one of them aligned with the social comparison dimension, this outcome was prioritized; (2) when both objective (for example, objective step count) and subjective outcomes (for example, subjective step count) were reported, objective outcomes were prioritized; and (3) when a trial involved arms with different kinds of SC-BCTs, the primary SC-BCT arm (reported as such by the primary authors of the given trial) was prioritized. Data excluded from one analysis could still be included in other (sub-)analyses as long as data independence was met.

### Primary outcome

The primary outcome metric was the standardized mean difference (Hedges' $g$) for the primary behaviour outcome (see above) between the SC-BCT arm and the comparator arm at a given assessment time point (see above). Following Cohen's conventions[121], effect sizes were interpreted as small (0.2), moderate (0.5) or large (0.8). In the present work, positive values of $g$ indicate behaviour change in the intended direction (that is, higher increases in desired outcomes and higher decreases in undesired outcomes), which allowed for meaningful pooling of effects across desired and undesired outcomes.

### Desired versus undesired behaviour

SC-BCT studies may target an increase in desired behaviour (for example, steps per day or course grades) or a decrease in undesired behaviour (for example, alcohol consumption or water usage). We sub-classified studies accordingly and analysed data both across all classes and in isolation.

### Risk-of-bias assessment

Risk of bias for each (independent) RCT was assessed using the Cochrane risk of bias tool v.2.0 (ref. [127]), which assesses the methodological rigour of RCTs on five domains. It assesses bias arising from the randomization process (D1), deviations from the intended intervention (D2), missing data (D3), outcome assessment (D4) and selection of the reported result(s) (D5). On the basis of the Cochrane algorithm, an overall ordinal rating was derived for each trial indicating low risk, some concern or high risk. Supplementary Appendix B provides a detailed overview of criteria per domain. We visualized the risk-of-bias assessments via the free online tool robvis (https://mcguinlu.shinyapps.io/robvis/).

### Certainty of evidence

Certainty of evidence was assessed using GRADE criteria[128] via the following five domains: (1) risk of bias (for example, the pooled effect is mainly based on studies with insufficient randomization), (2) inconsistency (that is, unexplained heterogeneity), (3) indirectness (for example, the pooled effect is mainly based on interventions that were examined in a particular sub-sample without providing a rationale for selective inclusion), (4) imprecision (that is, the CI does not allow a firm conclusion about the effect and its direction) and (5) publication bias. The assessment of risk of bias was derived from risk of bias v.2.0 assessments (see above). Indirectness was assessed across four domains proposed by GRADE: population, intervention, comparator and outcome. The evaluations of heterogeneity, imprecision and publication bias were derived from the results of the meta-analytic analyses (see below). The interpretation of $I^2$ corresponds to the recommended classification by GRADE (for example, 75% to 100%, considerable heterogeneity).

Certainty of evidence overall can range from high (4) to very low (1). As recommended, certainty of evidence was assessed only for the overarching analyses and not repeated for the sub-analyses.

## Statistical analysis

Hedges' $g$ was calculated[129] whenever (raw) means and standard deviations were reported in the publication (or sent via email). In rare cases, publications reported Hedges' $g$ instead of means and standard deviations, which we then extracted. Hedges' $g$ values were pooled in random-effects meta-analyses given that the present work summarized heterogenous data (that is, diverse behaviours and diverse samples studied in diverse contexts). Statistical analyses were performed with the metafor package (v.3.4.0) in R (v.4.1.1)[130,131]. Two-sided tests were run with $\alpha = 0.05$. We performed a given meta-analysis only when the number of independent data points reached at least four ($k \geq 4$)[126]. We first analysed data in an overarching analysis across all data. We then performed sub-analyses, separating data by three factors that might influence the efficacy of SC-BCTs[5]: (1) desired versus undesired behaviours, (2) the applied social comparison direction of the SC-BCT and (3) the behavioural outcome domain (that is, climate change mitigation, health, performance or service participation; see more details in the Results). To examine whether efficacy between levels of these three factors differed significantly, we entered a given factor in sub-group moderator analyses. Moreover, one potential continuous moderator of efficacy was analysed in meta-regressions: the number of SC-BCT sessions (for example, the number of SC-BCT emails or letters sent to participants or the total number of SC-BCT lab sessions). Moderator analyses were carried out only when sufficient power to detect effects was present (whenever $k \geq 10$ for the continuous moderator and $k \geq 10$ per level of the dichotomous moderators[132]) and only for the overarching analyses to avoid risks of an inflated type I error rate. To examine heterogeneity in outcomes, we calculated the $Q$ statistic and $I^2$. The latter provides an estimation of true heterogeneity in outcomes between studies rather than heterogeneity due to sampling error. To estimate in which margin the true population effect size falls with a given margin of certainty, we calculated 95% CIs of effect sizes. Furthermore, we calculated 95% PIs. PIs supply a margin in which the true population effect is to be expected when similar future trials accumulate[133]. When both the CI and the PI exclude the null, there is particular certainty in the respective effect. To account for the potential effect of extreme observations on the pooled effect size, outlier-adjusted analyses were run whenever one or more outliers were identified. Outliers were defined as extraordinarily low and high effect sizes (that is, at least 3.3 standard deviations below or above the pooled $g$)[134]. To account for small study effects (for example, due to publication bias in the literature), we tested for significant funnel plot asymmetry with Egger's test[135] whenever $k \geq 10$ (ref. [136]). Whenever Egger's test was significant, we used the trim-and-fill method and reported asymmetry-adjusted results when the trim-and-fill method added one or more studies to establish symmetry[137].

## Reporting summary

Further information on research design is available in the Nature Portfolio Reporting Summary linked to this article.

## Data availability

The data that support the findings of this study, along with data collection templates, are available via the Open Science Framework at https://osf.io/uwtbx/?view_only=e7fa20396d3c400e91607f8f81c77359.

## Code availability

Custom analysis code that supports the findings of this study is available via the Open Science Framework at https://osf.io/uwtbx/?view_only=e7fa20396d3c400e91607f8f81c77359. Please note that the code includes forest plots for all performed analyses to maximize transparency and reproducibility.

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

## Acknowledgements

We thank N. Englert, A. S. Lindemann, P. Brouer, L. Olschewski, E. Pauli, D. Herzler and L. Schilling for their contributions to parts of the data extraction process. The authors received no specific funding for this work. All authors are affiliated with the University of Münster, and some authors (T.H.H., R.M.C., P.S. and N.M.) are in paid positions at the University of Münster. The University of Münster was not involved in the conception or conduct of this work.

## Author contributions

T.H.H.: conceptualization, review methodology, screening of literature, data extraction, risk-of-bias assessment, conduct of meta-analysis, writing and revision. R.M.C.: conceptualization, review methodology, screening of literature, data extraction, risk-of-bias assessment, conduct of meta-analysis, writing and revision. J.N.: conceptualization, review methodology, screening of literature and data extraction. F.L.: screening of literature and data extraction. P.S.: revision. N.M.: conceptualization, review methodology, supervision, writing and revision.

## Funding

## Competing interests

The authors declare no competing interests.

## Additional information

**Correspondence and requests for materials** should be addressed to Thole H. Hoppen.

# Reporting Summary

## Statistics

For all statistical analyses, confirm that the following items are present in the figure legend, table legend, main text, or Methods section.

| n/a | Confirmed | |
|---|---|---|
| ☐ | ☒ | The exact sample size (*n*) for each experimental group/condition, given as a discrete number and unit of measurement |
| ☐ | ☒ | A statement on whether measurements were taken from distinct samples or whether the same sample was measured repeatedly |
| ☐ | ☒ | The statistical test(s) used AND whether they are one- or two-sided<br>*Only common tests should be described solely by name; describe more complex techniques in the Methods section.* |
| ☒ | ☐ | A description of all covariates tested |
| ☒ | ☐ | A description of any assumptions or corrections, such as tests of normality and adjustment for multiple comparisons |
| ☐ | ☒ | A full description of the statistical parameters including central tendency (e.g. means) or other basic estimates (e.g. regression coefficient) AND variation (e.g. standard deviation) or associated estimates of uncertainty (e.g. confidence intervals) |
| ☐ | ☒ | For null hypothesis testing, the test statistic (e.g. *F*, *t*, *r*) with confidence intervals, effect sizes, degrees of freedom and *P* value noted<br>*Give P values as exact values whenever suitable.* |
| ☒ | ☐ | For Bayesian analysis, information on the choice of priors and Markov chain Monte Carlo settings |
| ☒ | ☐ | For hierarchical and complex designs, identification of the appropriate level for tests and full reporting of outcomes |
| ☐ | ☒ | Estimates of effect sizes (e.g. Cohen's *d*, Pearson's *r*), indicating how they were calculated |

*Our web collection on statistics for biologists contains articles on many of the points above.*

## Software and code

Policy information about availability of computer code

| Data collection | Data were extracted with Mircosoft Excel for Windows. |
|---|---|
| Data analysis | Statistical analyses were performed with the metafor package (v.3.4.0) in R (v.4.1.1). Risk of bias was evaluated with the Cochrane risk-of-bias tool (RoB 2.0). Visualisation of risk of bias assessments was done via the free online tool robvis (https://mcguinlu.shinyapps.io/robvis/). Certainty of evidence as assessed with GRADE criteria. All extracted data, the codebook, and analysis code are available on the Open Science Framework (OSF) at: https://osf.io/uwtbx/?view_only=e7fa20396d3c400e91607f8f81c77359 |

For manuscripts utilizing custom algorithms or software that are central to the research but not yet described in published literature, software must be made available to editors and reviewers. We strongly encourage code deposition in a community repository (e.g. GitHub). See the Nature Portfolio guidelines for submitting code & software for further information.

## Data

Policy information about availability of data

All manuscripts must include a data availability statement. This statement should provide the following information, where applicable:

- Accession codes, unique identifiers, or web links for publicly available datasets
- A description of any restrictions on data availability
- For clinical datasets or third party data, please ensure that the statement adheres to our policy

This work was pre-registered with the PROSPERO database (PROSPERO-ID: CRD42022343154). The full data, codebook, and analysis code that support the findings of this study are available at the Open Science Framework at: https://osf.io/uwtbx/?view_only=e7fa20396d3c400e91607f8f81c77359

## Research involving human participants, their data, or biological material

Policy information about studies with human participants or human data. See also policy information about sex, gender (identity/presentation), and sexual orientation and race, ethnicity and racism.

| | |
|---|---|
| Reporting on sex and gender | Not applicable given that no new primary data was collected. Rather, available data was summarized in this meta-analysis. Included RCTs mostly concerned mixed-sex/diverse samples from the general population. We considered gender of pariticpants within a given sample within the GRADE assessment of certainty of evidence (i.e., assessment of directness in terms of population within studies). |
| Reporting on race, ethnicity, or other socially relevant groupings | Not applicable given that no new primary data was collected. We do report on the country of conduct for all included RCTs and state that most data in the field are from the US and other high-income countries. Most included RCTs did not report on race, ethnicity, or other socially relevant groupings making a comprehensive/meaningful summary of these variables across included trials infeasible We also report on the implications of most data stemming from the US and other high-income countries (i.e. limited generalisability of results) in the limitations (discussion section). |
| Population characteristics | Not applicable given that no new primary data were collected. Included RCTs mostly concerned large diverse samples from the general population. We considered population characteristics within the GRADE assessment of certainty of evidence (i.e., assessment of directness in terms of population within studies). |
| Recruitment | Not applicable given that no new primary data were collected. We do report in our overview tables concerning RCT characteristics (see Appendix D in the online supplementary materials) whether a general population vs. a specific population sample was recruited in the given RCT. |
| Ethics oversight | Not applicable given that no new primary data were collected. |

Note that full information on the approval of the study protocol must also be provided in the manuscript.

# Field-specific reporting

Please select the one below that is the best fit for your research. If you are not sure, read the appropriate sections before making your selection.

☐ Life sciences     ☒ Behavioural & social sciences     ☐ Ecological, evolutionary & environmental sciences

For a reference copy of the document with all sections, see nature.com/documents/nr-reporting-summary-flat.pdf

# Behavioural & social sciences study design

All studies must disclose on these points even when the disclosure is negative.

| | |
|---|---|
| Study description | Systematic review and meta-analysis with quantitative analysis of standardized mean differences |
| Research sample | Randomised controlled trials evaluating social comparison as a behaviour change technique. Studies had to meet all of the following inclusion criteria to be included in the present work: 1) RCT; 2) at least one arm investigated the efficacy of SC (as defined by Wood) as a BCT (as defined by Michie et al.) and was either a stand-alone or primary SC-BCT (research question 1) or an add-on SC-BCT (research question 2); 3) data for at least one behavioural outcome (e.g., electricity usage) were reported; 4) the outcome was assessed at least 24 hours after the induction of the (first) SC-BCT session to exclude experimental studies assessing only immediate reactions to social comparison; 5) available outcome data included at least ten participants per arm to exclude potential chance findings; and 6) data needed to be peer-reviewed. Withdrawn (i.e., retracted) publications were not included. |
| Sampling strategy | We formulated clear inclusion and exclusion criteria and then conducted a systematic literature search on MEDLINE, PsycINFO, and Web of Science up to Jan 2nd 2024. All studies that met the inclusion criteria and did not fulfill any exclusion criterion were included for analyses. |
| Data collection | Articles found through the abovementioned systematic search that matched our inclusion criteria were collected and data was extracted independently by two of the authors (THH & RMC). |

| Timing | This project was pre-registered on 17th July 2022. Conceptualization of this project started a few months earlier. Given that various steps of conduct took considerable time and given that we wanted to summarize a recent stage of the literature, the present work involved various search waves (with identical search strategy), which were added up and analyzed as a whole. The last search was performed on 2nd January 2024. This last search wave was added as part of the first (major) revision as part of the peer-review process. We worked continously on the present work since conceptualization (April 2022). |
|---|---|
| Data exclusions | Twelve RCTs did not report the relevant data in a usable format and did not reply to at least two data request emails and were thus excluded. Another two publications targeted preventing undesired behaviour from occurring in the future (i.e., alcohol consumption on a specific future occasion) and included various participants who did not engage in this undesired behaviour at baseline (i.e., abstainers) and were thus excluded. Lastly, another four publications reported RCTs that only assessed long-term cognitive or affective change following SC-BCTs but not behaviour change and were thus excluded. We provide basic characteristics of these 18 excluded RCTs (12+2+4) in Appendix C in the online supplementary materials. |
| Non-participation | Not applicable given that no new primary data were collected. In the present meta-analysis, we tried to minimize the risk of bias posed through attrition / data not missing at random in the primary studies (i.e., RCTs) by means of a) generally prioritizing ITT (intent-to-treat) data over completer data and b) by asking primary authors/corresponding authors of the primary work (i.e., RCTs) for ITT (rather than completer) data in our data request emails, which we sent when relevant data was missing. Missing outcome data of randomized participants was one of the five RoB2.0 domains and considered in the GRADE assessment of certainty of evidence. |
| Randomization | Not applicable given that no new primary data were collected. As mentioned above, the present work only included RCTs. "Risk of bias posed through shortcomings in the randomization process" was one of the five RoB2.0 domains evaluated and considered in the GRADE assessment of certainty of evidence. |

# Reporting for specific materials, systems and methods

We require information from authors about some types of materials, experimental systems and methods used in many studies. Here, indicate whether each material, system or method listed is relevant to your study. If you are not sure if a list item applies to your research, read the appropriate section before selecting a response.

## Materials & experimental systems

| n/a | Involved in the study |
|---|---|
| ☒ | ☐ Antibodies |
| ☒ | ☐ Eukaryotic cell lines |
| ☒ | ☐ Palaeontology and archaeology |
| ☒ | ☐ Animals and other organisms |
| ☒ | ☐ Clinical data |
| ☒ | ☐ Dual use research of concern |
| ☒ | ☐ Plants |

## Methods

| n/a | Involved in the study |
|---|---|
| ☒ | ☐ ChIP-seq |
| ☒ | ☐ Flow cytometry |
| ☒ | ☐ MRI-based neuroimaging |

## Plants

| Seed stocks | *Report on the source of all seed stocks or other plant material used. If applicable, state the seed stock centre and catalogue number. If plant specimens were collected from the field, describe the collection location, date and sampling procedures.* |
|---|---|
| Novel plant genotypes | *Describe the methods by which all novel plant genotypes were produced. This includes those generated by transgenic approaches, gene editing, chemical/radiation-based mutagenesis and hybridization. For transgenic lines, describe the transformation method, the number of independent lines analyzed and the generation upon which experiments were performed. For gene-edited lines, describe the editor used, the endogenous sequence targeted for editing, the targeting guide RNA sequence (if applicable) and how the editor was applied.* |
| Authentication | *Describe any authentication procedures for each seed stock used or novel genotype generated. Describe any experiments used to assess the effect of a mutation and, where applicable, how potential secondary effects (e.g. second site T-DNA insertions, mosiacism, off-target gene editing) were examined.* |

