## [Peer Review File · Nature Human Behaviour]

Meta-analysis of randomised controlled trials examining social comparison as a behaviour change technique across behavioural sciences

Corresponding Author: Dr Thole Hoppen

Version 0:

Decision Letter:

15th April 2024

Dear Dr Hoppen,

Thank you once again for your manuscript, entitled "Social comparison as a behaviour change technique: A systematic review and meta-analysis of randomised controlled trials across scientific disciplines", and for your patience during the peer review process.

Your Article has now been evaluated by 3 referees. You will see from their comments copied below that, although they find your work of potential interest, they have raised quite substantial concerns. In light of these comments, we cannot accept the manuscript for publication, but would be interested in considering a revised version if you are willing and able to fully address reviewer and editorial concerns.

We hope you will find the referees' comments useful as you decide how to proceed. If you wish to submit a substantially revised manuscript, please bear in mind that we will be reluctant to approach the referees again in the absence of major revisions. We are committed to providing a fair and constructive peer-review process. Do not hesitate to contact us if there are specific requests from the reviewers that you believe are technically impossible or unlikely to yield a meaningful outcome.

To guide the scope of the revisions, the editors discuss the referee reports in detail within the team, including with the chief editor, with a view to (1) identifying key priorities that should be addressed in revision and (2) overruling referee requests that are deemed beyond the scope of the current study. We hope that you will find the prioritised set of referee points to be useful when revising your study. Please do not hesitate to get in touch if you would like to discuss these issues further.

1. Reviewer 1 raises concerns about your risk of bias assessment, the subgroup analyses, as well as the moderator analyses. These methodological concerns are fundamental and should be thoroughly addressed.
2. Our reviewers find your work to be substantive, but also worry about the work trying to "achieve too much in one paper" and suggest that you only include the meta-analytic results in the paper. We leave this decision up to you but ask that if you decide to keep the narrative synthesis of the additional 15 studies in the paper, you follow Reviewer 1's advice and apply a suitable narrative synthesis method.
3. Reviewer 2 and Reviewer 3 mention that the framing of the work should be improved. Please carefully address these concerns, and revise your work to emphasize the key contribution of your work.
4. Based on Reviewer 1's feedback, we ask that you consider publication bias in more depth, rather than only considering funnel plots and discuss if and how publication bias might have affected your findings.
5. Reviewer 2 and Reviewer 3 mention that the framing of the work should be improved. Please carefully address these concerns and revise your work to emphasize the novelty of your work.

If you wish to submit a suitably revised manuscript, we would hope to receive it within 4 months. I would be grateful if you could contact us as soon as possible if you foresee difficulties with meeting this target resubmission date.

- Include a “Response to the editors and reviewers” document detailing, point-by-point, how you addressed each editor and referee comment. If no action was taken to address a point, you must provide a compelling argument. When formatting this document, please respond to each reviewer comment individually, including the full text of the reviewer comment verbatim followed by your response to the individual point. This response will be used by the editors to evaluate your revision and sent back to the reviewers along with the revised manuscript.
- Highlight all changes made to your manuscript or provide us with a version that tracks changes.

Link Redacted

Thank you for the opportunity to review your work. Please do not hesitate to contact me if you have any questions or would like to discuss the required revisions further.

Sincerely,

██████████
 ██████████
 ██████████
 Nature Human Behaviour

Reviewer expertise:

Reviewer #1: systematic reviews and meta-analysis

Reviewer #2: RCTs ; systematic reviews and meta-analysis ; behaviour change

Reviewer #3: social norms and social identity

REVIEWER COMMENTS:

Reviewer #1:

Remarks to the Author:

This manuscript presents a large-scale systematic review of randomised trials of interventions using social comparison approaches to influence a wide range of behaviours, concluding that these interventions appear to have small but beneficial effect, with the important characteristics of being low-cost and scalable to large populations. The review and its findings would be of great interest to practitioners and policy makers engaged in influencing behaviour across a wide range of topics, including health and climate change.

In reviewing this manuscript, I am drawing on my expertise in methodology for systematic reviews of health and public health interventions. I should not that I am not an expert in social care interventions specifically, and I am not a statistician.

Unfortunately, in my view there are two key methodological issues that may change the conclusions of the review and prevent me from recommending this review for publication. If these were addressed, I think this review would make a valuable contribution and would be a widespread interest. These issues are:

1. Risk of bias assessment

a. The authors have used self-designed criteria to assess risk of bias. The four criteria used are not drawn from the Cochrane risk of bias tool, although Version 1 of this tool is cited in the paper, and they do not adequately assess any of the key types of bias currently addressed by such tools based on evidence of the operation of bias in randomised trials, including, including bias arising from randomisation, bias due to deviations from the intervention, bias due to missing outcome data, bias in measurement of the outcome, and bias in selection of the reported result.

b. The use of dichotomous scoring systems is also no longer considered good practice, nor is summarising the risk of bias across a group of studies as a mean score. This approach does not provide sufficient information to understand and assess the potential impact of risk of bias on the results (that is, is there enough concern about the risk of bias in the set of studies contributing to this result that I should have less confidence in it?), and does not discriminate between incomplete reporting and concerns about the study design. I would not consider this self-designed numerical score an appropriate basis for assessing a moderator affect.

c. I would strongly recommend that the authors re-assess the risk of bias of their included studies using a published, good practice tool (of which there are a number available).

2. Analysis of subgroups and moderator effects

a. Subgroup analysis: The authors indicated that they are interested in the effect of the intervention in various subgroups of data, including increased desired vs decreased undesired behaviour, etc. These are presented as separate analyses, and not described as effect moderators in the methods, or considered for meta-regression. If the key question is the effect of each type of intervention, then I suggest the authors focus on interpreting each effect estimate (for example, noting that the intervention appeared to show a small beneficial effect in each case).

If the authors wish to comment on whether one group appears to be more effective than the other, or that one is effective and the other is not based on statistical significance, then using the separate effect estimates of each group is not the right approach. A

formal subgroup comparison (with a test for subgroup difference), or meta-regression is required. In most cases, based on a scan through the available forest plots, that the effects in each group substantively overlap, indicating that it is unlikely a test for subgroup effect would be significant.

I would strongly recommend that the authors replace these separate forest plots and statements about observed differences with a formal subgroup analysis or meta-regression, and revise their interpretation of the differences accordingly.

b. Moderator analysis: In addition to an overall assessment of the main analyses (including all available studies), the authors also conducted multiple moderator analysis in smaller subsets of the data, and reported some specific small subsets where statistically significant results were observed. I would question the utility and appropriateness of conducting multiple analyses in this way – it resembles data mining or fishing for P values when overarching analysis does not identify a meaningful effect from the moderator under investigation, and in the absence of a clear rationale why there might, for example, be a significant effect for climate-related behaviours and not others. I would focus on the overall analysis for investigating these moderators, and consider providing a clearer interpretation of the meaning of the results in the text, rather than a cherry-picked set of significant results without a clear plausible rationale.

c. There appears to be errors in how the moderator analysis is reported in the text. On p.9, the authors state, “study quality was negatively related with outcomes in some analyses (e.g., any direction SC-BCTs vs. passive controls...”, however, in Table 2 on p.45, the moderator of study quality for all outcomes vs passive controls, not split by direction, is $b = -0.02$, $p=0.670$, which does not indicate a significant association. If I am misunderstanding which result is referred to in the text, this may be assisted by reviewing the clarity of headings and terminology used to distinguish different analyses.

Additional major comments

3. GRADE: I would strongly recommend the authors consider using the GRADE approach to summarise the certainty of the evidence in this review. GRADE is widely accepted as best practice in systematic review methodology, and is recommended by multiple major review-producing organisations, such as Cochrane and JBI, and guideline producers such as WHO and NICE. GRADE provides a more appropriate alternative to considering only the statistical significance of a result, taking into consideration risk of bias, precision, heterogeneity, publication bias and the directness of applicability of the evidence to the review objectives. There is a lot of guidance around about GRADE, (e.g. in the GRADE Working Group’s long series of articles in the Journal of Clinical Epidemiology), but a good starting summary is in the Cochrane Handbook, Chapter 14 (<https://training.cochrane.org/handbook/current/chapter-14>).

4. Narrative synthesis:

a. The results presented for this section in Table 3 do not constitute a synthesis. While synthesis methods exist where meta-analysis is not possible (1-3), these have not been used by the authors. Where it is genuinely not possible to include the results in a meta-analysis, the studies should be grouped according to the same comparisons used in the main analysis, and where there is more than one study in that group, either an alternative synthesis method can be used (e.g. combining P values, reporting the median and IQR, vote counting based on direction of effect), or the results of each study can be summarised.

b. Results should be selected and reported in the same manner as for results included in the meta-analysis. This includes grouping according to the same comparisons, application of the same selection algorithm where multiple results are reported, and wherever possible calculating a single effect estimate (rather than, for example, reporting means and standard deviations for each group). I would strongly recommend a more succinct table of results, and also that the results for these studies should be reported alongside the results of the meta-analysis for the same comparison. Referring readers to the large and detailed table is less helpful than assisting their interpretation by organising related evidence together and commenting, e.g. on whether the findings of these studies are broadly consistent with the meta-analysis in terms of the direction and size of the effect (I would not take statistical significance into consideration here).

c. Looking at the available results, it appears to me that some could have been included in the meta-analysis, although I would defer to the opinion of a statistician in this matter.

Additional minor comments

5. Abstract - I would review the abstract in light of the PRISMA 2020 guidance on reporting of results in abstracts, and consider including more detail (e.g. more information on methods, key numerical results with confidence intervals, etc.)

6. Funding – the authors note no sources of funding for this project. I would enquire about the sources of e.g. funding for the authors’ salaries, even if not directly for this project.

7. Introduction:

a. I note that this paper does not include the very large body of literature on audit and feedback to influence health professional behaviour, which often has a social comparison component (e.g. comparison with peers or health system medians). This may arise from the use of different terminology to describe the interventions (I realise this review focused specifically on interventions that used social comparison terminology). Could you comment briefly on related intervention types that readers may be aware of that are not included in this review?

b. could you define cognitive, affective and behavioural for a non-expert audience?

8. Search – in accordance with PRISMA requirements, could the complete search strategies used in each database be included in an Appendix?

9. Eligibility criteria – the authors noted that included studies must be “peer reviewed”. Was this actually checked as part of the screening process? This is commonly written into methods for systematic reviews but usually assumed to be the case for all papers retrieved in the search. I wouldn’t consider this an essential component, but if stated in the methods it should be correct.

10. Publication bias – I would encourage the authors to consider publication bias in more depth, rather than only considering funnel plots. For example, see a more in-depth consideration of publication bias in the Cochrane Handbook, Chapter 13 (<https://training.cochrane.org/handbook/current/chapter-13>).

11. Results reporting - it would be helpful to report numerical data for all the secondary research questions, as uncertain evidence or a non-statistically significance effect is not the same as no evidence. Readers would benefit from seeing a summary of the main results, rather than referring them to the complex appendices document.

12. Discussion:

a. Given above comments on subgroup analysis, I would avoid drawing conclusions about the differences between the effects in different subgroups unless a formal test for subgroup analysis has been conducted, or on moderating effects that were observed sometimes but not in the overall analysis (without a good rationale). Also, comments about the length of follow-up in climate

change studies are not supported by any formal analysis based on length of follow up, and should be removed unless this analysis is undertaken to substantiate the comments.

b. The results should be discussed in the context of factors that contribute to their certainty – e.g. risk of bias, precision of the estimate, heterogeneity, etc. (this is where GRADE can be useful). Otherwise the Discussion gives the impression of certainty in results where there are in fact a number of factors that might render the results uncertain.

13. Tables:

a. Table 1 is nicely structured and mainly presents key information in succinct format, but purely because of the number of studies involved, it runs for 19 print pages. I would suggest moving this table to the Appendices for this reason, or presenting a modified, less detailed version in the main paper for feasibility reasons. One consideration might be to remove the final three columns – gender and age are not key parameters in your analysis, and the quality score is duplicated in its own table. The column headings could also be simplified, but this would make a limited difference to the overall size of the table.

b. Table 2 runs for 7 print pages, and is quite difficult to navigate. I strongly recommend rationalising this table to improve clarity of the difference between the results reported – dividing into separate tables consistent with the order in which comparisons and results are reported in the review may be helpful and make it considerably easier for the reader to refer between the text and the data. Less important results could be removed, e.g. sensitivity analyses that remove outliers (presenting this alongside the main analysis is duplicative when little difference is observed, and introduces confusion for readers about which is the main result). In addition, as discussed above, I would suggest that the multiple moderator analyses of different subsets of studies could be removed. Reducing the number of repetitive subheadings in favour of clear, succinct descriptors in the first column and e.g. use of differences in shading or indentation of results to indicate subsets of the main result instead could also improve clarity.

c. Table 3 runs to 11 print pages, almost a page per study, and appears to provide more information on these studies than for the main analysis. I would significantly reduce the length of this table, and separate the reporting of results from study characteristics.

14. Appendices: Overall, the appendices are very long and difficult to navigate. It would assist readers if you could take some steps to assist them in finding information, for example:

a. Always refer to specific forest plots by number and/or page number when referring to results in the main text of the paper.

b. Renumber forest plots with Appendix-specific numbers (e.g. E1, E2).

c. Collapse forest plots into subgroups on single forest plots where appropriate.

d. Ensure that Appendix, table and forest plot titles foreground the unique characteristics that differentiate them from others, and avoid repetition of similar characteristics.

e. Include headers or footers to indicate which Appendix you are in and what unique results it includes (e.g. BCT as only component, BCT as main or only component).

f. Organise data in the Appendices in the same order in which it is referred to in the text, rather than require authors to jump back and forth between multiple appendices.

References

1. Higgins JPT, López-López JA, Becker BJ, Davies SR, Dawson S, Grimshaw JM, et al. Synthesising quantitative evidence in systematic reviews of complex health interventions. *BMJ Global Health*. 2019;4(Suppl 1):e000858.

2. Campbell M, McKenzie JE, Sowden A, Katikireddi SV, Brennan SE, Ellis S, et al. Synthesis without meta-analysis (SWiM) in systematic reviews: reporting guideline. *BMJ*. 2020;368:l6890.

3. McKenzie JE, Brennan SE. Chapter 12: Synthesizing and presenting findings using other methods. 2019. In: *Cochrane Handbook for Systematic Reviews of Interventions* [Internet]. Cochrane. Available from: Version 6 (updated July 2019). Available from www.training.cochrane.org/handbook.

Signed:

Dr Miranda Cumpston

Australian Living Evidence Collaboration, School of Public Health and Preventive Medicine, Monash University

Reviewer #2:

Remarks to the Author:

This study reports on a large, cross-domain systematic review, with potential to provide comprehensive and insights about the impact of social-comparison across behavioural domains. It looks very well conducted, but the scale of the data reported presents a clear challenge in how to report all of this in a concise manner. At present, I would argue the authors are attempting to do too much – some aspects do not have enough detail, and other data is so extensive as to be hard to interpret. My suggestion would therefore be for greater prioritisation of what the driving questions are, a clearer structure to break up the amount of data presented so that distinct points are easier to extract. There is a significant imbalance between the size of the Results and Discussion written section, and the volume of tables and figures; this is most evident in that the narrative synthesis is only given two sentences (although the Results largely read well). To do justice to the scale of this work, I wonder whether the authors should consider splitting this into two papers – there are multiple secondary analyses that could be of interest (e.g., comparisons across behavioural domains, differences in length, or control group etc).

Introduction

1) Final paragraph of the introduction. This paragraph as a whole came across as disjointed. More explanation and clearer links are needed at various points, including:

- Lines 75-76 – please provide some background as to what the previous systematic reviews have shown. Given that you are making the argument for the need for a combined version, it would be useful to report what level of effect these are showing, if the different domains appear to differ etc – this will help to justify why you think we need a combined study.

- The argument about looking at moderators is compelling – would be good to make this link more clearly to the later sentences.

E.g., lines 78-80 doesn't seem to link to the statement in 83-85, about how used. How do you see these two aspects linking?

- Similarly, line 84 onwards seems to be a different point – and I could not see how it related to the aims. i.e., if researchers do not measure SC in a group intervention, can it be included in this review? It doesn't look to be relevant, so I wonder if this statement is better suited to the Discussion.

2) It seemed like what you are describing is a study across behavioural domains, rather than across scientific disciplines – as the study of SC impact on cognitions, affect and behaviour is surely within the same discipline (i.e., behavioural science).

Methods

3) The order of presentation and some of the terminology of the methods was difficult to follow. Use of overarching subheadings as per convention may help (e.g., a section on analyses, a section on intervention characteristics etc).

- The paragraph starting in line 105 seems to give one account of how you are defining SC-BCTs, and would be clearer to state what terms or descriptors you would accept and how you decided on these. I think you're saying that descriptions by the study authors of what was made explicit to participants by way of social comparison, would have to qualify for the definition provided by Michie et al in 6.2 of the taxonomy? You are also stating that you are explicitly excluding anything classified as 6.3 (i.e., injunctive norm). Did you use any specific instructions to help coders to do this, if so it would be useful to see the criteria here? Who and how was it decided whether an intervention met the criteria for this BCT?

- The text from line 121 onwards is described as categorising BCTs, but this is confusing as BCTs can be just a small part of an intervention which combines them (as per your own definition in the previous paragraph), and the interventions you are comparing are more than just the BCTs but how they are implemented (i.e., the content and context). Perhaps this is arguable (i.e., I am used to people referring to a BCT as the 'pure' form of a theoretical technique, and an 'intervention' as the broader set of information on how BCTs are operationalised and combined). But at any rate is confusing to use the term BCT in both paragraphs. I would think it more in line with convention to say you are categorising interventions themselves at this point, i.e., on line 122 "SC interventions were categorised into three sub-categories: 1) social comparison as stand-alone SC-BCT (i.e., no other BCTs involved)..." etc. This would help to make these two paragraphs distinct (you use the term bundle, but I think the term intervention is more aligned with other research).

- Line 130 - 140 – The points made here fit better with the paragraph starting on line 105, as they refer to the definition of the BCT, not the design of an intervention (i.e., centrality of social comparison to a study).

- Line 140 – this statement fits better in the analysis section.

4) Line 128 – mixing BCTs may also enhance efficacy, so evidence does not fully support this point. Your reference is only for online apps, there are multiple papers from the Michie research group that show how BCTs are used and bring about effects in concert (and indeed are rarely used alone).

5) Line 142 onwards – a broader section on outcomes would be useful. While you're not restricting the outcomes to a particularly domain, you are restricting them to cognitive, affective or behavioural outcome – so this would be good to state with greater explanation, not least, how you decided which outcome to pick. I see that later you talk about hierarchies, but there may also be multiple outcomes within a similar domain/hierarchy (e.g., 2 behavioural outcomes, or motivation plus self-efficacy etc). This section may be better combined with your presentation of the algorithm of hierarchies.

6) line 154 – It could be helpful to provide more information on how you judged the search terms to be sufficient, e.g., how do these compare with previous systematic reviews?

7) Line 177 – did you compute effect sizes from data available, where not reported by the authors themselves? I see later you go on to state this, but it looks like you don't at this point. This may be better combined with the later section at line 235 so it is less disjointed.

8) Line 207 – what was the relative hierarchy of cognitive over affective outcomes?

9) Line 266 – please include the four items used to judge ROB in the main paper here. Fine to refer to appendices for detail, but the overview is needed.

Results

10) I could not see any clear presentation of differences between the impact of SC on affective, cognitive and behavioural outcomes. Typically behaviour is more challenging to influence, and from Table 2 it looks like there were a lot of behavioural outcomes – and indeed the results are written as if you only analysed the outcomes for behaviour. Is this right? Please report proportion of outcomes in each of the three specified groups, and at some point refer to how each is dealt with and is included in the Results.

11) There is an impressive amount of data and calculations gone into this, but Table 2 is too large to be easy to follow. It would be useful for the authors to set out what their lead findings are here – and prioritising the presentation of those. The remainder could be made supplementary. It may also be useful to split the table.

12) Table 3 takes up a lot of space – and thus implies these findings should be given a lot of weight. Yet, these are not selected as they are more relevant or robust than the remainder, and given very minimal coverage in the results, so this seems a bit off balance. A shorter summary table of primary findings may be more appropriate with an expanded discussion.

13) Take care with interpretations about the type of intervention design, when considered separately from domain and/or other characteristic, unless you are also able to look at these with any other systematic differences. E.g. (looking at lines 362 and thereabouts) could it be that studies using stand-alone BCTs are typically targeting simpler behaviours, or different settings, for different populations than others? That is, be careful to caveat conclusions with clarity that these factors haven't been controlled for.

14) Before reporting on Research question 1b, a summary of the range of active control conditions is needed.

15) Line 392 – this is very informal language, and difficult to follow. If included, it needs to be more informative – it's not clear if the authors are saying there is not really enough evidence to draw a conclusion, or that given the evidence, we have at least enough to suggest there is no effect. Line 399 – again, please provide the number of trials rather than "relatively few".

16) The comment on the narrative synthesis is too short to be meaningful. It isn't an adequate summary of the table – so again, if included, needs to be better presented to provide clear information on the findings and our confidence in them, and perhaps what this adds to the main meta-analytical findings.

Discussion

17) Line 414 – while you state that there are too few studies to warrant synthesis in service participation, does this include in the narrative synthesis. Can you tell us nothing about this area, or are there trends worth reporting?

18) Note: you refer only to behaviour change, please clarify (as per the comment above) if this is purposefully excluding findings for affect and cognition, or all you analysed.

19) It was not clear to me if you looked at potential unintended consequences (e.g., by sharing norms, people change their behaviour in the direction opposite to that intended – which has been reported in some trials such as for alcohol consumption in students). Could this be made clearer?

Reviewer #3:

Remarks to the Author:

Overview

This is an interesting and comprehensive review of the effectiveness of social comparison as a behaviour change technique. There are many strengths to this paper, particularly the thorough methodology. However, I recommend that the framing of the paper is improved to better demonstrate the key findings of the review and the importance of the review. Please see my more detailed comments regarding each section below.

Abstract

The abstract suggests that both seventy and 1,323,478 articles were reviewed. Please clarify that 70 articles were included after the initial literature results of 1,323,478 articles.

Methods

Please specify which PRISMA guidelines (e.g., PRISMA 2020) were followed for this review/

Was there a process for resolving disagreements between the reviewer? For example, was there a group discussion with consensus voting? A little detail would be useful to improve the clarity and replicability of the research.

The diversity of literature included is commendable (e.g., including English, Dutch and German articles). If space allows, could the authors clarify if the search terms and operators were the same for each database? This would help with replicability.

The inclusion criteria of sufficient statistical power is sensible, but I wonder if this excludes qualitative research. Is there a reason that qualitative research was not included? There may be a valid reason for not including qualitative research in the review, but I would mention it as a potential limitation in the discussion section.

Results

The results are explained thoroughly throughout.

Discussion

Throughout the paper, I was waiting for the 'here is why this research is important' statement and I was hoping to find it in this section. The authors have conducted a comprehensive review of the literature in great detail, but the importance of it needs to be made clearer. For example, when the authors mention that their results align with and extend prior findings in related fields, they could make explicit what new knowledge their review brings. The key take-home messages seem to be on page 11 when stating that 1) SC-BCTs can be effective in changing behaviour, and 2) that the effect might be somewhat overestimated. The authors discuss these findings but the importance of them is missing. I recommend that the authors foreground these findings and their importance concretely. This will help to more clearly show both the key findings of the research, the authors' novel contributions, and the messages that the authors want the reader to take from the paper. Please also make it clear how the results relate to the research questions which were laid out at the start of the introduction.

Other

Typo: Page7 line 218 'meaningful'.

General point about the structure: Please split up the paragraphs a little more for ease of reading. I found it a little difficult to identify the main arguments with the current long paragraph structure.

Version 1:

Decision Letter:

6th November 2024

Dear Dr Hoppen,

Thank you once again for your revised manuscript, entitled "Meta-analysis of randomised controlled trials examining social comparison as a behaviour change technique across behavioural sciences," and for your patience during the re-review process.

Your manuscript has now been evaluated by Reviewers 1 from the original round of review, as well as a new Reviewer with expertise in RCTs and behaviour change. All reviewer feedback is included at the end of this letter. Although the reviewers found your manuscript to have improved during revision, they also raise some important outstanding concerns. We remain very interested

in the possibility of publishing your study in Nature Human Behaviour, but would like to consider your response to these outstanding concerns in the form of a revised manuscript before we make a decision on publication.

1. Reviewer 1 raises important concerns about biases introduced due to the classification used in the RoB 2.0 tool. We ask that you thoroughly address these concerns and revise the classification. Please also follow Reviewer 4's advice and provide a visualisation of the risk of bias.

2. Reviewer 4 asks that you provide more information on how BCTs were coded, and which occurred alongside SC. In addition, the reviewer argues that SC cannot be considered a BCT in examples where the intended outcome was not behaviour change. We agree with the reviewer and ask you to remove these trials from the analysis.

3. Please use GRADE to summarise the certainty of evidence as requested by Reviewer 4.

In sum, we invite you to revise your manuscript taking into account all reviewer and editor comments. We are committed to providing a fair and constructive peer-review process. Do not hesitate to contact us if there are specific requests from the reviewers that you believe are technically impossible or unlikely to yield a meaningful outcome.

We hope to receive your revised manuscript within 4-8 weeks. I would be grateful if you could contact us as soon as possible if you foresee difficulties with meeting this target resubmission date.

- Include a "Response to the editors and reviewers" document detailing, point-by-point, how you addressed each editor and referee comment. If no action was taken to address a point, you must provide a compelling argument. This response will be used by the editors and reviewers to evaluate your revision.
- Highlight all changes made to your manuscript or provide us with a version that tracks changes.

Link Redacted

We look forward to seeing the revised manuscript and thank you for the opportunity to review your work. Please do not hesitate to contact me if you have any questions or would like to discuss these revisions further.

Sincerely,

[Redacted]
[Redacted]
[Redacted]
Nature Human Behaviour

Reviewer expertise:

Reviewer #1: systematic reviews and meta-analysis

Reviewer #4: RCTs , behaviour change

REVIEWER COMMENTS:

Reviewer #1 (Remarks to the Author):

I appreciate the extensive additional work conducted by the authors in response to the three peer review reports. Most issues have been satisfactorily addressed.

My one outstanding concern relates to the risk of bias assessment. The switch to using the Risk of Bias 2.0 tool is an appropriate choice, and I appreciate how much work has gone into assessing this large group of included studies. However, the authors have incorporated a novel element of their own design in converting these assessments to numerical scores, which is in contradiction to

the guidance provided for this tool. The reference given at the end of Appendix B for the basis of the scoring system contains no information related to numerical scoring, and instead presents a clear algorithm for converting the domain-level assessments to an ordinal overall conclusion of either low risk, some concerns or high risk, which the authors appear to have applied accurately in the final column of the figure in Appendix D.

The RoB 2.0 tool domains are not intended to be additive - a serious concern in any single domain is sufficient to render the result at high risk of bias. As such, a study rated at overall high risk of bias using RoB 2.0 could score anything between 0 and 8 using the authors' scoring system, and a study rated overall as having some concerns may score between 5 and 9. This overlap can be clearly seen in comparing the studies that scored 7 in Appendix C with the overall rating in Appendix D, where these studies were sometimes rated overall at high risk and sometimes at some concerns. I would suggest that the addition of domains into a numerical scale is an inappropriate interpretation of the risk of bias assessments, and should not be analysed as a continuous variable in the analysis. In the text of the paper, reporting the mean score across studies as 7 points and classifying this as "moderate" (which is not a category in this rating system), could also be misleading.

I would suggest that a better approach would be to use the overall ordinal classification in the final column of Appendix D, and to remove the references to numerical scores. If the authors decide to retain the numerical scoring system, it should be made transparent that this is their own method and not a component of the RoB 2.0 tool.

Signed: Dr Miranda Cumpston
Australian Living Evidence Collaboration, School of Public Health and Preventive Medicine, Monash University.

Reviewer #4 (Remarks to the Author):

The is a comprehensive piece of research, albeit with relatively modest implications due to the small effect sizes and heterogeneity of behaviours that are targeted. However, it is still a useful and interesting contribution to readers and the wider community of behavioural sciences.

As this manuscript has already undergone three detailed reviews, I have highlighted only the things that I think are important that were not mentioned originally.

- RQ2 would be clearer to the reader as something like 'how effective are interventions which contain SC-BCT alongside other BCTs'
- The inclusion criteria allowed for the outcome to be cognitive or affective, and the authors covered all of this under the term 'behavioural outcome'. In my opinion, SC cannot be considered a BCT in examples where the intended outcome was not behaviour change. As there are only four trials, I would remove these.
- There is no information in the method or results about how other BCTs were coded, and which ones occurred alongside SC. If this was not done in a robust manner, is there a chance that the studies were 'under-coded'? Relatedly, it is quite surprising that so many studies only tested the efficacy of a single BCT for quite complex behaviours related to domains such health and the environment. Normally complex interventions feature several BCTs. I am currently finishing a review focusing on another single BCT and there are no studies that isolate only that BCT (i.e., the interventions always feature other BCTs as well). This leads me to question whether the studies in this review that only contain SC-BCT are different from the others. For example, are they conducted in a lab setting, and therefore, less ecologically valid?
- Risk of bias (RoB) coding results are normally presented in a visual RoB figure, which is built into the RoB2 tool from Cochrane. It would be ideal if this was in there.
- The first paragraph of the discussion should mention the small nature of the effects across all meta-analysis variations, particularly as the authors state that lower quality studies probably inflated pooled effects. The authors note that small effects might still be able to produce meaningful change, because of scalability and cost. For this point to be more powerful the focus should be on the health behaviours (using an example or two), rather than the examples such free throw shooting and other performance related behaviours, which are less compelling (and important to readers).

Author responses to reviewers

- Overall, the authors have responded well to reviewer comments with a more streamlined article. However, I do not think that the authors have responded to the reviewer 1 comment about using GRADE. This is an appropriate suggestion and the authors have either misunderstood it or not done it. Summarising the certainty in evidence using GRADE would not take much more work for the authors as they have completed most of the work in separate sections, but it would more clearly frame the findings.

Version 2:

Decision Letter:

12th February 2025

Dear Dr Hoppen,

Thank you once again for your revised manuscript, entitled "Meta-analysis of randomised controlled trials examining social

comparison as a behaviour change technique across behavioural sciences," and for your patience during the re-review process.

Your manuscript has now been evaluated by Reviewers 1 and 4 from the previous round of review. All reviewer feedback is included at the end of this letter. Although the reviewers found your manuscript to have improved during revision, they also raise some important outstanding concerns. We remain very interested in the possibility of publishing your study in Nature Human Behaviour, but would like to consider your response to these outstanding concerns in the form of a revised manuscript before we make a decision on publication.

Specifically, we ask that you thoroughly address the remaining concerns of Reviewer 4 about the identification of SC-BCT interventions as stand-alone.

In sum, we invite you to revise your manuscript taking into account all reviewer and editor comments. We are committed to providing a fair and constructive peer-review process. Do not hesitate to contact us if there are specific requests from the reviewers that you believe are technically impossible or unlikely to yield a meaningful outcome.

We hope to receive your revised manuscript within 4-8 weeks. I would be grateful if you could contact us as soon as possible if you foresee difficulties with meeting this target resubmission date.

- Include a "Response to the editors and reviewers" document detailing, point-by-point, how you addressed each editor and referee comment. If no action was taken to address a point, you must provide a compelling argument. This response will be used by the editors and reviewers to evaluate your revision.
- Highlight all changes made to your manuscript or provide us with a version that tracks changes.

Link Redacted

We look forward to seeing the revised manuscript and thank you for the opportunity to review your work. Please do not hesitate to contact me if you have any questions or would like to discuss these revisions further.

Sincerely,

[Redacted Signature]

[Redacted Name]

Nature Human Behaviour

Reviewer expertise:

Reviewer #1: systematic reviews and meta-analysis

Reviewer #4: RCTs , behaviour change

REVIEWER COMMENTS:

Reviewer #1 (Remarks to the Author):

Thanks again for the considerable amount of work the authors have done in response to the peer review comments. I have no further comments.

Miranda Cumpston
Monash University, Australia

Reviewer #4 (Remarks to the Author):

Overall, the authors have responded well to the additional reviewer comments, particularly regarding RoB and GRADE assessments, and clarification on additional BCT coding. The GRADE in particular was a fair amount of additional work so this effort to revise the manuscript is noted. From my perspective there is only one minor outstanding issue to clarify.

Although it is good to know that only a relatively small percentage of studies were lab-based, I am still not convinced of the claims that there are as many stand-alone SC-BCT interventions as currently stated. The authors provide an example of 'sending a short letter or email containing only SC information, such as a) personal energy consumption vs. b) energy consumption of social comparison standard'. In this example, the participants are being provided with feedback on behaviour by being given a breakdown of their energy consumption, which is another BCT (2.2 in the BCTTv1). To be a stand-alone SC-BCT intervention, the letter or email would not contain information on their personal energy consumption, only information about how their usage compared to reference group. For example, 'your energy consumption was 10% higher than the average household in your area'.

In addition, a quick look at the first few examples of papers that have been classified as stand-alone SC-BCT interventions confirms this.

- Brülisauer et al. (2020) provide feedback in behaviour (2.2) in the intervention groups (mentioned in the title and detailed in Table 1).

- Chapman et al. (2016), study 1, there is more than one additional BCT. The participants are asked to self-monitor their steps (2.3) and are set goals (1.1), which have a frequency and are therefore also an action plan (1.4). If the authors only included study 2 of this paper, then please highlight this in the study characteristics table.

- In Gonçalves et al. (2018), participants appear to receive feedback on their behaviour after every 10 trials (2.2).

Please can the authors elaborate on some of these details.

Dr Neil Howlett

Version 3:

Decision Letter:

Our ref: NATHUMBEHAV-24030906C

27th February 2025

Dear Dr. Hoppen,

Thank you for submitting your revised manuscript "Meta-analysis of randomised controlled trials examining social comparison as a behaviour change technique across behavioural sciences" (NATHUMBEHAV-24030906C). It has now been seen by the original referees and their comments are below. As you can see, the reviewers find that the paper has improved in revision. We will therefore be happy in principle to publish it in Nature Human Behaviour, pending minor revisions to satisfy the referees' final requests and to comply with our editorial and formatting guidelines.

We are now performing detailed checks on your paper and will send you a checklist detailing our editorial and formatting requirements within two weeks. Please do not upload the final materials and make any revisions until you receive this additional information from us.

Sincerely,

[Redacted signature]

[Redacted signature]

Nature Human Behaviour

Reviewer #4 (Remarks to the Author):

I appreciate the authors' considered response to the issue of defining stand-alone SC-BCTs. I note the changes in the method, and that references to this have been removed from the results section (and supplementary tables) altogether.

Apologies for dragging this issue on a bit, but I am left with the question of whether there are any interventions that would be classified as 'stand-alone' now that the definition has been changed. This information has now been removed from Appendix C and D, so the reader cannot tell what the split is between studies that have SC as the stand-alone BCT or as the main behaviour change intervention.

Version 4:

Decision Letter:

Dear Dr Hoppen,

We are pleased to inform you that your Article "Meta-analysis of randomised controlled trials examining social comparison as a behaviour change technique across behavioural sciences", has now been accepted for publication in Nature Human Behaviour.

Authors may need to take specific actions to achieve [compliance with funder and institutional open access mandates](https://www.springernature.com/gp/open-research/funding/policy-compliance-faqs). If your research is supported by a funder that requires immediate open access (e.g. according to [Plan S principles](https://www.springernature.com/gp/open-research/plan-s-compliance)) then you should select the gold OA route, and we will direct you to the compliant route where possible. For authors selecting the subscription publication route, the journal's standard licensing terms will need to be accepted, including [self-archiving policies](https://www.springernature.com/gp/open-research/policies/journal-policies). Those licensing terms will supersede any other terms that the author or any third party may assert apply to any version of the manuscript.

With best regards,

[Redacted signature]

[Redacted signature]

Nature Human Behaviour

P.S. Click on the following link if you would like to recommend Nature Human Behaviour to your librarian
<http://www.nature.com/subscriptions/recommend.html#forms>

** Visit the Springer Nature Editorial and Publishing website at http://editorial-jobs.springernature.com?utm_source=ejp_NHumB_email&utm_medium=ejp_NHumB_email&utm_campaign=ejp_NHumB for more information about our career opportunities. If you have any questions please click [here](mailto:editorial.publishing.jobs@springernature.com).

Editorial comments:

1. Reviewer 1 raises concerns about your risk of bias assessment, the subgroup analyses, as well as the moderator analyses. These methodological concerns are fundamental and should be thoroughly addressed.

RESPONSE: We agree with Reviewer 1. In response, we have assessed all 83 RCTs with the RoB2.0 criteria (for RCTs), as described in more detail in response to Reviewer 1's remarks below. In our re-analyses of the data, we conducted formal moderator analysis and we only compared efficacy for these moderator analyses.

2. Our reviewers find your work to be substantive, but also worry about the work trying to "achieve too much in one paper" and suggest that you only include the meta-analytic results in the paper. We leave this decision up to you but ask that if you decide to keep the narrative synthesis of the additional 15 studies in the paper, you follow Reviewer 1's advice and apply a suitable narrative synthesis method.

RESPONSE: We agree and have excluded the narrative synthesis part from the present work. We have also changed the PRISMA flowchart accordingly (Fig.1). We have moved the trial characteristics table of these 14 excluded studies (note that Ye et al., 2022 sent data in the meantime upon our data request emails and could therefore be included in the re-analysis) to Appendix E in the supplementary material rather than removing it altogether for the following main reason. Appendix E provides information on which trials were in principle eligible but did not report data in sufficient detail to calculate effect sizes (nor were data sent through personal communication on repeated requests) and provides characteristics of these trials, which should be informative for some readers. Please note that we also shortened this table substantially (see Appendix E) by deleting the main results (i.e., narrative synthesis part). That is, only main characteristics are provided in Appendix E.

3. Reviewer 2 and Reviewer 3 mention that the framing of the work should be improved. Please carefully address these concerns, and revise your work to emphasize the key contribution of your work.

RESPONSE: We agree and now clarify the contribution of our work as described in detail in our responses to the comments made by Reviewer 2 and Reviewer 3. Please note that your 5th comment was identical to this 3rd comment, which is why we have deleted the 5th comment.

4. Based on Reviewer 1's feedback, we ask that you consider publication bias in more depth, rather than only considering funnel plots and discuss if and how publication bias might have affected your findings.

RESPONSE: In addition to inspecting funnel plots, we have performed Egger's test (whenever $k \geq 10$, as recommended). Whenever Egger's test was significant, we performed the trim-and-fill adjustment and reported trim-and-fill-adjusted results. In response to this comment, we have clarified our approach in the method section on page 7:

"To account for small study effects (e.g., due to publication bias in the literature), we tested for significant funnel plot asymmetry with the Egger's test⁴⁶ whenever $k \geq 10$ ⁴⁷. Whenever Egger's test was significant, we used the trim-and-fill method and reported asymmetry-adjusted results when the trim-and-fill method added one or more studies to establish symmetry⁴⁸."

We now also report trim-and-fill-adjusted results in more detail for instance on page 9:

"In the overarching analysis at short-term, SC-BCTs were also significantly ($P < .001$) more efficacious in changing behaviour relative to active control conditions, with a small pooled effect ($g = 0.24$, 95% CI = 0.17 – 0.32; $k = 45$; $I^2 = 96\%$; Figure 3). Heterogeneity in outcomes was high and significant. Egger's test indicated significant small study effects and the trim-and-fill analysis added 11 studies to the left. Results remained similar ($g = 0.16$, $P < .001$, 95% CI = 0.07 – 0.25; $k = 46$; $I^2 = 98\%$)."

We have further added the following to the limitation section (page 11):

"Fourth, while we adjusted for the potential impact of publication bias on meta-analytic synthesis (i.e., Egger's test and the trim-and-fill method), we excluded non-peer-reviewed data. While maximizing internal validity through synthesizing only quality-controlled data, this decision may have diminished external validity of results and results might be biased by publication bias (i.e., underreporting of null results). Notably, some included RCTs did report on null results. Yet, it remains unknown to what extent the present results are affected by publication bias."

REVIEWER COMMENTS:

Reviewer #1:

Remarks to the Author:

This manuscript presents a large-scale systematic review of randomised trials of interventions using social comparison approaches to influence a wide range of behaviours, concluding that these interventions appear to have small but beneficial effect, with the important characteristics of being low-cost and scalable to large

populations. The review and its findings would be of great interest to practitioners and policy makers engaged in influencing behaviour across a wide range of topics, including health and climate change.

In reviewing this manuscript, I am drawing on my expertise in methodology for systematic reviews of health and public health interventions. I should not that I am not an expert in social care interventions specifically, and I am not a statistician.

Unfortunately, in my view there are two key methodological issues that may change the conclusions of the review and prevent me from recommending this review for publication. If these were addressed, I think this review would make a valuable contribution and would be a widespread interest. These issues are:

1. Risk of bias assessment

a. The authors have used self-designed criteria to assess risk of bias. The four criteria used are not drawn from the Cochrane risk of bias tool, although Version 1 of this tool is cited in the paper, and they do not adequately assess any of the key types of bias currently addressed by such tools based on evidence of the operation of bias in randomised trials, including, including bias arising from randomisation, bias due to deviations from the intervention, bias due to missing outcome data, bias in measurement of the outcome, and bias in selection of the reported result.

b. The use of dichotomous scoring systems is also no longer considered good practice, nor is summarising the risk of bias across a group of studies as a mean score. This approach does not provide sufficient information to understand and assess the potential impact of risk of bias on the results (that is, is there enough concern about the risk of bias in the set of studies contributing to this result that I should have less confidence in it?), and does not discriminate between incomplete reporting and concerns about the study design. I would not consider this self-designed numerical score an appropriate basis for assessing a moderator affect.

c. I would strongly recommend that the authors re-assess the risk of bias of their included studies using a published, good practice tool (of which there are a number available).

RESPONSE: We are grateful to Reviewer 1 for their positive feedback and the constructive comments that helped to improve our manuscript.

In response to this comment, we have assessed risk of bias with the RoB2.0 criteria and re-run moderator analyses. Overall, the results remained very similar. That is, risk of bias was negatively associated with effect sizes in one (but not the other) moderator analysis. As mentioned above, we were able to add another search wave (literature published Feb 2023 to Jan 2024). This additional search wave yielded 13 additional eligible RCTs, resulting in a total of 83 RCTs included in the meta-analysis. RoB2.0 assessments of all 83 RCTs are depicted in Appendix C and visualized in Appendix D. For readers unfamiliar with the RoB2.0, we have summarized the RoB (sub)domains in Appendix B including their scoring. We also summarize risk of bias across studies in the results section (page 8):

"Study quality sum scores ranged from 3 to 10 (possible range = 0 to 10). Appendix B provides all scores. The mean quality sum score was 7 (SD = 1.6), indicating moderate quality across the 83 trials. In most trials (k = 67; 80%), some concern of bias arose from the randomisation process. Only in eight trials (10%), risk of bias arising from the randomisation process was judged to be low. Conversely, eight trials (10%) were judged to be at high risk of bias due to concerns with randomisation. Half of the trials (k = 45; 54%) presented with low risk of bias due to deviations from the intended interventions, 29% of trials (k = 24) with some concern for bias, and 17% of trials (k = 14) with high risk of bias. Most studies (k = 65; 78%) presented with low risk of bias due to missing outcome data and the remaining trials with high risk. Most studies (k = 77; 93%) presented with low risk of bias in measurement of the outcome and the remaining trials with high risk. Lastly, most trials (k = 68; 82%) presented with some concern for bias in terms of selection of the reported result(s) given that pre-registrations and pre-specified analysis protocols were very rare. The remaining 15 trials (18%) presented with low risk of bias in terms of selection of the reported result(s)."

2. Analysis of subgroups and moderator effects

a. Subgroup analysis: The authors indicated that they are interested in the effect of the intervention in various subgroups of data, including increased desired vs decreased undesired behaviour, etc. These are presented as separate analyses, and not described as effect moderators in the methods, or considered for meta-regression. If the key question is the effect of each type of intervention, then I suggest the authors focus on interpreting each effect estimate (for example, noting that the intervention appeared to show a small beneficial effect in each case).

If the authors wish to comment on whether one group appears to be more effective than the other, or that one is effective and the other is not based on statistical significance, then using the separate effect estimates of each group is not the right approach. A formal subgroup comparison (with a test for subgroup difference), or meta-regression is required. In most cases, based on a scan through the available forest plots, that the effects in each group substantively overlap, indicating that it is unlikely a test for subgroup effect would be significant.

I would strongly recommend that the authors replace these separate forest plots and statements about observed differences with a formal subgroup analysis or meta-regression, and revise their interpretation of the differences accordingly.

RESPONSE: We agree with the reviewer and have performed moderator analyses for all comparisons of efficacy. We have now clarified our approach in the method section on page 6:

"We first analysed data in an overarching analysis across all data. We then performed sub-analyses separating data by four factors that might influence the efficacy of SC-BCTs⁵: a) desired vs. undesired behaviours, b), the applied social comparison direction of the SC-BCT, c) the behavioural outcome domain (i.e., climate change mitigation-, health-, performance-

, and service participation-related behaviour, see more details in the results), and d) social comparison as the stand-alone BCT vs. social comparison as the main BCT. To examine whether efficacy between levels of these four factors differed significantly, we entered a given factor in sub-group moderator analyses. Moreover, two potential continuous moderators of efficacy were analysed in meta-regressions: a) study quality (see risk of bias assessment above) and b) number of SC-BCT sessions (e.g., number of SC-BCT emails or letters sent to participants or total number of SC-BCT lab sessions). Moderator analyses were only carried out when sufficient power to detect effects was present (whenever $k \geq 10$ for continuous moderators and whenever $k \geq 10$ per level of the dichotomous moderators⁴²) and only for the overarching analyses to avoid risks of an inflated type one error rate."

Results of the overarching and sub-analyses are presented in Table 1, whereas results of the moderator analyses have been separated and are now presented in Table 2. We also report these data (i.e., efficacy results vs. moderations) separately under different subheadings in the results sections. This separation of results increased clarity and structure of the present work.

b. Moderator analysis: In addition to an overall assessment of the main analyses (including all available studies), the authors also conducted multiple moderator analysis in smaller subsets of the data, and reported some specific small subsets where statistically significant results were observed. I would question the utility and appropriateness of conducting multiple analyses in this way – it resembles data mining or fishing for P values when overarching analysis does not identify a meaningful effect from the moderator under investigation, and in the absence of a clear rationale why there might, for example, be a significant effect for climate-related behaviours and not others. I would focus on the overall analysis for investigating these moderators, and consider providing a clearer interpretation of the meaning of the results in the text, rather than a cherry-picked set of significant results without a clear plausible rationale.

RESPONSE: We agree and, as suggested, we have restricted moderator analyses to the overarching analyses, which we have now clarified in our method section (page 6):

"Moderator analyses were only carried out when sufficient power to detect effects was present (whenever $k \geq 10$ for continuous moderators and whenever $k \geq 10$ per level of the dichotomous moderators⁴²) and only for the overarching analyses to avoid risks of an inflated type one error rate."

We ensured that both significant as well as non-significant moderator results are reported (pages 9-10):

"Table 2 displays all moderator analysis results. In the overarching analysis comparing SC-BCTs to passive controls at short-term, no significant moderations were observed. In the analysis comparing SC-BCTs to active controls, study quality was negatively associated with

short-term efficacy of SC-BCTs ($k = 45$, $b = -0.09$, $P < .001$), whereas number of SC-BCT sessions ($k = 36$, $b = 0.02$, $P = .008$), desired (vs. undesired) behaviour ($k = 44$, $b = 0.17$, $P = .027$), and SC-BCT as the only (vs. main) intervention ($k = 45$, $b = 0.21$, $P = .005$) were all positively related to short-term efficacy of SC-BCTs. No evidence was found for a significant difference in efficacy of upward vs. not restricted SC-BCTs (in both overarching analyses, relative to passive and active controls, respectively), nor for SC-BCTs targeting health vs. climate change mitigation behaviours (overarching analysis comparing to passive controls), nor for SC-BCTs targeting health vs. performance behaviours (overarching analysis comparing to active controls)."

In our discussion and interpretation of the moderator results, we have ensured that a balanced overview is given, including the (implications of) non-significant moderator analyses and the necessity for further investigation in future research (pages 10-11):

"The present work extends prior work by performing various moderator analyses. In the overarching analysis comparing SC-BCTs to active controls, number of SC-BCT sessions was positively associated with efficacy, suggesting a dose-response relationship, which might be attributable to cumulative reinforcement, habit formation, or cognitive or emotional shifts, fostering sustained engagement and internalization of change. However, in the overarching analysis comparing SC-BCTs to passive controls, the number of SC-BCT sessions did not moderate efficacy. Consequently, to clearly discern when and under what circumstances a dose-response relationship can be assumed, further research is needed. Similarly, the significant negative moderation of study quality in the overarching analysis comparing SC-BCTs to active (but not passive) control conditions suggests that effects are overestimated. Pooled effects are biased by lower-quality evidence finding larger effects. Future research needs to improve methodological quality to ensure accurate estimation of effects. As more trials accumulate over time, future meta-analytic research should re-evaluate the present finding, especially as the present work only found this association in one analysis (but not the other). When comparing SC-BCTs to active (but not passive) conditions, SC-BCTs targeting desired behaviours were associated with higher efficacy compared to SC-BCTs targeting undesired behaviour. This suggests that SC-BCTs might be more effective when they target a desired behaviour (e.g., healthy diet) rather than an undesired behaviour (e.g., unhealthy diet). However, given mixed findings across the moderator analyses relative to passive and active controls, this association remains preliminary and requires further testing. SC-BCTs involving social comparison as the stand-alone intervention were associated with significantly higher efficacy of behaviour change compared to SC-BCTs involving social comparison as the main intervention (e.g., next to providing information). However, again, this moderation was only observed in the studies comparing to active (but not passive) controls and hence requires further investigation. Also, trials on stand-alone vs. main SC-BCTs might differ on third variables (e.g., behaviours studied, study settings, populations studied), which may explain the significant difference found. More research is needed."

c. There appears to be errors in how the moderator analysis is reported in the text. On p.9, the authors state, "study quality was negatively related with outcomes in some analyses (e.g., any direction SC-BCTs vs. passive controls..." however, in Table 2 on p.45, the moderator of study quality for all outcomes vs passive controls, not split by direction, is $b = -0.02$, $p=0.670$, which does not indicate a significant association. If I am misunderstanding which result is referred to in the text, this may be assisted by reviewing the clarity of headings and terminology used to distinguish different analyses.

RESPONSE: As mentioned above, we now restricted moderator analyses to the overarching analyses. As a result, the information the reviewer refers to in the comment has been deleted.

Additional major comments

3. GRADE: I would strongly recommend the authors consider using the GRADE approach to summarise the certainty of the evidence in this review. GRADE is widely accepted as best practice in systematic review methodology, and is recommended by multiple major review-producing organisations, such as Cochrane and JBI, and guideline producers such as WHO and NICE. GRADE provides a more appropriate alternative to considering only the statistical significance of a result, taking into consideration risk of bias, precision, heterogeneity, publication bias and the directness of applicability of the evidence to the review objectives. There is a lot of guidance around about GRADE, (e.g. in the GRADE Working Group's long series of articles in the Journal of Clinical Epidemiology), but a good starting summary is in the Cochrane Handbook, Chapter 14 (<https://training.cochrane.org/handbook/current/chapter-14>).

RESPONSE: As mentioned above, we have assessed risk of bias of the 83 included RCTs using the Cochrane's RoB2.0 criteria for randomised controlled trials and report our methodology and results accordingly. We also calculated 95% prediction intervals in addition to 95% confidence intervals to estimate the certainty of the evidence. We describe our approach on pages 6-7 as follows:

"Furthermore, we calculated 95% prediction intervals (PIs). PIs supply a margin in which the true population effect is to be expected when similar future trials accumulate⁴³. When the CI as well as the PI exclude the null there is particular certainty in the respective effect."

We also report limited certainty based on most prediction intervals including the null in the discussion (page 10):

"Across 83 RCTs, we found evidence for the efficacy of SC-BCT in shaping behaviour in the desired direction. Evidence was found across type of control condition (i.e., passive controls and active controls), across behavioural domains (i.e., health behaviour, climate change mitigation behaviour, performance behaviour, and service participation behaviour), across

desired and undesired behaviours, and across time (i.e., short- and long-term assessments). However, the prediction intervals often included the null, limiting certainty in effects. More research is necessary to examine the robustness of the results."

4. Narrative synthesis:

a. The results presented for this section in Table 3 do not constitute a synthesis. While synthesis methods exist where meta-analysis is not possible(1-3), these have not been used by the authors. Where it is genuinely not possible to include the results in a meta-analysis, the studies should be grouped according to the same comparisons used in the main analysis, and where there is more than one study in that group, either an alternative synthesis method can be used (e.g. combining P values, reporting the median and IQR, vote counting based on direction of effect), or the results of each study can be summarised.

b. Results should be selected and reported in the same manner as for results included in the meta-analysis. This includes grouping according to the same comparisons, application of the same selection algorithm where multiple results are reported, and wherever possible calculating a single effect estimate (rather than, for example, reporting means and standard deviations for each group). I would strongly recommend a more succinct table of results, and also that the results for these studies should be reported alongside the results of the meta-analysis for the same comparison. Referring readers to the large and detailed table is less helpful than assisting their interpretation by organising related evidence together and commenting, e.g. on whether the findings of these studies are broadly consistent with the meta-analysis in terms of the direction and size of the effect (I would not take statistical significance into consideration here).

c. Looking at the available results, it appears to me that some could have been included in the meta-analysis, although I would defer to the opinion of a statistician in this matter.

RESPONSE: In response to the suggestion by the editor that we only include the meta-analytic results in the paper, we have deleted this part from the present work (i.e., excluded these trials). We have also changed the PRISMA flowchart accordingly (Fig.1). We have moved the trial characteristics table of these 14 excluded studies (note that Ye et al., 2022 sent data in the meantime upon our data request emails and could therefore be included in the re-analysis) to Appendix E in the supplementary material. Appendix E is included in the supplementary material to inform readers on which trials were in principle eligible but did not report sufficiently detailed data to calculate effect sizes (nor were data sent through personal communication on repeated requests) and now only provides basic trial characteristics. That is, we have deleted the main results of these excluded trials in response to this comment. We made sure to clarify the reason for exclusion in the main manuscript as follows (page 7):

"A total of 665 full-texts were excluded for not meeting inclusion criteria. Twelve of these⁴⁸⁻⁵⁹ were principally eligible but did not report data in a usable format (i.e., Hedges' gs could

not be calculated from reported data) and primary authors did not respond to at least two data request emails. A short summary of the characteristics of these trials is provided in Appendix E. Two other publications^{60,61} targeted preventing undesired behaviour from occurring in the future (i.e., alcohol consumption on a specific future occasion) and included various participants who did not engage in this undesired behaviour at baseline (i.e., abstainers) and were thus excluded (see Appendix E)."

Additional minor comments

5. Abstract - I would review the abstract in light of the PRISMA 2020 guidance on reporting of results in abstracts, and consider including more detail (e.g. more information on methods, key numerical results with confidence intervals, etc.)

RESPONSE: We have changed the abstract accordingly (page 2) as follows. Please note that the abstract now slightly exceeds the word limit (172 words):

"Research on social comparison as a behaviour change technique (SC-BCT) has increased recently. We conducted a meta-analysis of randomised controlled trials (RCTs) investigating SC-BCTs across behavioural sciences (PROSPERO: CRD42022343154). We searched MEDLINE, PsycINFO, and Web of Science from inception to Jan 2024. Eighty-three RCTs (N = 1,357,062) were included and investigated effects on climate change mitigation-, health-, performance-, and service-related behaviour. Study quality was assessed with Cochrane risk-of-bias 2.0 criteria and was moderate on average. At short-term (M = 3.5 months post-intervention), SC-BCTs produced small effects relative to both passive ($g = 0.17$, 95% CI = 0.11 – 0.23, $P < .001$) and active control conditions ($g = 0.24$, 95% CI = 0.17 – 0.32, $P < .001$). Number of SC-BCT sessions and desired (vs. undesired) behaviours were positively and study quality was negatively associated with effects. Moderation was found only in some analyses, calling for further testing. SC-BCTs also produced significant small effects at long-term (M = 6 months post-intervention). Small effects should be interpreted in the context of low-cost and large scalability."

6. Funding – the authors note no sources of funding for this project. I would enquire about the sources of e.g. funding for the authors' salaries, even if not directly for this project.

RESPONSE: We now report (page 2):

"Funding: This project did not receive any specific funding. All authors are affiliated with the University of Münster and some authors (THH, RMC, PS, & NM) are in paid positions at the University of Münster. The University of Münster was not involved in the conception or conduct of the present work."

7. Introduction:

a. I note that this paper does not include the very large body of literature on audit and

feedback to influence health professional behaviour, which often has a social comparison component (e.g. comparison with peers or health system medians). This may arise from the use of different terminology to describe the interventions (I realise this review focused specifically on interventions that used social comparison terminology). Could you comment briefly on related intervention types that readers may be aware of that are not included in this review?

RESPONSE: We discuss this point in more depth now in the limitation section as follows (page 11):

"There are limitations to our study. First, we aimed at only including research that explicitly referred to social comparison or social comparison-related terms (e.g., upward/downward/lateral comparison). This choice was made in an effort to maximize internal validity. Yet, this may have resulted in missing related literature. For example, research on audit and feedback to influence health professional behaviour often involves interpersonal comparison (e.g., comparison with median performance) as one part of a BCT bundle¹³⁷. Likewise, any group-based intervention is likely to be influenced (partly) by social comparison processes. Yet, studies not explicitly referring to social comparison (recall that we performed all-fields searches) were deemed unlikely to entail a study design that enables the evaluation of the individual impact of SC-BCTs on behaviour change. Despite the restriction on research explicitly referring to social comparison terms, the present work covered a broad range of literature (83 RCTs) and a very large number of included participants (N = 1,357,062)."

b. could you define cognitive, affective and behavioural for a non-expert audience?

RESPONSE: We now give examples as recommended in the paragraph on inclusion criteria as follows (page 5):

"(...); 3) data for at least one behavioural outcome (e.g., electricity usage), cognitive outcome (e.g., intention to change electricity usage), or affective outcome (e.g., feelings about changing electricity usage) was reported; (...)."

8. Search – in accordance with PRISMA requirements, could the complete search strategies used in each database be included in an Appendix?

RESPONSE: The search strategy is now provided in Appendix A.

9. Eligibility criteria – the authors noted that included studies must be "peer reviewed". Was this actually checked as part of the screening process? This is commonly written into methods for systematic reviews but usually assumed to be the case for all papers retrieved in the search. I wouldn't consider this an essential component, but if stated in the methods it should be correct.

RESPONSE: We verify that this is correct. All 83 included RCTs were reported in published journal articles after formal peer-review (some recent ones in "first view"/"online first" format, which we highlighted with the notion: "in print").

10. Publication bias – I would encourage the authors to consider publication bias in more depth, rather than only considering funnel plots. For example, see a more in-depth consideration of publication bias in the Cochrane Handbook, Chapter 13 (<https://training.cochrane.org/handbook/current/chapter-13>).

RESPONSE: We would like to clarify that in addition to inspecting funnel plots, we performed Egger's test (whenever $k \geq 10$, as recommended). Whenever Egger's test was significant, we performed the trim-and-fill adjustment and reported trim-and-fill-adjusted results. In response to this comment, we have clarified our approach in the method section on page 7:

"To account for small study effects (e.g., due to publication bias in the literature), we tested for significant funnel plot asymmetry with the Egger's test⁴⁵ whenever $k \geq 10$ ⁴⁶. Whenever Egger's test was significant, we used the trim-and-fill method and reported asymmetry-adjusted results when the trim-and-fill method added one or more studies to establish symmetry⁴⁷."

We also report trim-and-fill-adjusted results on page 9:

"In the overarching analysis at short-term, SC-BCTs were also significantly ($P < .001$) more efficacious in changing behaviour relative to active control conditions, with a small pooled effect ($g = 0.24$, 95% CI = 0.17 – 0.32; $k = 45$; $I^2 = 96\%$; Figure 3). Heterogeneity in outcomes was high and significant. Egger's test indicated significant small study effects and the trim-and-fill analysis added 11 studies to the left. Results remained similar ($g = 0.16$, $P < .001$, 95% CI = 0.07 – 0.25; $k = 46$; $I^2 = 98\%$).

We have further added the following to the limitation section (pages 11-12):

"Fourth, while we adjusted for the potential impact of publication bias on meta-analytic synthesis (i.e., Egger's test and the trim-and-fill method), we excluded non-peer-reviewed data. While maximizing internal validity through synthesizing only quality-controlled data, this decision may have diminished external validity of results and results might be biased by publication bias (i.e., underreporting of null results). Notably, some included RCTs did report on null results. Yet, it remains unknown to what extent the present results are affected by publication bias."

11. Results reporting - it would be helpful to report numerical data for all the secondary

research questions, as uncertain evidence or a non-statistically significance effect is not the same as no evidence. Readers would benefit from seeing a summary of the main results, rather than referring them to the complex appendices document.

RESPONSE: We now report the results concerning research question 2 (SC-BCT as add-on intervention) in Table 3. We also clarified the interpretation of results, as suggested (page 10):

"Table 3 shows all results. Eight trials investigated this research question. In both the overarching analysis and the sub-analysis (i.e., upward add-on SC-BCTs only), differences in efficacy between BCT bundles with and without the add-on SC-BCTs were nonsignificant. The low number of trials illustrates the low statistical power to detect significant differences."

12. Discussion:

a. Given above comments on subgroup analysis, I would avoid drawing conclusions about the differences between the effects in different subgroups unless a formal test for subgroup analysis has been conducted, or on moderating effects that were observed sometimes but not in the overall analysis (without a good rationale). Also, comments about the length of follow-up in climate change studies are not supported by any formal analysis based on length of follow up, and should be removed unless this analysis is undertaken to substantiate the comments.

RESPONSE: As mentioned above, we have performed moderator analyses and only compare efficacy based on moderator analyses (rather than comparing results of subgroup analyses). Given that the moderator results were mixed, we discuss results cautiously in the discussion as follows (pages 10-11):

"The present work extends prior work by performing various moderator analyses. In the overarching analysis comparing SC-BCTs to active controls, number of SC-BCT sessions was positively associated with efficacy, suggesting a dose-response relationship, which might be attributable to cumulative reinforcement, habit formation, or cognitive or emotional shifts, fostering sustained engagement and internalization of change. However, in the overarching analysis comparing SC-BCTs to passive controls, the number of SC-BCT sessions did not moderate efficacy. Consequently, to clearly discern when and under what circumstances a dose-response relationship can be assumed, further research is needed. Similarly, the significant negative moderation of study quality in the overarching analysis comparing SC-BCTs to active (but not passive) control conditions suggests that effects are overestimated. Pooled effects are biased by lower-quality evidence finding larger effects. Future research needs to improve methodological quality to ensure accurate estimation of effects. As more trials accumulate over time, future meta-analytic research should re-evaluate the present finding, especially as the present work only found this association in one analysis (but not the other). When comparing SC-BCTs to active (but not passive) conditions, SC-BCTs

targeting desired behaviours were associated with higher efficacy compared to SC-BCTs targeting undesired behaviour. This suggests that SC-BCTs might be more effective when they target a desired behaviour (e.g., healthy diet) rather than an undesired behaviour (e.g., unhealthy diet). However, given mixed findings across the moderator analyses relative to passive and active controls, this association remains preliminary and requires further testing. SC-BCTs involving social comparison as the stand-alone intervention were associated with significantly higher efficacy of behaviour change compared to SC-BCTs involving social comparison as the main intervention (e.g., next to providing information). However, again, this moderation was only observed in the studies comparing to active (but not passive) controls and hence requires further investigation. Also, trials on stand-alone vs. main SC-BCTs might differ on third variables (e.g., behaviours studied, study settings, populations studied), which may explain the significant difference found. More research is needed.”

We also removed the sentence on differential length of follow-up in studies investigating climate change mitigation behaviours, as suggested.

b. The results should be discussed in the context of factors that contribute to their certainty – e.g. risk of bias, precision of the estimate, heterogeneity, etc. (this is where GRADE can be useful). Otherwise the Discussion gives the impression of certainty in results where there are in fact a number of factors that might render the results uncertain.

RESPONSE: We now report limits to certainty in results in more detail in the discussion section. Page 10:

“Across 83 RCTs, we found evidence for the efficacy of SC-BCT in shaping behaviour in the desired direction. Evidence was found across type of control condition (i.e., passive controls and active controls), across behavioural domains (i.e., health behaviour, climate change mitigation behaviour, performance behaviour, and service participation behaviour), across desired and undesired behaviours, and across time (i.e., short- and long-term assessments). However, the prediction intervals often included the null, limiting certainty in effects. More research is necessary to examine the robustness of the results.”

13. Tables:

a. Table 1 is nicely structured and mainly presents key information in succinct format, but purely because of the number of studies involved, it runs for 19 print pages. I would suggest moving this table to the Appendices for this reason, or presenting a modified, less detailed version in the main paper for feasibility reasons. One consideration might be to remove the final three columns – gender and age are not key parameters in your analysis, and the quality score is duplicated in its own table. The column headings could

also be simplified, but this would make a limited difference to the overall size of the table.

RESPONSE: In response to this comment, we have moved the table to the supplementary material (see Appendix F). We decided to keep the three mentioned columns given that we believe that these might be informative to readers (i.e., all main characteristics of a given trial visible on one spot).

b. Table 2 runs for 7 print pages, and is quite difficult to navigate. I strongly recommend rationalising this table to improve clarity of the difference between the results reported – dividing into separate tables consistent with the order in which comparisons and results are reported in the review may be helpful and make it considerably easier for the reader to refer between the text and the data. Less important results could be removed, e.g. sensitivity analyses that remove outliers (presenting this alongside the main analysis is duplicative when little difference is observed, and introduces confusion for readers about which is the main result). In addition, as discussed above, I would suggest that the multiple moderator analyses of different subsets of studies could be removed. Reducing the number of repetitive subheadings in favour of clear, succinct descriptors in the first column and e.g. use of differences in shading or indentation of results to indicate subsets of the main result instead could also improve clarity.

RESPONSE: This table can now be found in Table 1. We have decided to move the moderator results to a separate table (see Table 2), which makes both tables more comprehensible. We have also removed redundant subheadings in the Tables and the respective Tables are now shorter (e.g., Table 1 is now 2 pages shorter despite covering more analyses in the light of the 13 added trials from the new search wave).

c. Table 3 runs to 11 print pages, almost a page per study, and appears to provide more information on these studies than for the main analysis. I would significantly reduce the length of this table, and separate the reporting of results from study characteristics.

RESPONSE: As mentioned, this table is now presented in Appendix E and studies were excluded from synthesis. Appendix E only provides an overview of main study characteristics (not main results).

14. Appendices: Overall, the appendices are very long and difficult to navigate. It would assist readers if you could take some steps to assist them in finding information, for example:

a. Always refer to specific forest plots by number and/or page number when referring to results in the main text of the paper.

RESPONSE: We have decided to delete all forest plots from the supplementary material, which shortened it substantially and increased structure. These forest plots were included

at first to maximize transparency and reproducibility. To remain transparent and reproducible, the revised version of the manuscript includes the following change (page 14):

"Data availability:

The full data that support the findings of this study are available at the Open Science Framework: https://osf.io/uwtbx/?view_only=e7fa20396d3c400e91607f8f81c77359

Code availability:

The full code (R-Script) that supports the findings of this study is available at the Open Science Framework (see link below). Please note that the code includes forest plots for all performed analyses which is meant to maximize transparency and reproducibility: https://osf.io/uwtbx/?view_only=e7fa20396d3c400e91607f8f81c77359"

b. Renumber forest plots with Appendix-specific numbers (e.g. E1, E2).

RESPONSE: As mentioned above, these were deleted.

c. Collapse forest plots into subgroups on single forest plots where appropriate.

RESPONSE: We now only report forest plots in Fig 2 (overarching analysis in comparison to passive controls) and Fig 3 (overarching analysis in comparison to active controls). All other forest plots can be reproduced via open data and open code, as mentioned above.

d. Ensure that Appendix, table and forest plot titles foreground the unique characteristics that differentiate them from others, and avoid repetition of similar characteristics.

RESPONSE: Done.

e. Include headers or footers to indicate which Appendix you are in and what unique results it includes (e.g. BCT as only component, BCT as main or only component).

RESPONSE: Done. And all Appendices are hyperlinked to help the reader navigate.

f. Organise data in the Appendices in the same order in which it is referred to in the text, rather than require authors to jump back and forth between multiple appendices.

RESPONSE: Done. We thank Reviewer 1 once more for these very helpful points that have substantially improved the present manuscript.

References

1. Higgins JPT, López-López JA, Becker BJ, Davies SR, Dawson S, Grimshaw JM, et al. Synthesising quantitative evidence in systematic reviews of complex health interventions.

BMJ Global Health. 2019;4(Suppl 1):e000858.

2. Campbell M, McKenzie JE, Sowden A, Katikireddi SV, Brennan SE, Ellis S, et al. Synthesis without meta-analysis (SWiM) in systematic reviews: reporting guideline. *BMJ*. 2020;368:l6890.

3. McKenzie JE, Brennan SE. Chapter 12: Synthesizing and presenting findings using other methods. 2019. In: *Cochrane Handbook for Systematic Reviews of Interventions* [Internet]. Cochrane. Available from: Version 6 (updated July 2019). Available from www.training.cochrane.org/handbook.

Signed:

Dr Miranda Cumpston

Australian Living Evidence Collaboration, School of Public Health and Preventive Medicine, Monash University

Reviewer #2:

Remarks to the Author:

This study reports on a large, cross-domain systematic review, with potential to provide comprehensive and insights about the impact of social-comparison across behavioural domains. It looks very well conducted, but the scale of the data reported presents a clear challenge in how to report all of this in a concise manner. At present, I would argue the authors are attempting to do too much – some aspects do not have enough detail, and other data is so extensive as to be hard to interpret. My suggestion would therefore be for greater prioritisation of what the driving questions are, a clearer structure to break up the amount of data presented so that distinct points are easier to extract. There is a significant imbalance between the size of the Results and Discussion written section, and the volume of tables and figures; this is most evident in that the narrative synthesis is only given two sentences (although the Results largely read well). To do justice to the scale of this work, I wonder whether the authors should consider splitting this into two papers – there are multiple secondary analyses that could be of interest (e.g., comparisons across behavioural domains, differences in length, or control group etc).

Introduction

1) Final paragraph of the introduction. This paragraph as a whole came across as disjointed. More explanation and clearer links are needed at various points, including:
- Lines 75-76 – please provide some background as to what the previous systematic reviews have shown. Given that you are making the argument for the need for a combined version, it would be useful to report what level of effect these are showing, if the different domains appear to differ etc – this will help to justify why you think we need a combined study.

RESPONSE: We are grateful to Reviewer 2 for their positive feedback and the helpful comments. Before answering to the first remark, we would like to point out the following: We were able to add another search wave (literature published Feb 2023 to Jan 2024), which yielded 13 additional eligible RCTs, resulting in a total of 83 RCTs included in the meta-analysis. Consequently, we have re-run all analyses. Results remained very similar, which is why our conclusions remain (by and large) the same.

With regards to the first point of Reviewer 2, we agree and we have clarified the main rationale for the present work in the last paragraph of the introduction as follows (page 3):

"The potential of social comparison as a behaviour change technique (SC-BCT) to increase desired behaviour (e.g., recycling or sunscreen use)²⁰⁻²² or to decrease undesired behaviour (e.g., usage of finite resources or alcohol consumption)^{20,23,24} has been investigated in numerous RCTs and some domain-specific meta-analytic syntheses thereof. Attention to SC-BCTs has increased significantly during the last decades^{20,25-32}. Yet, a systematic review and meta-analysis on the use and efficacy of SC-BCTs across scientific disciplines is lacking. The present work attempts to fill this gap by means of a comprehensive systematic review and meta-analysis summarizing data from various scientific disciplines. We were interested in how social comparison was used as a BCT in randomised controlled trials (RCTs) and to what effect. As such, our work attempted to answer the following two main research questions: 1) How effective are SC-BCTs?; 2) How effective is SC-BCT as an add-on intervention when added to a bundle of BCTs?"

Due to the word limit, we were only able to report results of previous reviews in the discussion section as follows (page 10):

"The present results align with prior findings in related fields. Other meta-analyses that investigated the efficacy of SC-BCTs also revealed significant results on behaviour change, including the field of climate change mitigation behaviour^{14,25,28,31} and work performance³². In one review, for instance, SC-BCTs have been found to be one of the two most effective BCTs for climate change mitigation, alongside financial approaches²⁵."

- The argument about looking at moderators is compelling – would be good to make this link more clearly to the later sentences. E.g., lines 78-80 doesn't seem to link to the statement in 83-85, about how used. How do you see these two aspects linking?

RESPONSE: We decided against mentioning the moderator analyses in the introduction given that we had to cut the introduction to meet the word limit. We now report all information regarding moderator analyses in the method section (page 6). We also

mention in the discussion that our focus on moderators of efficacy extends previous work (page 10).

- Similarly, line 84 onwards seems to be a different point – and I could not see how it related to the aims. i.e., if researchers do not measure SC in a group intervention, can it be included in this review? It doesn't look to be relevant, so I wonder if this statement is better suited to the Discussion.

RESPONSE: We have moved this point to the discussion section, as suggested, and it now reads as follows (page 11):

"There are limitations to our study. First, we aimed at only including research that explicitly referred to social comparison or social comparison-related terms (e.g., upward/downward/lateral comparison). This choice was made in an effort to maximize internal validity. Yet, this may have resulted in missing related literature. For example, research on audit and feedback to influence health professional behaviour often involves interpersonal comparison (e.g., comparison with median performance) as one part of a BCT bundle¹³⁷. Likewise, any group-based intervention is likely to be influenced (partly) by social comparison processes. Yet, studies not explicitly referring to social comparison (recall that we performed all-fields searches) were deemed unlikely to entail a study design that enables the evaluation of the individual impact of SC-BCTs on behaviour change. Despite the restriction on research explicitly referring to social comparison terms, the present work covered a broad range of literature (83 RCTs) and a very large number of included participants (N = 1,357,062)."

2) It seemed like what you are describing is a study across behavioural domains, rather than across scientific disciplines – as the study of SC impact on cognitions, affect and behaviour is surely within the same discipline (i.e., behavioural science).

RESPONSE: We agree. In response to this comment, we changed the title as follows:

"Meta-analysis of randomised controlled trials examining social comparison as a behaviour change technique across behavioural sciences"

We have also generally updated the manuscript in this respect and we have clarified this point in the paragraph detailing the search methods as follows (page 5):

"No restrictions were made concerning scientific disciplines. Yet, all data turned out to be from behavioural sciences."

Methods

3) The order of presentation and some of the terminology of the methods was difficult to

follow. Use of overarching subheadings as per convention may help (e.g., a section on analyses, a section on intervention characteristics etc).

RESPONSE: We have re-ordered the methods and we have used more (and conventional) subheadings in response to this comment.

- The paragraph starting in line 105 seems to give one account of how you are defining SC-BCTs, and would be clearer to state what terms or descriptors you would accept and how you decided on these. I think you're saying that descriptions by the study authors of what was made explicit to participants by way of social comparison, would have to qualify for the definition provided by Michie et al in 6.2 of the taxonomy? You are also stating that you are explicitly excluding anything classified as 6.3 (i.e., injunctive norm). Did you use any specific instructions to help coders to do this, if so it would be useful to see the criteria here? Who and how was it decided whether an intervention met the criteria for this BCT?

RESPONSE: As stated on page 4, we followed Wood's (1996) definition of social comparison as "thinking about social information in relation to the self". We used Michie et al's taxonomy to decide whether a particular study can be labelled as a BCT. Now Michie et al. also provide a definition of SC-CBT, which is similar to Wood's (1996) definition of social comparison. Nonetheless, our primary source for defining social comparison was Wood's definition. Accordingly, if a study that Michie et al would define as an injunctive norm intervention met Wood's definition of social comparison, we would have included that study. However, pure injunctive norm interventions only make use of information about the extent to which some behaviour is approved or disapproved by others and hence do not meet Wood's criteria for social comparison. As such, these studies were not included. In line with Wood's (1996) definition of social comparison, the coders (all are authors of the present work) had the instruction to classify a BCT as a SC-BCT if the intervention instructed study participants to "think about social information in relation to the self". We have made this point clearer on pages 3-4, as follows:

"We primarily adhered to Wood's³ definition of social comparison as thinking about social information in relation to the self. Furthermore, we followed Morina's general comparative-processing model⁵ to conceptualize social comparison as a process. Lastly, we followed the taxonomy of BCTs portrayed by Michie et al.³⁴. Accordingly, BCTs are defined as observable, replicable, and irreducible interventions (i.e., stand-alone BCT) or components of a more complex interventions (i.e., bundle of BCTs) designed to have causal influence on behaviour change³⁴. Michie et al. defined SC-BCTs as interventions in which attention is drawn to the performance of others to enable social comparison on a particular dimension. Nonetheless, decision on whether a study used social comparison as an intervention was primarily based on Wood's definition of social comparison. For this comparison to occur, sufficient information needs to be provided, particularly for covert behaviour. For instance, an intervention on reducing energy usage needs to provide quantitative feedback about one's own energy usage and the usage of a given social standard. Accordingly, descriptive norm

interventions that supply such quantitative feedback belong to the category of SC-BCTs, whereas injunctive norm interventions (supplying information about the extent to which some behaviour is approved or disapproved by others^{32,35}) do not belong to SC-BCTs."

In addition, we have clarified who coded the data on page 3 as follows:

"The systematic literature search, data extractions, and statistical analyses were conducted by at least two authors independently (THH, RMC, JN, & FL). In cases of disagreement, consensus was reached among THH, RMC, and NM in personal discussions."

- The text from line 121 onwards is described as categorising BCTs, but this is confusing as BCTs can be just a small part of an intervention which combines them (as per your own definition in the previous paragraph), and the interventions you are comparing are more than just the BCTs but how they are implemented (i.e., the content and context). Perhaps this is arguable (i.e., I am used to people referring to a BCT as the 'pure' form of a theoretical technique, and an 'intervention' as the broader set of information on how BCTs are operationalised and combined). But at any rate is confusing to use the term BCT in both paragraphs. I would think it more in line with convention to say you are categorising interventions themselves at this point, i.e., on line 122 "SC interventions were categorised into three sub-categories: 1) social comparison as stand-alone SC-BCT (i.e., no other BCTs involved)..." etc. This would help to make these two paragraphs distinct (you use the term bundle, but I think the term intervention is more aligned with other research).

RESPONSE: We agree and have re-phrased this sentence as follows (page 4):

"Interventions were categorized in three sub-categories: 1) social comparison as stand-alone SC-BCT (i.e., no other BCT involved), 2), main SC-BCT (i.e., social comparison was the main intervention in an BCT bundle, e.g., alongside SC-BCT, information on how to reduce electricity usage was presented), or 3) add-on SC-BCT (i.e., SC-BCT was part of a BCT bundle and was not the main intervention in this bundle)."

- Line 130 - 140– The points made here fit better with the paragraph starting on line 105, as they refer to the definition of the BCT, not the design of an intervention (i.e., centrality of social comparison to a study).

RESPONSE: We agree and have moved these sentences, as suggested.

- Line 140 – this statement fits better in the analysis section.

RESPONSE: We agree and have moved the respective statement.

4) Line 128 – mixing BCTs may also enhance efficacy, so evidence does not fully support this point. Your reference is only for online apps, there are multiple papers from the Michie research group that show how BCTs are used and bring about effects in concert (and indeed are rarely used alone).

RESPONSE: We agree and in response have decided against citing Spohrer et al. (2021) and deleted this argument altogether, while keeping the respective sub-analysis.

5) Line 142 onwards – a broader section on outcomes would be useful. While you're not restricting the outcomes to a particularly domain, you are restricting them to cognitive, affective or behavioural outcome – so this would be good to state with greater explanation, not least, how you decided which outcome to pick. I see that later you talk about hierarchies, but there may also be multiple outcomes within a similar domain/hierarchy (e.g., 2 behavioural outcomes, or motivation plus self-efficacy etc). This section may be better combined with your presentation of the algorithm of hierarchies.

RESPONSE: We now provide examples for potential cognitive, affective, and behavioural outcomes as follows (page 5):

"(...); 3) data for at least one behavioural outcome (e.g., electricity usage), cognitive outcome (e.g., intention to change electricity usage), or affective outcome (e.g., feelings about changing electricity usage) was reported; (...)."

We have also adjusted the sub-heading for the paragraph on data prioritisation (page 5):

"Data prioritisation: Determining the primary outcome and primary comparison"

In this paragraph we have further clarified the algorithm when studies reported multiple outcomes and a choice was required on pages 5-6:

"For complex trials investigating multiple outcomes and/or multiple arms, we applied a data prioritisation algorithm to avoid data dependencies. When applicable, data were prioritised as follows: 1) When multiple outcomes were reported, but only one of them aligned with the social comparison dimension, this outcome was prioritised; 2) When several types of outcomes (i.e., behavioural, cognitive, & affective) aligned with the social comparison dimension, outcome types were prioritised in the following order (starting with the prioritised type): behavioural outcomes, cognitive outcomes, and then affective outcomes; 3) When both objective (e.g., objective step count) and subjective outcomes (e.g., subjective step count) were reported, objective outcomes were prioritised; 4) When a trial involved arms with different kinds of SC-BCTs, the primary SC-BCT arm (reported as such by the primary authors of the given trial) was prioritised. Nonetheless, data excluded from one analysis could still be included in other (sub-)analyses as long as data independence was met."

6) line 154 – It could be helpful to provide more information on how you judged the search terms to be sufficient, e.g., how do these compare with previous systematic reviews?

RESPONSE: We now report on page 4 that our choice of search terms was primarily based on Gerber et al. (2018; Psychological Bulletin):

"In line with the systematic search by Gerber et al.¹² who chose to conduct a broad search by using only the term "social comparison", our equally broad search used the category "all-fields" and included only terms related to social comparison (TX "social compar" OR TX upward comparison* OR TX downward comparison* OR TX "lateral comparison*")."*

We also inform the reader that further information about our search strategy can be found in Appendix A in the online supplementary material.

7) Line 177 – did you compute effect sizes from data available, where not reported by the authors themselves? I see later you go on to state this, but it looks like you don't at this point. This may be better combined with the later section at line 235 so it is less disjointed.

RESPONSE: Whenever (raw) means and SDs were reported in the publication (or sent via email), we computed effect sizes ourselves. In rare cases, the authors reported Hedges' *g* instead of means and SDs, which we then extracted. As recommended by the reviewer, we have combined the two pieces of information mentioned above. In response to this comment, we made this point clearer on page 6:

*"Hedges' *g* was calculated³⁹ whenever (raw) means and SDs were reported in the publication (or sent via email). In rare cases, publications reported Hedges' *g* instead of means and SDs, which we then extracted."*

8) Line 207 – what was the relative hierarchy of cognitive over affective outcomes?

RESPONSE: We clarified this point on page 5 as follows:

"(...) 2) When several types of outcomes (i.e., behavioural, cognitive, & affective) aligned with the social comparison dimension, outcome types were prioritised in the following order (starting with the prioritised type): behavioural outcomes, cognitive outcomes, and then affective outcomes; (...)."

9) Line 266 – please include the four items used to judge ROB in the main paper here. Fine to refer to appendices for detail, but the overview is needed.

RESPONSE: Based on the comments of the Editor and Reviewer 1, we have changed ROB assessment to RoB2.0 Cochrane criteria for RCTs. The criteria and their scoring are detailed for readers in Appendix B. The RoB2.0 assessments for all 83 RCTs are provided in Appendix C and a visualization thereof in Appendix D. We report on the ROB2.0 items in the methods as follows (page 6):

"Study quality was assessed by means of the Cochrane risk of bias tool 2.0³⁸, which assesses the methodological rigor of RCTs on the following five domains: D1) Bias arising from the randomisation process; D2) Bias due to deviations from intended intervention; D3) Bias due to missing outcome data; D4) Bias in measurement of the outcome; D5) Bias in selection of the reported result. Each (independent) RCT was evaluated on these 5 domains and received a score of 0 ("high risk of bias"), 1 ("some concern"), or 2 ("low risk of bias"). Sum scores (possible range 0-10) were used for moderator analyses concerning study quality. Appendix B provides a detailed overview of the scoring per domain including sub-domains. All risk of bias assessments are presented in Appendix C and a visualization thereof in Appendix D."

Results

10) I could not see any clear presentation of differences between the impact of SC on affective, cognitive and behavioural outcomes. Typically behaviour is more challenging to influence, and from Table 2 it looks like there were a lot of behavioural outcomes – and indeed the results are written as if you only analysed the outcomes for behaviour. Is this right? Please report proportion of outcomes in each of the three specified groups, and at some point refer to how each is dealt with and is included in the Results.

RESPONSE: We now specify the proportion per type of outcome at the beginning of the results section (page 7):

"Almost all trials assessed behavioural outcomes aligning with the social comparison dimension (95% or $k = 79$). Three trials (4%) exclusively assessed cognitive outcomes (i.e., body dissatisfaction⁸⁸, quality of life⁶⁸, & estimated breast cancer risk⁷⁴) and one trial (1%) only reported data for an affective outcome (i.e., gratitude⁷⁶)."

We have also clarified how we treated these types of outcomes for analysis in the paragraph on statistical analysis (page 6):

"Studies investigating behavioural, cognitive, and affective primary outcomes were pooled whilst the vast majority of trials investigated a behavioural outcome aligning with the comparison dimension (see more details in the results section)."

11) There is an impressive amount of data and calculations gone into this, but Table 2 is too large to be easy to follow. It would be useful for the authors to set out what their

lead findings are here – and prioritising the presentation of those. The remainder could be made supplementary. It may also be useful to split the table.

RESPONSE: This table can now be found in Table 1 (given that the trial characteristics Table was moved to the supplementary materials). Table 1 is now a lot shorter. This was achieved by splitting moderator results (see Table 2) as well as deleting redundant subheadings. Note that Table 2 is very short given that we only analysed moderators for the overarching analyses. We believe that tables (and results in general) are better to follow now.

12) Table 3 takes up a lot of space – and thus implies these findings should be given a lot of weight. Yet, these are not selected as they are more relevant or robust than the remainder, and given very minimal coverage in the results, so this seems a bit off balance. A shorter summary table of primary findings may be more appropriate with an expanded discussion.

RESPONSE: As mentioned above, these studies have been excluded from the present work and readers can find a (shortened) table in Appendix E detailing main study characteristics of these excluded trials.

13) Take care with interpretations about the type of intervention design, when considered separately from domain and/or other characteristic, unless you are also able to look at these with any other systematic differences. E.g. (looking at lines 362 and thereabouts) could it be that studies using stand-alone BCTs are typically targeting simpler behaviours, or different settings, for different populations than others? That is, be careful to caveat conclusions with clarity that these factors haven't been controlled for.

RESPONSE: We agree and now interpret more cautiously as follows (page 11):

"SC-BCTs involving social comparison as the stand-alone intervention were associated with significantly higher efficacy of behaviour change compared to SC-BCTs involving social comparison as the main intervention (e.g., next to providing information on how to reduce electricity usage). However, again, this moderation was only observed in the studies comparing to active (but not passive) controls and hence requires further investigation. Also, trials on stand-alone vs. main SC-BCTs might differ on third variables (e.g., behaviours studied, study settings, populations studied), which may explain the significant difference found. More research is needed."

14) Before reporting on Research question 1b, a summary of the range of active control conditions is needed.

RESPONSE: We have added this information on page 9 as follows:

"The most commonly applied active control condition concerned supplying intra-individual feedback on the given dimension allowing for temporal comparisons (e.g., comparisons with past own electricity usage). Other examples for active control conditions were shaping knowledge (e.g., instruction on how to lower electricity usage) or goal setting and planning (e.g., setting a goal concerning lowering electricity usage). See the 9th column of Appendix F for all active control conditions."

15) Line 392 – this is very informal language, and difficult to follow. If included, it needs to be more informative – it's not clear if the authors are saying there is not really enough evidence to draw a conclusion, or that given the evidence, we have at least enough to suggest there is no effect. Line 399 – again, please provide the number of trials rather than "relatively few".

RESPONSE: We agree and have updated the paragraph as follows (page 10):

"Appendix I contains all results. Number of trials was relatively low (largest $k = 9$). Across this thin evidence base, no evidence was found for significant differences in the short-term efficacy of upward SC-BCTs vs. downward SC-BCTs, upward SC-BCTs vs. not restricted SC-BCTs, or SC-BCTs presenting more vs. less attainable social comparison standards. At long-term, however, SC-BCTs presenting more (vs. less) attainable social comparison standards were superior ($P < .01$) with a small pooled effect ($g = 0.18$, 95% CI = 0.05 – 0.31; $k = 4$; $I^2 = 0\%$).

16) The comment on the narrative synthesis is too short to be meaningful. It isn't an adequate summary of the table – so again, if included, needs to be better presented to provide clear information on the findings and our confidence in them, and perhaps what this adds to the main meta-analytical findings.

RESPONSE: As mentioned above, we now exclude the narrative synthesis altogether.

Discussion

17) Line 414 – while you state that there are too few studies to warrant synthesis in service participation, does this include in the narrative synthesis. Can you tell us nothing about this area, or are there trends worth reporting?

RESPONSE: With the inclusion of the new search wave (literature up until Jan 2024) as mentioned above, we were able to synthesize data on service participation in isolation. The manuscript has been updated accordingly. The results concerning service participation behaviour were (also) significant (see Table 1).

18) Note: you refer only to behaviour change, please clarify (as per the comment above) if this is purposefully excluding findings for affect and cognition, or all you analysed.

RESPONSE: As mentioned above, we have clarified that behaviour change concerned mostly (95% of trials) behavioural outcomes (as well as three trials assessing cognitive outcomes and one trial an affective outcome). As mentioned above, we now clearly state that we analysed data across types of outcomes in the statistical analysis paragraph.

19) It was not clear to me if you looked at potential unintended consequences (e.g., by sharing norms, people change their behaviour in the direction opposite to that intended – which has been reported in some trials such as for alcohol consumption in students). Could this be made clearer?

RESPONSE: We added the following to the limitations (page 12):

"Fifth, results only concern collective behaviour (i.e., group mean differences). Future qualitative and quantitative research is necessary to investigate individual processes evoked by SC-BCT (e.g., investigating how many participants do vs. do not change behaviour and for which reasons), which may help in tailoring and optimizing SC-BCTs."

Reviewer #3:

Remarks to the Author:

Overview

This is an interesting and comprehensive review of the effectiveness of social comparison as a behaviour change technique. There are many strengths to this paper, particularly the thorough methodology. However, I recommend that the framing of the paper is improved to better demonstrate the key findings of the review and the importance of the review. Please see my more detailed comments regarding each section below.

Abstract

The abstract suggests that both seventy and 1,323,478 articles were reviewed. Please clarify that 70 articles were included after the initial literature results of 1,323,478 articles.

RESPONSE: We are grateful to Reviewer 3 for their positive feedback and helpful comments. Please note that we were able to add another search wave (literature published Feb 2023 to Jan 2024). This additional search wave yielded 13 additional eligible RCTs, resulting in a total of 83 RCTs included in the meta-analysis. Consequently, we have re-analysed all analyses. Results remained very similar, which is why our conclusions remain (by and large) the same.

These 83 independent RCTs included a total of 1,357,062 participants. That is, N stands for “number of total independent participants across trials” rather than “number of hits screened”. The abstracts now (after the update) reads (page 2):

“Eighty-three RCTs (N = 1,357,062) were included (...).”

And in the results (page 7):

“The 83 RCTs involved a total of N = 1,357,062 participants.”

Methods

Please specify which PRISMA guidelines (e.g., PRISMA 2020) were followed for this review/

RESPONSE: We now specify on page 3 that we followed PRISMA 2020 guidelines.

Was there a process for resolving disagreements between the reviewer? For example, was there a group discussion with consensus voting? A little detail would be useful to improve the clarity and replicability of the research.

RESPONSE: We now clarify on page 3 how we handled disagreements:

“In cases of disagreement, consensus was reached among THH, RMC, and NM in personal discussions.”

The diversity of literature included is commendable (e.g., including English, Dutch and German articles). If space allows, could the authors clarify if the search terms and operators were the same for each database? This would help with replicability.

RESPONSE: We now clarify on page 4 as follows:

“Languages of publications were limited to English, Dutch, and German and searches were carried out with English search terms only. The full search strategy is shown in Appendix A.”

The inclusion criteria of sufficient statistical power is sensible, but I wonder if this excludes qualitative research. Is there a reason that qualitative research was not included? There may be a valid reason for not including qualitative research in the review, but I would mention it as a potential limitation in the discussion section.

RESPONSE: Given our main aim of performing a meta-analysis summarizing the efficacy of SC-BCTs in RCT, we excluded qualitative research (see also our inclusion criteria on page 5 and the primary outcome metric on page 6). In response to this comment, we now report on page 12 in the limitations section:

“Fifth, results only concern collective behaviour (i.e., group mean differences). Future

qualitative and quantitative research is necessary to investigate individual processes evoked by SC-BCT (e.g., investigating how many participants do vs. do not change behaviour and for which reasons), which may help in tailoring and optimizing SC-BCTs."

Results

The results are explained thoroughly throughout.

Discussion

Throughout the paper, I was waiting for the 'here is why this research is important' statement and I was hoping to find it in this section. The authors have conducted a comprehensive review of the literature in great detail, but the importance of it needs to be made clearer. For example, when the authors mention that their results align with and extend prior findings in related fields, they could make explicit what new knowledge their review brings. The key take-home messages seem to be on page 11 when stating that 1) SC-BCTs can be effective in changing behaviour, and 2) that the effect might be somewhat overestimated. The authors discuss these findings but the importance of them is missing. I recommend that the authors foreground these findings and their importance concretely. This will help to more clearly show both the key findings of the research, the authors' novel contributions, and the messages that the authors want the reader to take from the paper. Please also make it clear how the results relate to the research questions which were laid out at the start of the introduction.

RESPONSE: We agree and now report on the key results and their implications with more clarity, within the permitted word limit (pages 10-11):

"The present work extends prior work by performing various moderator analyses. In the overarching analysis comparing SC-BCTs to active controls, number of SC-BCT sessions was positively associated with efficacy, suggesting a dose-response relationship, which might be attributable to cumulative reinforcement, habit formation, or cognitive or emotional shifts, fostering sustained engagement and internalization of change. However, in the overarching analysis comparing SC-BCTs to passive controls, the number of SC-BCT sessions did not moderate efficacy. Consequently, to clearly discern when and under what circumstances a dose-response relationship can be assumed, further research is needed. Similarly, the significant negative moderation of study quality in the overarching analysis comparing SC-BCTs to active (but not passive) control conditions suggests that effects are overestimated. Pooled effects are biased by lower-quality evidence finding larger effects. Future research needs to improve methodological quality to ensure accurate estimation of effects. As more trials accumulate over time, future meta-analytic research should re-evaluate the present finding, especially as the present work only found this association in one analysis (but not the other). When comparing SC-BCTs to active (but not passive) conditions, SC-BCTs targeting desired behaviours were associated with higher efficacy compared to SC-BCTs targeting undesired behaviour. This suggests that SC-BCTs might be more effective when they target a desired behaviour (e.g., healthy diet) rather than an undesired behaviour (e.g.,

unhealthy diet). However, given mixed findings across the moderator analyses relative to passive and active controls, this association remains preliminary and requires further testing. SC-BCTs involving social comparison as the stand-alone intervention were associated with significantly higher efficacy of behaviour change compared to SC-BCTs involving social comparison as the main intervention (e.g., next to providing information on how to reduce electricity usage). However, again, this moderation was only observed in the studies comparing to active (but not passive) controls and hence requires further investigation. Also, trials on stand-alone vs. main SC-BCTs might differ on third variables (e.g., behaviours studied, study settings, populations studied), which may explain the significant difference found. More research is needed. Lastly, trials directly comparing upward SC-BCTs portraying a more vs. less attainable social comparison standard were associated with higher long-term behaviour change, whereas behaviour change at short-term did not differ significantly. This suggests that more (vs. less) attainable upward SC-BCTs might yield longer lasting behaviour change. Given that the number of trials was rather low, also this association should be interpreted with caution and re-examined in future research.

Overall, the small effect sizes found for SC-CBTs need to be interpreted in the context of low cost in developing and disseminating SC-BCTs, making them realistic options for large-scale intervention. For instance, in various studies only one or two letters or emails with social comparison information (e.g., energy usage of target and standard) were sent to thousands of participants. Small effects according to statistical benchmarks³⁷ might nevertheless have large real-life impact when costs are low and scalability is large."

Other

Typo: Page7 line 218 'meaningful'.

RESPONSE: We corrected this typo.

General point about the structure: Please split up the paragraphs a little more for ease of reading. I found it a little difficult to identify the main arguments with the current long paragraph structure.

RESPONSE: We re-structured the manuscript in response to this comment.

Editorial comments:

1. Reviewer 1 raises important concerns about biases introduced due to the classification used in the RoB 2.0 tool. We ask that you thoroughly address these concerns and revise the classification. Please also follow Reviewer 4's advice and provide a visualisation of the risk of bias.

RESPONSE: We are grateful to you and the reviewers for the helpful comments that helped to further improve our work. We agree with you and Reviewer 1 regarding the classification used in RoB 2.0 tool and have updated it accordingly. In response to Reviewer 4's feedback, we now report the visualisation of the risk of bias in Figure 4 (previously reported in Appendix D).

2. Reviewer 4 asks that you provide more information on how BCTs were coded, and which occurred alongside SC. In addition, the reviewer argues that SC cannot be considered a BCT in examples where the intended outcome was not behaviour change. We agree with the reviewer and ask you to remove these trials from the analysis.

RESPONSE: In response to Reviewer 4's feedback, we now report the coding of SC-BCTs in more detail (pp. 3-4, see our response to Reviewer 4's comment below for more details). We also agree with the reviewer on the latter point and have adjusted our work accordingly. That is, we have removed the four trials not assessing behavior outcomes and re-analysed the data. The results and conclusions remained very similar.

3. Please use GRADE to summarise the certainty of evidence as requested by Reviewer 4.

RESPONSE: We now summarise the certainty of evidence using GRADE criteria (see our response to Reviewer 4's last comment for more details).

REVIEWER COMMENTS:

Reviewer #1 (Remarks to the Author):

I appreciate the extensive additional work conducted by the authors in response to the three peer review reports. Most issues have been satisfactorily addressed.

My one outstanding concern relates to the risk of bias assessment. The switch to using the Risk of Bias 2.0 tool is an appropriate choice, and I appreciate how much work has gone into assessing this large group of included studies. However, the authors have incorporated a novel element of their own design in converting these assessments to numerical scores, which is in contradiction to the guidance provided for this tool. The reference given at the end of Appendix B for the basis of the scoring system contains no information related to numerical scoring, and instead presents a clear algorithm for

converting the domain-level assessments to an ordinal overall conclusion of either low risk, some concerns or high risk, which the authors appear to have applied accurately in the final column of the figure in Appendix D.

The RoB 2.0 tool domains are not intended to be additive - a serious concern in any single domain is sufficient to render the result at high risk of bias. As such, a study rated at overall high risk of bias using RoB 2.0 could score anything between 0 and 8 using the authors' scoring system, and a study rated overall as having some concerns may score between 5 and 9. This overlap can be clearly seen in comparing the studies that scored 7 in Appendix C with the overall rating in Appendix D, where these studies were sometimes rated overall at high risk and sometimes at some concerns. I would suggest that the addition of domains into a numerical scale is an inappropriate interpretation of the risk of bias assessments, and should not be analysed as a continuous variable in the analysis. In the text of the paper, reporting the mean score across studies as 7 points and classifying this as "moderate" (which is not a category in this rating system), could also be misleading.

I would suggest that a better approach would be to use the overall ordinal classification in the final column of Appendix D, and to remove the references to numerical scores. If the authors decide to retain the numerical scoring system, it should be made transparent that this is their own method and not a component of the RoB 2.0 tool.

RESPONSE: We thank the reviewer for their positive feedback and for supplying additional comments that helped to further improve our manuscript.

We agree with the reviewer and adjusted the RoB assessments as suggested. That is, we restricted RoB assessments to the common ordinal scale ("low risk", "some concerns", and "high risk"). All previous information on numerical scoring (as well as the corresponding moderator analysis concerning RoB) has been removed from the manuscript. In response to this point, we have moved the visualization of the risk of bias from Appendix D to Figure 4).

Reviewer #4 (Remarks to the Author):

The is a comprehensive piece of research, albeit with relatively modest implications due to the small effect sizes and heterogeneity of behaviours that are targeted. However, it is still a useful and interesting contribution to readers and the wider community of behavioural sciences.

As this manuscript has already undergone three detailed reviews, I have highlighted only the things that I think are important that were not mentioned originally.

- RQ2 would be clearer to the reader as something like 'how effective are interventions which contain SC-BCT alongside other BCTs'

RESPONSE: We agree and have re-phrased RQ2 as suggested (p. 3):

"... and 2) how effective are interventions that incorporate a SC-BCT alongside other BCTs?"

- The inclusion criteria allowed for the outcome to be cognitive or affective, and the authors covered all of this under the term 'behavioural outcome'. In my opinion, SC cannot be considered a BCT in examples where the intended outcome was not behaviour change. As there are only four trials, I would remove these.

RESPONSE: We agree and have deleted the four trials in question and have re-analyzed the data and updated the manuscript accordingly. As such, we also rephrased the respective inclusion criterion (p. 5):

"... 3) data for at least one behavioural outcome (e.g., electricity usage) were reported; ..."

The results and conclusions remained very similar.

- There is no information in the method or results about how other BCTs were coded, and which ones occurred alongside SC. If this was not done in a robust manner, is there a chance that the studies were 'under-coded'? Relatedly, it is quite surprising that so many studies only tested the efficacy of a single BCT for quite complex behaviours related to domains such health and the environment. Normally complex interventions feature several BCTs. I am currently finishing a review focusing on another single BCT and there are no studies that isolate only that BCT (i.e., the interventions always feature other BCTs as well). This leads me to question whether the studies in this review that only contain SC-BCT are different from the others. For example, are they conducted in a lab setting, and therefore, less ecologically valid?

RESPONSE: We have expanded our description on the coding of SC as a BCT as well as BCTs alongside SC (pp. 3-4):

"We primarily adhered to Wood's³ definition of social comparison as thinking about social information in relation to the self. Furthermore, we followed Morina's general comparative-processing model⁵ to conceptualize social comparison as a process encompassing selection of the social comparison standard, the basic comparison process itself (i.e., evaluation of similarity or discrepancy between target and social standard), and resulting reactions. Lastly, we followed the definition and taxonomy of BCTs portrayed by Michie et al.³⁴ Accordingly, BCTs are defined as observable, replicable, and irreducible interventions (i.e., stand-alone BCT) or components of a more complex interventions (i.e., bundle of BCTs) designed to have causal influence on behaviour change³⁴. Michie et al. defined SC-BCTs as interventions in which attention is drawn to the performance of others to enable social comparison on a particular dimension. Nonetheless, our decision on whether a study used

social comparison as an intervention was primarily based on Wood's definition of social comparison. For this comparison to occur, sufficient information needs to be provided, particularly for covert behaviour. For instance, an intervention on reducing energy usage needs to provide quantitative feedback about one's own energy usage and the usage of a given social standard. Accordingly, descriptive norm interventions that supply such quantitative feedback belong to the category of SC-BCTs, whereas injunctive norm interventions (supplying information about the extent to which some behaviour is approved or disapproved by others^{32,35}) do not belong to SC-BCTs. Interventions were categorized in three sub-categories: 1) social comparison as stand-alone SC-BCT (i.e., no other BCT involved), 2) main SC-BCT (i.e., social comparison was the main intervention in an BCT bundle, e.g., information on how to reduce electricity usage was presented alongside SC-BCT), or 3) add-on SC-BCT (i.e., SC-BCT was part of a BCT bundle and was not the main intervention in this bundle). Sub-categories 1) and 2) concern the first research question and sub-category 3) concern the second research question. Categorizations of BCTs were based on Michie et al.'s taxonomy and conducted independently by two of the authors (THH & RMC). Discrepancies were discussed amongst three authors (THH, RMC, & NM) until consensus was reached."

We further clarify other BCTs (e.g., active control conditions) on p. 10 as follows.

"The most commonly used active control condition involved the provision of intra-individual feedback on given dimension in question, allowing for temporal comparisons (e.g., comparisons with past personal electricity usage). Other examples for active control conditions were shaping knowledge (e.g., instruction on how to lower electricity usage) or goal setting and planning (e.g., setting a goal concerning lowering electricity usage). See Appendix D (column 9) for all active control conditions."

To respond more directly to the two concerns raised by the reviewer: First, we confirm that numerous studies had used SC as a stand-alone BCT (e.g., sending a short letter or email containing only SC information, such as a) personal energy consumption vs. b) energy consumption of social comparison standard). As stated more clearly in the revised manuscript now, the first two authors independently conducted the categorizations of BCTs and compared codings thereafter. Disagreements were solved in personal discussions amongst the first two authors and the last author until consensus was reached, obeying a standard coding procedure in meta-analytic research. Second, only 12 trials (15% of all included trials) were performed in lab settings (as described on p. 8 in the manuscript) and our inclusion criterion regarding the timing of outcome assessments (i.e., at least 24hours after [first, if multiple] SC induction) ensured that the "behaviour change assessment" was not restricted to the lab setting for these 12 trials either. By means of excluding trials assessing immediate behaviour change (e.g., behaviour change in lab settings), we aimed at maximizing ecological validity. The fact that 85% of included trials induced the SC-BCTs outside of laboratory settings, such as using apps or sending SC-BCT letters/emails, enhances confidence in external validity. We report the other settings in detail in the manuscript (p. 8):

"Thirty-eight trials (48%) used online methods (e.g., email or apps such as leaderboards) to apply SC-BCTs. In 12 trials (15%), the SC-BCT was applied personally in a lab setting. Another 16 trials (20%) sent SC-BCT-letters home. Other studies sent SC-BCT-letters to schools (k = 7; 9%) or work environments (k = 5; 6%). One trial (k = 1; 1%) applied the SC-BCT in a hospital (i.e., to inpatients before discharge)⁹⁶."

- Risk of bias (RoB) coding results are normally presented in a visual RoB figure, which is built into the RoB2 tool from Cochrane. It would be ideal if this was in there.

RESPONSE: In the previous version of our manuscript, the RoB figure was part of the supplements (previously: Appendix D). In response to this comment, we have moved the RoB figure to the main manuscript (now: Fig. 4).

- The first paragraph of the discussion should mention the small nature of the effects across all meta-analysis variations, particularly as the authors state that lower quality studies probably inflated pooled effects. The authors note that small effects might still be able to produce meaningful change, because of scalability and cost. For this point to be more powerful the focus should be on the health behaviours (using an example or two), rather than the examples such free throw shooting and other performance related behaviours, which are less compelling (and important to readers).

RESPONSE: We have adjusted the first paragraph of the discussion as suggested (pp. 12-13):

"This work reports a comprehensive meta-analysis covering data from RCTs on the efficacy of SC-BCTs across behavioural sciences. In 79 RCTs, we found evidence supporting the efficacy of SC-BCTs in shaping behaviour in the desired direction, albeit with small magnitude of pooled effects. (...) However, certainty of evidence was often limited, mainly due to concern about risk of bias and considerable unexplained heterogeneity. More high-quality research is necessary to further examine the robustness of the results found in the present work. (...) Overall, the small effect sizes found for SC-BCTs need to be interpreted in the context of low cost in developing and disseminating SC-BCTs, making them realistic options for large-scale implementation (e.g., for preventive health interventions). For instance, in various studies only one or two letters or emails with social comparison information (e.g., personal energy usage vs. that of a social standard) were sent to thousands of participants, or scalable low-cost digital health interventions utilizing peer comparison or leaderboards were applied. Small effects according to statistical benchmarks³⁷ might nevertheless have large real-life impact when costs are low and scalability is large."

- Overall, the authors have responded well to reviewer comments with a more streamlined article. However, I do not think that the authors have responded to the reviewer 1 comment about using GRADE. This is an appropriate suggestion and the

authors have either misunderstood it or not done it. Summarising the certainty in evidence using GRADE would not take much more work for the authors as they have completed most of the work in separate sections, but it would more clearly frame the findings.

RESPONSE: We agree and we now report the certainty of evidence using GRADE.

On page 6 we describe the GRADE methodology:

"Certainty of evidence

Certainty of evidence was assessed using GRADE criteria³⁹ via the following five domains: a) risk of bias (e.g., pooled effect is mainly based on studies with insufficient randomisation), b) inconsistency (i.e., unexplained heterogeneity), c) indirectness (e.g., pooled effect is mainly based on interventions that were examined in a particular sub-sample without providing a rationale for selective inclusion), d) imprecision (i.e., the confidence interval does not allow a firm conclusion about the effect and its direction) and e) publication bias. Assessment of risk of bias was derived from risk of bias 2.0 assessments (see above). Indirectness was assessed across four domains proposed by GRADE: population, intervention, comparator, and outcome. Evaluation of heterogeneity, imprecision and publication bias were derived from the results of the meta-analytic analyses (see below). Interpretation of I^2 corresponds to the recommended classification by GRADE (e.g., 75% to 100%, considerable heterogeneity). Certainty of evidence overall can range from high (4) to very low (1). As recommended, certainty of evidence was only assessed for the overarching analyses and not repeated for sub-analyses."

In the results section, we describe the certainty of evidence for overarching analyses in detail (e.g., pp. 8-9):

"Certainty of evidence was rated as low due to concern about risk of bias and considerable significant unexplained heterogeneity between outcomes. Risk of bias emerged from insufficient reporting of the randomisation process, deviations from the intended intervention (e.g., study participants or interventionists were likely to be aware of the assigned intervention), missing outcome data (e.g., data were not available for all randomised participants) or lack of pre-defined analysis protocols in several studies. Significant unexplained heterogeneity was addressed by multiple sub-analyses. Non-important heterogeneity was found for four sub-analyses: intended upward SC BCTs (across outcomes), desired outcomes only, health outcomes only, and performance outcomes only. No concern regarding indirectness emerged from the selection of the population (i.e., the majority of trials presented directness regarding population, whereas ten trials presented probable indirectness regarding population because they investigated a relatively specific sample without providing a rationale for its selection, e.g., examining the efficacy of a SC-BCT for climate change mitigation behaviour in a student sample). No concern about

indirectness were identified in relation to intervention (i.e., social comparison and SC-BCT definitions were always met, see inclusion criteria), comparator (i.e., assessment only in passive control conditions) and outcome (i.e., all behavioural outcomes matched the given social comparison dimension in SC-BCTs). All directness (vs. indirectness) ratings of evidence for each included RCT using GRADE criteria are detailed in Appendix E (i.e., for this analysis as well as all other overarching analyses). The confidence interval for the pooled effect excluded the null, signaling confidence in a significant behavior change in the desired direction."

We discuss limited certainty of evidence in the discussion twice; Pp. 12-13:

"However, certainty of evidence was often limited, mainly due to concern about risk of bias and considerable unexplained heterogeneity. Further high-quality research is needed to thoroughly examine the robustness of the findings in the present work."

P. 14 (last sentence of the discussion):

"Certainty of evidence was constrained by various sources of bias in the current literature, highlighting the need for more high-quality research to more robustly examine the efficacy of SC-BCTs in future research and meta-analytic syntheses."

REVIEWER COMMENTS:

Reviewer #1 (Remarks to the Author):

Thanks again for the considerable amount of work the authors have done in response to the peer review comments. I have no further comments.

RESPONSE: We thank Dr. Cumpston for her time, effort, and the valuable feedback provided during this peer review process.

Reviewer #4 (Remarks to the Author):

Overall, the authors have responded well to the additional reviewer comments, particularly regarding RoB and GRADE assessments, and clarification on additional BCT coding. The GRADE in particular was a fair amount of additional work so this effort to revise the manuscript is noted. From my perspective there is only one minor outstanding issue to clarify.

Although it is good to know that only a relatively small percentage of studies were lab-based, I am still not convinced of the claims that there are as many stand-alone SC-BCT interventions as currently stated. The authors provide an example of 'sending a short letter or email containing only SC information, such as a) personal energy consumption vs. b) energy consumption of social comparison standard'. In this example, the participants are being provided with feedback on behaviour by being given a breakdown of their energy consumption, which is another BCT (2.2 in the BCTTv1). To be a stand-alone SC-BCT intervention, the letter or email would not contain information on their personal energy consumption, only information about how their usage compared to reference group. For example, 'your energy consumption was 10% higher than the average household in your area'.

In addition, a quick look at the first few examples of papers that have been classified as stand-alone SC-BCT interventions confirms this.

- Brülisauer et al. (2020) provide feedback in behaviour (2.2) in the intervention groups (mentioned in the title and detailed in Table 1).
- Chapman et al. (2016), study 1, there is more than one additional BCT. The participants are asked to self-monitor their steps (2.3) and are set goals (1.1), which have a frequency and are therefore also an action plan (1.4). If the authors only included study 2 of this paper, then please highlight this in the study characteristics table.
- In Gonçalves et al. (2018), participants appear to receive feedback on their behaviour after every 10 trials (2.2).

Please can the authors elaborate on some of these details.

RESPONSE: We thank Dr. Howlett for his time, effort, and the valuable feedback provided during this peer review process. The manuscript has significantly improved due to his feedback, for which we are highly grateful. With regards to the point raised by Dr. Howlett, we largely agree and we have revised the manuscript accordingly, as detailed below. However, we respectfully disagree with the following statement: *"To be a stand-alone SC-BCT intervention, the letter or email would not contain information on their personal energy consumption, only information about how their usage compared to reference group. For example, 'your energy consumption was 10% higher than the average household in your area'."*

We contend that whether social comparison information is presented in relative terms, as in Dr. Howlett's example (i.e., *'your energy consumption was 10% higher than the average household in your area'*) or in absolute terms (e.g., *'your energy consumption was X and that of the average household was Y for period Z'*), both align with Wood's (1996) definition of social comparison, namely thinking about social information in relation to the self. Please note that we explicitly state in the introduction (e.g. p. 4) that *our decision on whether a study used social comparison as an intervention was primarily based on Wood's definition of social comparison.*

Nonetheless, we fully acknowledge Dr. Howlett's concern that any SC-BCTs providing intra-individual feedback (e.g., one's own electricity usage) at least twice enables participants to engage in temporal comparisons in addition to social comparisons. Accordingly, we agree that the intervention should not be classified as a stand-alone SC-BCT. We also acknowledge that this aspect required further clarification and have revised the manuscript accordingly (pp. 3-4):

"We primarily adhered to Wood's³ definition of social comparison as thinking about social information in relation to the self. Furthermore, we followed the general comparative-processing model⁵ to conceptualize social comparison as a process encompassing selection of the social comparison standard, the basic comparison process itself (i.e., evaluation of similarity or discrepancy between target and social standard), and resulting reactions. Lastly, we followed the definition and taxonomy of BCTs portrayed by Michie et al.³⁴ Accordingly, BCTs are defined as observable, replicable, and irreducible interventions or components of a more complex intervention designed to have causal influence on behaviour change³⁴. Michie et al. defined SC-BCTs as interventions in which attention is drawn to the performance of others to enable social comparison on a particular dimension. Nonetheless, our decision on whether a study used social comparison as an intervention was primarily based on Wood's definition of social comparison. For this comparison to occur, sufficient information needs to be provided, particularly for covert behaviour. For instance, an intervention on reducing energy usage needs to provide quantitative feedback about one's own energy usage and the usage of a given social standard. Accordingly, descriptive norm interventions that provide such quantitative feedback belong to the category of SC-BCTs, whereas injunctive norm interventions (i.e., interventions providing

information on the degree to which a particular behaviour is socially approved or disapproved^{32,35}) do not belong to SC-BCTs. Notably, the efficacy of SC-BCTs may be accompanied by other potentially contributing factors, even when trialists aim to explicitly and solely focus on the efficacy of SC-BCTs. For instance, if an intervention provides quantitative feedback on an individual's performance (alongside the social standard) multiple times, it enables not only social comparison but also temporal comparison (i.e., intra-individual comparison with past performance). Similarly, an intervention may include a SC-BCT accompanied by a frowning face or a downward-pointing thumb, introducing additional potential influences on behaviour change processes. As a result, other factors could potentially interfere with the efficacy of SC-BCTs, preventing them from functioning as the stand-alone intervention. Regarding our first research question on the efficacy of SC-BCT, we focused on trials examining SC-BCT either as the stand-alone BCT or as the primary BCT."

Reviewer #4 (Remarks to the Author):

I appreciate the authors' considered response to the issue of defining stand-alone SC-BCTs. I note the changes in the method, and that references to this have been removed from the results section (and supplementary tables) altogether.

Apologies for dragging this issue on a bit, but I am left with the question of whether there are any interventions that would be classified as 'stand-alone' now that the definition has been changed. This information has now been removed from Appendix C and D, so the reader cannot tell what the split is between studies that have SC as the stand-alone BCT or as the main behaviour change intervention.

RESPONSE: We highly appreciate Dr. Hawlett's thorough review and thank him once again for his valuable time and effort invested in the peer review process, which significantly contributed to improving the present work. We agree that the remaining point is informative and have therefore added the following to the results section (p. 8):

"A total of 71 RCTs investigated the efficacy of SC-CBTs as the primary intervention (i.e., the first research question), whereas 8 RCTs^{73,76,83,99,101,104,105,110} investigated the efficacy of SC-BCTs as an add-on intervention (i.e., the second research question). Notably, one dismantling RCT investigated both research questions¹¹⁰. Of those assessing the efficacy of SC-CBTs as the primary intervention, 14 independent RCTs reported in 13 publications^{21,64-66,74,75,79,86,87,91,92,97,100} investigated SC-BCTs as a stand-alone BCT. The remaining 58 independent RCTs reported in 54 publications^{22,24,51-63,67-72,77,78,80-82,84,85,88-90,93-96,98,102,103,106-121} investigated SC-BCTs in conjunction with other BCTs. Provision of intra-individual feedback on the behavioural dimension, alongside social feedback on the same dimension, delivered to participants on two or more occasions (i.e., enabling both temporal and social comparison) was the most common complementary BCT accompanying the SC-BCT among the 56 RCTs investigating SC-BCT as the primary (but not stand-alone) intervention."

Furthermore, we have added the following sentence to the discussion section (p. 13):

"Notably, the vast majority of studies did not investigate SC-BCTs as a stand-alone intervention, but rather as the primary BCT accompanied by another BCT, such as the provision of repeated intra-individual feedback enabling temporal comparison in addition to social comparison."